# ADAPTIVE BACKTRACKING LINE SEARCH

**Joao V. Cavalcanti[1], Laurent Lessard[2] & Ashia C. Wilson[1]**
[1] MIT, [2] Northeastern University
{caval,ashia}@mit.edu, l.lessard@northeastern.edu

## ABSTRACT

Backtracking line search is foundational in numerical optimization. The basic idea is to adjust the step-size of an algorithm by a *constant* factor until some chosen criterion (e.g. Armijo, Descent Lemma) is satisfied. We propose a novel way to adjust step-sizes, replacing the constant factor used in regular backtracking with one that takes into account the degree to which the chosen criterion is violated, with no additional computational burden. This light-weight adjustment leads to significantly faster optimization, which we confirm by performing a variety of experiments on over fifteen real world datasets. For convex problems, we prove adaptive backtracking requires no more adjustments to produce a feasible step-size than regular backtracking does. For nonconvex smooth problems, we prove adaptive backtracking enjoys the same guarantees of regular backtracking. Furthermore, we prove adaptive backtracking preserves the convergence rates of gradient descent and its accelerated variant.

## 1 INTRODUCTION

We consider learning settings that can be posed as the unconstrained optimization problem

$$\underset{x \in \mathbb{R}^d}{\arg\min} \ F(x). \tag{1}$$

Typically, algorithms solve (1) iteratively, refining the current iterate $x_k$ by taking a step $\alpha_k d_k$:

$$x_k + \alpha_k d_k. \tag{2}$$

Here, $\alpha_k$ is the size of the step taken in the direction $d_k$. Examples of iteration algorithms include gradient descent (GD), Newton's method, quasi-Newton methods (Moré & Sorensen, 1982; Nocedal & Wright, 2006), Nesterov's accelerated gradient method (AGD) (Nesterov, 1983), adaptive gradient methods (Ruder, 2016) and their stochastic and coordinate-update variants (Boyd et al., 2011). To find an appropriate step-size, iterative algorithms typically call a *line search* (LS) subroutine, which adopts some criterion and adjusts a tentative step-size until this criterion is satisfied. For many popular criteria, if the direction $d_k$ selected by the base algorithm is somewhat aligned with the gradient of $F$, then a feasible step-size can be produced in a finite number of updates by successively reducing an initial tentative step-size until the criteria are satisfied. The standard practice for this process, known as *backtracking*, is to multiply the tentative step-size by a *predefined constant factor* to update it. We propose a simple alternative to standard practice:

*to adjust the step-size by an **online variable** factor that depends on the line search criterion violation.*

While in principle this idea can be applied to many criteria, this paper focuses on illustrating it in the context of two line search criteria: the Armijo condition (Armijo, 1966), arguably the most popular example of such criteria, and the descent lemma (Bertsekas, 2016, proposition A.24) in the context of composite objectives (Beck & Teboulle, 2009). After motivating our choices of online adaptive factors, we show that they enjoy the best theoretical guarantees one can hope for. Moreover, we prove that adaptive backtracking preserves the convergence rates of GD and AGD. To conclude, we present numerical experiments on several real-world problems confirming that using online adaptive factors in line search subroutines can *produce higher-quality step-sizes* and *significantly reduce* the total number of function evaluations standard backtracking subroutines require.

**Contributions.**   Our contributions can be summarized as follows.

- In Section 2, we propose a template for adaptive backtracking procedures with broad applicability.
- In Section 2.4, we apply the template to enforce the Armijo condition and, in Section 3, present experiments on real-world problems showcasing that the adaptive subroutine outperforms regular backtracking and improves the performance of standard baseline optimization algorithms.
- In Section 2.5, we apply the template on proximal-based algorithms to satisfy the descent lemma and present more real-world problems in Section 3 illustrating that the adaptive subroutine improves the performance of FISTA.
- For both subroutines, in Section 4 we prove that for convex problems, adaptive backtracking takes no more function evaluations to terminate than regular backtracking in any single iteration. We also give global theoretical guarantees for adaptive backtracking in nonconvex smooth problems, which match those of regular backtracking. Furthermore, we show that adaptive backtracking preserves the convergence rates of gradient descent and its accelerated variant. In particular, the proof of accelerated convergence is based on a novel technical argument, to the best of our knowledge.

## 2   ADAPTIVE BACKTRACKING

### 2.1   LINE SEARCH: CRITERIA AND SEARCH PROCEDURES

Every line search subroutine can be decomposed into the *criteria* that it enforces and the *procedure* it uses to return a feasible step-size. We now briefly discuss each component and provide examples.

**Criteria.**   The most popular of line search criteria is the Armijo condition (Armijo, 1966), which requires that the objective function sufficiently decrease along consecutive iterates. Other popular sets of criteria are the weak and strong Wolfe conditions (Wolfe, 1969), which comprise the Armijo condition and an additional curvature condition that prevents excessively small step-sizes and induces step-sizes for which the objective function decreases even more. In contrast, nonmonotone criteria (Grippo et al., 1986; Zhang & Hager, 2004) only require that some aggregate metric of the objective function values (e.g., an exponential moving average) decrease along consecutive iterates.

**Search procedures.**   The second component of a line search subroutine is the procedure that finds a step-size satisfying the target criteria. For example, Wolfe line search is often implemented by bracketing procedures based on polynomial interpolation (Nocedal & Wright, 2006, pp. 60–61). In contrast, several criteria consisting in a single condition such as Armijo and nonmonotone, are provably satisfied by sufficiently small step-sizes. For them, the standard procedure to compute step-sizes fixes an initial tentative step-size and then consecutively multiplies it by a constant $\rho \in (0, 1)$ until the criteria are satisfied. This procedure is generally known as *backtracking line search* (BLS).

### 2.2   ADAPTIVE BACKTRACKING

BLS often enforces an inequality that is *affine* in the step-size. In this case, BLS can be reformulated as computing $v(\alpha_k)$, which is less than 1 when the line search criterion evaluated at the tentative step-size $\alpha_k$ is violated, and then scaling $\alpha_k$ by a factor until $v(\alpha_k)$ is greater than 1. BLS (Algorithm 1) employs a fixed factor $\rho \in (0, 1)$. We propose a simple modification of this procedure:

*to replace $\rho$ with an adaptive factor $\hat{\rho}(v(\alpha_k))$ chosen as a nontrivial function of the violation $v(\alpha_k)$.*

---

**Algorithm 1** Backtracking Line Search

**Input:** $\alpha_0 > 0, v \colon \mathbb{R}_+ \to \mathbb{R}, \rho \in (0, 1)$
**Output:** $\alpha_k$
 1: $\alpha_k \leftarrow \alpha_0$
 2: **while** $v(\alpha_k) < 1$ **do**
 3:     $\alpha_k \leftarrow \rho \cdot \alpha_k$
 4: **end while**

---

**Algorithm 2** Adaptive Backtracking Line Search

**Input:** $\alpha_0 > 0, v \colon \mathbb{R}_+ \to \mathbb{R}, \hat{\rho} \colon \mathbb{R} \to (0, 1)$
**Output:** $\alpha_k$
 1: $\alpha_k \leftarrow \alpha_0$
 2: **while** $v(\alpha_k) < 1$ **do**
 3:     $\alpha_k \leftarrow \hat{\rho}(v(\alpha_k)) \cdot \alpha_k$
 4: **end while**

---

## 2.3 RELATED WORK

Backtracking is a simple and effective alternative to exact line search subroutines (Nocedal & Wright, 2006, Ch. 3), which helps to explain why it remains a popular procedure to enforce various line search criteria (Beck & Teboulle, 2009; Nesterov, 2013; Vaswani et al., 2019b; Galli et al., 2023; Aujol et al., 2024). Notwithstanding, there is no standard practice to select the adjustment factor $\rho$, but rather some rough guidelines suggesting that the parameter $\rho$ "is often chosen to be between 0.1 (which corresponds to a very crude search) and 0.8 (which corresponds to a less crude search)" (Boyd & Vandenberghe, 2004, p. 466) and that it "is usually chosen from $1/2$ to $1/10$, depending on the confidence we have on the quality of the initial step-size" (Bertsekas, 2016, p. 36). Indeed, we find that $\rho$ varies somewhat arbitrarily depending on empirical performance and the conditions enforced by backtracking. For example, $\rho = 0.8$ is used in (Aujol et al., 2024) to enforce the descent lemma, while $\rho = 0.5$ is used in (Vaswani et al., 2019b) to enforce the Armijo conditions. With that in mind, in our experimental validation, we compare our proposed adaptive subroutine with regular backtracking for several values of $\rho$. Our goal is to exhibit compelling evidence for a simple alternative to a classic method that remains popular, rather than champion a particular line search subroutine. Accordingly, we do not compare against subroutines that do not enforce similar line search criteria, such as (Fridovich-Keil & Recht, 2020; Orseau & Hutter, 2023), nor subroutines that do not release code (de Oliveira & Takahashi, 2021). Likewise, we leave recent twists on backtracking, such as (Truong & Nguyen, 2021; Calatroni & Chambolle, 2019; Rebegoldi & Calatroni, 2022), for future work, since these methods could in principle also benefit from adaptive adjustments (see Section 5.)

## 2.4 CASE STUDY: ARMIJO CONDITION

The most popular criterion used in line search is the *Armijo condition* (Armijo, 1966), which is specified by a hyperparameter $c \in (0, 1)$ and requires sufficient decrease in the objective function:

$$F(x_k + \alpha_k d_k) - F(x_k) \leq c \cdot \alpha_k \langle \nabla F(x_k), d_k \rangle. \tag{3}$$

For the Armijo condition, the direction $d_k$ is usually assumed to be a descent direction:

**Assumption 1** (descent direction). *The direction $d_k$ satisfies $\langle \nabla F(x_k), d_k \rangle < 0$.*

We define the *violation* of (3) as

$$v(\alpha_k) := \frac{F(x_k + \alpha_k d_k) - F(x_k)}{c \cdot \alpha_k \langle \nabla F(x_k), d_k \rangle}. \tag{4a}$$

Under Assumption 1, (3) can be written as $v(\alpha_k) \geq 1$. To account for the information conveyed by (3) when violated, we choose the corresponding adaptive geometric factor $\hat{\rho}(v(\alpha_k))$ as

$$\hat{\rho}(v(\alpha_k)) := \max\left(\epsilon, \rho^{\frac{1-c}{1-c \cdot v(\alpha_k)}}\right), \tag{4b}$$

where $\epsilon > 0$ is a small factor that prevents occasional numerical errors in $v(\alpha_k)$ from spreading to $\hat{\rho}(v(\alpha_k))$. Although (4b) is parameterized by $\epsilon$ and $\rho$, for each method we fix their values on all experiments, effectively making Algorithm 2 parameter free. We use our adaptive BLS procedure to find suitable step-sizes for three standard base methods: gradient descent (GD), Nesterov's accelerated gradient descent (AGD) (Nesterov, 1983) and Adagrad (Duchi et al., 2011). The standard implementations that we use for these algorithms are given in Appendix C. Incorporating line search into GD and Adagrad is straightforward, but the case of AGD merits further comment.

**Backtracking and AGD.** Unlike GD, AGD is not necessarily a monotone method in the sense that $F(x_k + \alpha_k d_k) \leq F(x_k)$ need not hold. But AGD is a *multistep* method, one being a GD step, for which line search can help to compute a step-size or, equivalently, to estimate the Lipschitz constant $L$. If the estimate of $L$ satisfies (3) with $c = 1/2$ and is increasingly multiplied by a lower bounded positive geometric factor, then AGD with line search enjoys essentially the same theoretical guarantees of AGD tuned with constant parameters. We also consider AGD with memoryless line search with fixed predetermined initial step-sizes. Then, unless some variant such as Scheinberg et al. (2014) is used, the theoretical guarantees are not necessarily preserved when AGD is combined with memoryless line search. For some values of $\rho$, however, we find empirically that not only does the resulting method converge, but it does so much faster than the monotone line search variant, which in turn typically converges faster than AGD tuned with a pre-computed estimate $\bar{L}$ (see Appendix D.1.)

## 2.5 CASE STUDY: DESCENT LEMMA

A standard assumption in the analysis and design of several optimization algorithms is that gradients are Lipschitz-smooth, which implies (Nesterov, 2018, Thm. 2.1.5.) that there is some $L > 0$ such that

$$f(y) \leq f(x) + \langle \nabla f(x), y - x \rangle + \tfrac{L}{2} \|y - x\|^2, \qquad \forall x, y. \tag{5}$$

Inequality (5) is commonly known as the *descent lemma* (Bertsekas, 2016). In particular, it is commonly assumed (5) holds for algorithms that solve problems with composite objective functions $F := f + \psi$, where $f$ is Lipschitz-smooth convex and $\psi$ is continuous, possibly nonsmooth, convex. A prototypical example of such an algorithm is FISTA (Beck & Teboulle, 2009), which is an extension of Nesterov's AGD to composite problems (for details see Appendix C.) FISTA assumes that it produces points $y_k$ and $p_{\alpha_k}(y_k)$ satisfying (5) applied to $F$ with $x = y_k$ and $y = p_{\alpha_k}(y_k)$, where $p_\alpha$ denotes the proximal operator (Parikh & Boyd, 2014) parameterized by $\alpha > 0$ and defined by

$$p_\alpha(y) := \arg\min_x \left\{ \psi(x) + \frac{1}{2\alpha} \left\| x - \big(y - \alpha \nabla f(y)\big) \right\|^2 \right\}. \tag{6}$$

In practice, $\alpha = 1/L$ is seldom known for a given $f$, and FISTA estimates it with some $\alpha_k$ by checking

$$F(p_{\alpha_k}(y_k)) \leq f(y_k) + \langle \nabla f(y_k), p_{\alpha_k}(y_k) - y_k \rangle + \frac{1}{2\alpha_k} \|p_{\alpha_k}(y_k) - y_k\|^2 + \psi(p_{\alpha_k}(y_k)). \tag{7}$$

Since (7) holds for any $\alpha_k \leq 1/L$, an estimate $\alpha_k$ can be precomputed from analytical upper bounds on $L$ for particular cases of $f$, but these bounds tend to be overly conservative and can lead to poor performance. A better alternative, adopted by FISTA and many methods (Nesterov, 2013; Scheinberg et al., 2014), is to backtrack: reduce $\alpha_k$ by some constant factor $\rho < 1$ until (7) holds.

We define the *violation* of Eq. (7) as

$$v(\alpha_k) := \tfrac{1}{2\alpha_k} \|p_{\alpha_k}(y_k) - y_k\|^2 \Big/ \Big( f(p_{\alpha_k}(y_k)) - f(y_k) - \langle \nabla f(y_k), p_{\alpha_k}(y_k) - y_k \rangle \Big), \tag{8a}$$

and the corresponding adaptive factor as:

$$\boxed{\hat{\rho}(v(\alpha_k)) := \rho v(\alpha_k).} \tag{8b}$$

In experiments below, we use (8b) to find suitable step-sizes for FISTA, with a fixed $\rho < 1$ value.

## 3 EMPIRICAL PERFORMANCE

We present four experiments illustrating different ways and scenarios in which our adaptive backtracking line search (ABLS) subroutine (Algorithm 2) can outperform regular backtracking (Algorithm 1).

### 3.1 CONVEX OBJECTIVE: LOGISTIC REGRESSION + ARMIJO

First, we consider the logistic regression objective with $L_2$ regularization, defined by

$$F(x) = -\frac{1}{n} \sum_{i=1}^{n} \big( y_i \log(\sigma(a_i^\top x)) + (1 - y_i) \log(1 - \sigma(a_i^\top x)) \big) + \frac{\gamma}{2} \|x\|^2, \tag{9}$$

where $\sigma(z) = 1/(1 + \exp(-z))$ is the sigmoid function, $\gamma > 0$ and $(A_i, b_i) \in \mathbb{R}^d \times \{0, 1\}$ are $n$ observations from a given dataset. For each dataset, $\bar{L} = \lambda_{\max}(A^\top A)/(4n)$ provides an upper bound on the true Lipschitz parameter of the first term in (9), with which we fix $\gamma = \bar{L}/(10n)$ and the step-size of gradient descent to $1/(\bar{L} + \gamma)$. In all experiments, the initial point $x_0$ is the origin as is standard.

**Result Summary.** A succinct summary of our results contained in Table 1 and Appendix D is that

*across datasets and step-size initializations considered, adaptive backtracking is more robust than regular backtracking and often leads to significant improvements with respect to base methods.*

| | | Backtracking Line Search (BLS) | | | | | | Adaptive BLS (ABLS) | | | |
|---|---|---|---|---|---|---|---|---|---|---|---|
| | | $\rho = 0.2$ | | | $\rho = 0.3$ | | | $\rho = 0.3$ | | | |
| Method | Dataset | #f | #∇f | ET [s] | #f | #∇f | ET [s] | #f | #∇f | ET [s] | **gain** |
| GD | ADULT | 148597.2 | 32582.0 | 1319.0 | 74258.5 | 18749.0 | **694.7** | 37296.0 | 13583.8 | **370.0** | **46.7%** |
| | G_SCALE | 58488.5 | 18252.2 | **3050.4** | 65429.2 | 17111.5 | 3059.7 | 25917.2 | 11700.5 | **1607.0** | **47.3%** |
| | MNIST | 148469.8 | 41726.2 | **10986.4** | 207786.8 | 46475.5 | 13474.8 | 52616.5 | 22385.8 | **4841.9** | **55.9%** |
| | MUSHROOMS | 14170.5 | 6507.2 | **31.7** | 14800.8 | 6628.5 | 33.9 | 7611.0 | 3693.8 | **17.3** | **45.5%** |
| | PHISHING | 33388.2 | 8059.8 | 63.0 | 34193.5 | 7938.5 | **62.4** | 16543.2 | 6434.5 | **26.4** | **57.6%** |
| | PROTEIN | 28011.0 | 13260.5 | **4481.6** | 35733.2 | 14868.8 | 5282.0 | 27656.0 | 13207.2 | **4137.4** | **7.7%** |
| | WEB-1 | 6721.5 | 3139.0 | 9.0 | 6156.8 | 2972.8 | **8.4** | 6192.2 | 3024.8 | **7.9** | **(5.4%)\*** |

| | | $\rho = 0.5$ | | | $\rho = 0.6$ | | | $\rho = 0.9$ | | | |
|---|---|---|---|---|---|---|---|---|---|---|---|
| Method | Dataset | #f | #∇f | ET [s] | #f | #∇f | ET [s] | #f | #∇f | ET [s] | **gain** |
| AGD | ADULT | 17288.8 | 2014.2 | 116.6 | 49331.2 | 3806.2 | 247.0 | 7999.0 | 2275.0 | **49.2** | **52.5%** |
| | G_SCALE | 3580.2 | 592.5 | **114.9** | 18530.8 | 1964.0 | 512.9 | 2204.5 | 728.5 | **84.1** | **26.8%** |
| | MNIST | 8934.2 | 1524.5 | **452.3** | 14365.8 | 1846.5 | 643.7 | 4943.2 | 1666.2 | **283.0** | **37.4%** |
| | MUSHROOMS | 1100.5 | 372.2 | 1.7 | 1146.8 | 367.8 | **1.6** | 850.0 | 376.2 | **1.4** | **15.5%** |
| | PHISHING | 6763.2 | 944.2 | 9.1 | 8344.2 | 944.8 | **8.6** | 3699.0 | 1058.0 | **4.0** | **53.6%** |
| | PROTEIN | 2865.0 | 1232.2 | 396.3 | 3397.5 | 1272.5 | **346.5** | 2743.2 | 1257.0 | **291.6** | **15.8%** |
| | WEB-1 | 651.8 | 208.5 | 0.6 | 699.8 | 202.2 | **0.5** | 519.2 | 217.8 | **0.4** | **3.3%** |

| | | $\rho = 0.2$ | | | $\rho = 0.3$ | | | $\rho = 0.3$ | | | |
|---|---|---|---|---|---|---|---|---|---|---|---|
| Method | Dataset | #f | #∇f | ET [s] | #f | #∇f | ET [s] | #f | #∇f | ET [s] | **gain** |
| Adagrad | ADULT | 124102.0 | 20001.0 | 699.5 | 145159.8 | 19789.8 | 746.5 | 27179.0 | 8000.5 | **178.8** | **74.4%** |
| | G_SCALE | 274023.2 | 33071.5 | **7396.5** | 361852.0 | 34933.8 | 9740.9 | 84201.0 | 17176.2 | **2425.4** | **67.2%** |
| | MNIST | 86240.5 | 13843.8 | **3921.2** | 99021.8 | 14760.8 | 4377.2 | 12366.2 | 3521.8 | **679.5** | **82.7%** |
| | MUSHROOMS | 7794.8 | 1967.8 | **9.0** | 7239.5 | 1693.0 | 9.3 | 3751.5 | 1446.0 | **6.0** | **33.2%** |
| | PHISHING | 74737.5 | 15833.5 | **68.9** | 117564.0 | 20001.0 | 96.4 | 18053.0 | 6375.0 | **19.4** | **71.8%** |
| | PROTEIN | 6103.0 | 809.2 | 420.6 | 4040.2 | 429.0 | **257.9** | 1845.8 | 446.5 | **164.4** | **36.2%** |
| | WEB-1 | 4384.2 | 1027.0 | 2.9 | 3857.8 | 745.2 | **2.5** | 1726.5 | 568.8 | **1.3** | **50.1%** |

Table 1: Logistic regression. $\#f$ and $\#\nabla f$ denote the number of function and gradient evaluations and ET refers to elapsed time in seconds. The **gain** is given by $1 - (\text{ET of ABLS})/(\text{ET of BLS})$ with the best ET for BLS across $\rho$ in each experiment, which is bolded. We ran each BLS experiment with a grid of four $\rho$'s and present the best two in the table. The gain for GD on WEB-1 is colored orange because although ABLS terminated before the best performing BLS variant, it required more function and gradient evaluations. This anomaly can be attributed to the relatively small ET for this problem.

**Datasets and methods.** We take observations from seven datasets, whose details can be found in Appendix D. We consider GD, AGD, and Adagrad, described in in Appendix C, with BLS for $\rho \in \{0.2, 0.3, 0.5, 0.6\}$ and our ABLS with a pre-set $\rho$.

**Initialization.** We set the starting point $x_0$ as the origin and fix $\epsilon = 0.01$ in (4b) on all experiments. We also fix $\rho$, but change it according to the base method. For more details, see Appendix D.

**Evaluation.** We run all methods for long enough to produce solutions with designated precision, then average various metrics over different initial step-sizes. For more details, see Appendix D. All experiments were run on a supercomputer cluster containing Intel Xeon Platinum 8260 and Intel Xeon Gold 6248 CPUs Reuther et al. (2018).

**Remarks.** Table 1 shows that ABLS significantly outperform BLS. For GD and Adagrad, ABLS variants outperforms BLS for almost every combination of $\rho$ and $\alpha_0$ by saving function evaluations and returning better step-sizes, which speed up convergence. Fig. 1 illustrates this point by showing how the suboptimality gap evolves with time for the baseline GD, its ABLS variant and BLS variants for two choices of initial step-size. In particular, increasing the initial step-size helps BLS in some datasets but not in others. Fig. 2 shows a similar trend for the case of AGD. In general, ABLS is more robust to the choice of initial step-size and that is the main reason why the ABLS variant of AGD outperforms its BLS counterparts. In Appendix D.2, Figs. 14 and 15 show the corresponding step-sizes. Initially, ABLS returns smaller GD step-sizes than BLS, but this trend quickly reverses. A plausible explanation is that BLS returns the largest step-sizes that satisfy (3), within a factor of $\rho$. If the step-sizes are excessively large initially, they can lead to worse optimization paths (e.g., more zig-zagging). For AGD, the step-sizes follow a similar trend initially, but then the adaptive step-size seems to converge while the regular step-sizes not always do. This can be indicative of another shortcoming of regular backtracking, namely that it can only return step-sizes that are powers of $\rho$ times the initial step-size, in contrast with adaptive backtracking.

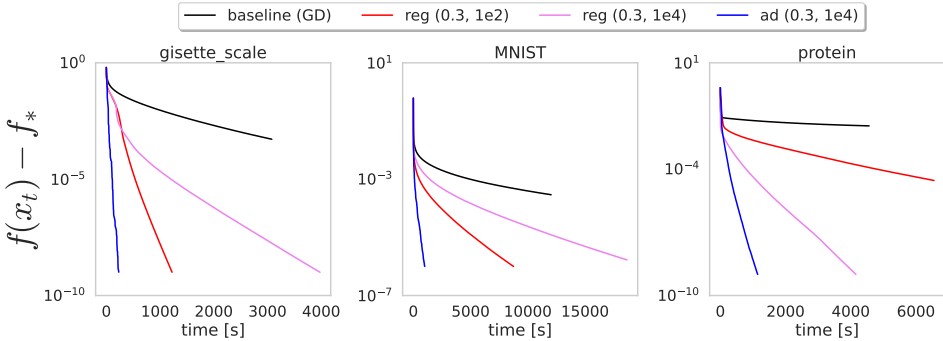

Figure 1: Baseline: GD with constant $\alpha_k = 1/\bar{L}$; reg $(\rho, \beta)$ and ad $(\rho, \beta)$: GD with, respectively, regular and adaptive memoryless BLS parameterized by $\rho$ and $\alpha_0 = \beta/\bar{L}$.

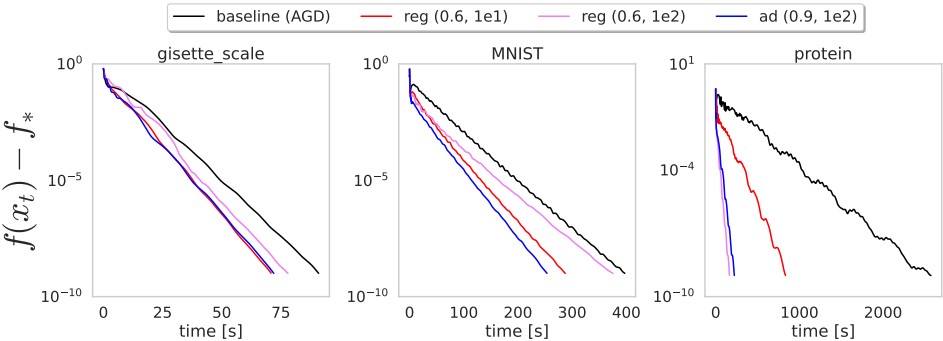

Figure 2: Baseline: AGD with constant $\alpha_k = 1/\bar{L}$; reg $(\rho, \beta)$ and ad $(\rho, \beta)$: AGD with, respectively, regular and adaptive memoryless BLS parameterized by $\rho$ and $\alpha_0 = \beta/\bar{L}$.

### 3.2 CONVEX OBJECTIVE: LINEAR INVERSE PROBLEMS + DESCENT LEMMA

The goal of a linear inverse problem is to recover the sparse signal $x$ from a noisy measurement model $y = Ax + \epsilon$, where $A \in \mathbb{R}^{n \times d}$ and $y \in \mathbb{R}^n$ are known, and $\epsilon$ is unknown noise. The problem of estimating $x$ is typically posed as a Lasso objective (Santosa & Symes, 1986; Tibshirani, 1996)

$$F(x) = \frac{1}{2}\|Ax - y\|^2 + \lambda\|x\|_1.$$

**Datasets.** We take observations $A$ from eight datasets, see Appendix E for details.

**Methods.** We consider FISTA (Beck & Teboulle, 2009) (Algorithm 6) and BLS variants. For BLS, $\rho = \{1/2, 1/3, 1/5\}$ and $\rho = 1/1.1 \approx .9$ for ABLS, mirroring the choice for AGD above. All BLS methods start with the same initial Lipschitz constant estimate increase it accordingly.

**Initialization.** For each dataset, we empirically find values of $\alpha_0 = 1/L_0$ around which backtracking becomes active, and then increase them successively (values reported on Appendix E.)

**Results Summary.** The ABLS variant of FISTA outperforms its BLS counterparts across all datasets tested. Moreover, the best value of $\rho$ for BLS changes from one dataset to the other. Since the Lipschitz constant estimate is monotone, function evaluations vary little across methods and have small impact on performance. Nevertheless, ABLS requires fewer function evaluations for all datasets.

### 3.3 NONCONVEX OBJECTIVE: ROSENBROCK + ARMIJO

We consider the classical nonconvex problem given by the Rosenbrock objective function $F(u, v) = 100(u - v^2)^2 + (1 - v)^2$. We use the origin as the initial point and $0.1$ as the initial step-size. Fig. 3 shows the optimization paths for BLS and ABLS memoryless variants of GD and AGD after 1000

| | | Backtracking Line Search (BLS) | | | | Adaptive BLS (ABLS) | | |
| | | $\rho = 0.5$ | | $\rho = 0.3$ | | $\rho = 0.9$ | | |
| Method | Dataset | $\Delta\#f$ | $\#\nabla f$ | $\Delta\#f$ | $\#\nabla f$ | $\Delta\#f$ | $\#\nabla f$ | **gain** |
|---|---|---|---|---|---|---|---|---|
| FISTA | DIGITS | 15.5 | **28282.25** | 10 | 34730.75 | 4.75 | 16756 | **40.8%** |
| | IRIS | 10.75 | **726.25** | 7 | 816.5 | 4 | 710 | **2.2%** |
| | OLIVETTI | 10 | **242709.5** | 6.25 | 246827.75 | 2 | 212930.75 | **12.3%** |
| | LFW* | 26.25 | 49093.75 | 17 | **49014.5** | 2 | 45070.75 | **8.0%** |
| | SPEAR3 | 13.25 | **328328.75** | 9 | 506308.5 | 2 | 255417.5 | **22.2%** |
| | SPEAR10 | 44.75 | **18691** | 29.75 | 19992.75 | 8 | 15128 | **19.1%** |
| | SPARCO3 | 27.75 | **266.25** | 18.5 | 276.75 | 3.25 | 251 | **5.7%** |
| | WINE | 48 | **529333.25** | 27.75 | 564293.75 | 8.5 | 472527 | **10.7%** |

Table 2: Linear inverse problem. $\#\nabla f$ and $\Delta\#f$ denote the number of gradient and excess function evaluations (total function evaluations minus two times total iterations). The **gain** is given by $1 - (\#\nabla f$ of ABLS$)/(\#\nabla f$ of BLS$)$ with the best ET for BLS across $\rho$ in each dataset (bolded.) We run each BLS experiment with three $\rho$'s and present the best two in the table. ABLS reached the desired precision in all testpoints while the asterisk on LFW* indicates BLS did not in at least one testpoint.

iterations, using $\rho = 0.3$ and $\rho = 0.9$, respectively. We see that the ABLS variants achieve better losses, requiring far fewer function evaluations and less time to do so.

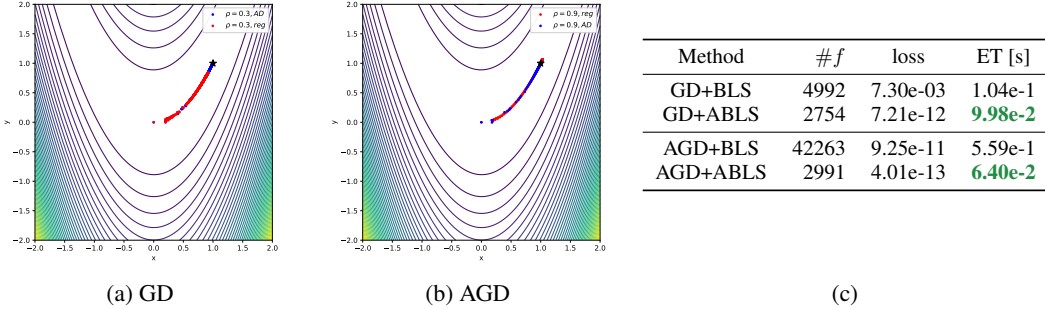

| | Method | $\#f$ | loss | ET [s] |
|---|---|---|---|---|
| | GD+BLS | 4992 | 7.30e-03 | 1.04e-1 |
| | GD+ABLS | 2754 | 7.21e-12 | **9.98e-2** |
| | AGD+BLS | 42263 | 9.25e-11 | 5.59e-1 |
| | AGD+ABLS | 2991 | 4.01e-13 | **6.40e-2** |

(a) GD  (b) AGD  (c)

Figure 3: Performance of GD and AGD regular (red) and adaptive (blue) BLS variants on Rosenbrock. "loss" refers to the final loss after 1000 iterations.

## 3.4 NONCONVEX OBJECTIVE: MATRIX FACTORIZATION + ARMIJO

Lastly, we consider the nonconvex problem of matrix factorization, defined by the objective $F(U,V) = \frac{1}{2}\|UV^\top - A\|_F^2$, where $A \in \mathbb{R}^{m \times n}$, $U \in \mathbb{R}^{m \times r}$, $V \in \mathbb{R}^{n \times r}$ and $r < \min\{m,n\}$. We take $A$ from the MovieLens 100K dataset (Harper & Konstan, 2015) and consider three rank values $r \in \{10, 20, 30\}$ (see Appendix F for further details and full plots.)

For this experiment we replicate the initialization and evaluation methodologies of Section 3.1, except we disconsider Adagrad, and pick different values for initial step-sizes, $\{0.05, 0.5, 5, 50\}$, and $\rho$. Namely, we let $\rho \in \{0.2, 0.3, 0.5, 0.6\}$ for the BLS variants, but fix $\rho = 0.3$ and $\rho = 0.9$ for the ABLS GD and AGD variants, respectively. Table 3 summarizes the results for ABLS and the top two BLS variants. Once again, the best value of $\rho$ for BLS is inconsistent: $\rho = 0.3$ for ranks 10 and 20 but $\rho = 0.2$ for rank 30. ABLS requires significantly fewer gradient and function evaluations than the top BLS variant does, which leads to considerable gains in time to achieve the desired precision.

## 4 MOTIVATION AND THEORETICAL RESULTS

In this section, we motivate our choices of adaptive factors and characterize them theoretically.

The particular choices of adaptive factors were made with two goals in mind: *generate more aggressive backtracking factors to save function evaluations* and *guarantee reasonably large step-sizes to achieve fast convergence*. To meet our first goal, $\hat{\rho}(v(\alpha_k)) \in (0, \rho)$ must hold. Indeed, if (3) is violated, then $v(\alpha_k) < 1$ because $d_k$ is assumed a descent direction, where $v$ is defined by (4a). So, if in addition

| Method | Rank | Backtracking Line Search | | | | | | Adaptive BLS (ABLS) | | | |
| | | $\rho = 0.2$ | | | $\rho = 0.3$ | | | $\rho = 0.3$ | | | |
| | | #$f$ | #$\nabla f$ | ET [s] | #$f$ | #$\nabla f$ | ET [s] | #$f$ | #$\nabla f$ | ET [s] | gain |
| GD | 10 | 39960.0 | 5683.5 | 1034.5 | 33531.8 | 4897.8 | **999.5** | 52075.8 | 1631.8 | **64.8** | **93.5%** |
| | 20 | 127480.2 | 18734.5 | 1153.2 | 111322.0 | 16109.2 | **1010.9** | 377111.8 | 5133.0 | **170.9** | **83.1%** |
| | 30 | 231170.5 | 35639.8 | **2035.5** | 394472.2 | 50001.0 | 4195.1 | 681952.0 | 14802.8 | **669.2** | **67.1%** |
| | | $\rho = 0.3$ | | | $\rho = 0.5$ | | | $\rho = 0.9$ | | | |
| AGD | 10 | 362029.7 | 21909.3 | 2377.0 | 44873.0 | 6198.3 | **379.4** | 22707.7 | 6663.7 | **172.0** | **54.7%** |
| | 20 | 479432.0 | 35672.0 | **2987.9** | 491753.3 | 33523.3 | 3588.6 | 80720.3 | 24588.7 | **738.0** | **75.3%** |
| | 30 | 357885.7 | 39234.3 | **2187.8** | 478084.3 | 42985.3 | 5157.0 | 95040.0 | 27454.0 | **855.1** | **60.9%** |

Table 3: Backtracking for matrix factorization. #$f$ and #$\nabla f$ denote the number of function and gradient evaluations and ET refers to elapsed time in seconds. The **gain** is given by $1 - (\text{ET of ABLS})/(\text{ET of BLS})$ with the best ET for BLS across $\rho$ in each experiment, which is bolded.

$\epsilon \in (0, \rho)$ in (4b), then $\hat{\rho}(v(\alpha_k)) < \rho$. To meet the second goal, the returned step-size must be non-trivially lower bounded. To this end, in Section 4.2 we show that if the objective function is Lipschitz-smooth, then ABLS returns a step-size on par with the greatest step-size that is guaranteed to satisfy (3). For now, we note that if (3) is violated, then $1 - c \cdot v(\alpha_k) > 0$, since $c \in (0, 1)$ by assumption. Thus, $\hat{\rho}(v(\alpha_k))$ is bounded away from zero. Moreover, that same bound applies to BLS. Similar conclusions can be reached if the BLS criterion is (7) and $\hat{\rho}$ and $v$ are chosen as (8b) and (8a).

## 4.1 THE SCOPE OF THEORETICAL GUARANTEES FOR A BACKTRACKING SUBROUTINE

Before characterizing adaptive backtracking theoretically, we state a theoretical bound on how well a general line search procedure can do relative to regular backtracking. Then, we state three further facts that substantiate why comparing line search procures is hard. These statements are consequences of simple examples described in Appendix B.1.

The fundamental obstacle we face in theoretically comparing adaptive backtracking, or any other line search procedure, with regular backtracking is described by Example 1, which shows that

> *no line search procedure can be provably better than regular backtracking to enforce any set of line search criteria that includes the Armijo condition or the descent lemma, for any class of functions that includes quadratics.*

We compare line search procedures on the basis of the number of adjustments to return a feasible step-size and the length of the returned step-size. To establish this fact, in Appendix B.1 we construct a simple scalar quadratics example in which regular backtracking returns the greatest feasible step-size after one adjustment and furthermore this step-size leads to the minimum. Also in Appendix B.1, we present a similar example for the descent lemma.

In general, comparing line search procedures is challenging because different procedures return different step-sizes, which lead to different optimization paths. For example, a procedure that reduces step-sizes more in each adjustment need not require fewer adjustments (function evaluations) overall than a procedure that reduces step sizes less because each procedure leads to a different sequence of iterates. In fact, comparison is challenging *even for a single backtracking call* because:

1. the step-size returned by backtracking is not monotone in $\rho$, *even for convex problems*.

2. For nonconvex problems, given step-sizes $\alpha'$ and $\alpha$ with $\alpha' < \alpha$, $\alpha$ being feasible does not imply $\alpha'$ is also feasible.

3. For nonconvex problems, decreasing $\rho$ may increase the number of criteria evaluations required to compute a feasible step-size.

Example 2 establishes the first fact and Example 3 establishes the second and third facts.

With the above in mind, in the following we provide the best set of performance guarantees that one can hope for. Namely, we show that adaptive backtracking requires no more function evaluations

than regular backtracking to return a feasible step-size, and that the step-size lower bounds for the two procedures are the same, which leads to the same global convergence rate results.

## 4.2 Theoretical results

In this subsection, we present theoretical results regarding regular and adaptive backtracking. Several additional convergence results and full proofs can be found in Appendix B.

**Convex problems.** We show that only the first fact above holds for convex problems, which expands the extent to which two backtracking subroutines can be compared.

**Proposition 1.** *Let $F$ be convex differentiable. Given a point $x_k$, a direction $d_k$ and a step-size $\alpha_k > 0$ satisfying (3) for some $c$, then $x_k$, $d_k$ and $\alpha_k'$ also satisfy (3) for any $\alpha_k' \in (0, \alpha_k)$.*

The following proposition refers to "compatible inputs." By this we mean:

**Definition 1** (Compatibility). *The inputs to Algorithms 1 and 2 are said to be* compatible *if $\alpha_0, c, v$ coincide and the input $\hat{\rho}$ to Algorithm 2 is parameterized by the same $\rho$ that Algorithm 1 takes as input.*

**Proposition 2.** *Let $F$ be convex differentiable. Fixing all other inputs, the number of function evaluations that Algorithm 1 and Algorithm 2 take to return a feasible step-size is nondecreasing in the input $\rho$. Moreover, given compatible inputs with a descent direction and $\epsilon < \rho$, Algorithm 2 takes no more function evaluations to return a feasible step-size than Algorithm 1 does.*

**Nonconvex problems.** For convex problems, we were able to compare the number of times regular and adaptive backtracking must evaluate their criteria in order to return a single feasible step-size. But what really matters is the total number of criteria evaluations up to a given iteration. We bound this number for general nonconvex problems, hinging on the following properties.[1]

**Definition 2** (Smoothness). *A function $F$ is said to be* Lipschitz-smooth *if (5) holds for some $L > 0$.*

**Definition 3** (Gradient related). *The directions $d_k$ are said to be* gradient related *if there are $c_1 > 0$ and $c_2 > 0$ such that $\langle \nabla F(x_k), -d_k \rangle \geq c_1 \|\nabla F(x_k)\|^2$ and $\|d_k\| \leq c_2 \|\nabla F(x_k)\|$, for all $k \geq 0$.*

**Assumption 2.** *We assume $F$ is Lipschitz-smooth and $d_k$ are gradient related.*

Gradient relatedness ensures that $d_k$ is not "too large" or "too small" with respect to $\nabla F(x_k)$ and that the angle between $d_k$ and $\nabla F(x_k)$ is not "too close" to being perpendicular (Bertsekas, 2016, p. 41). Together with $c_1$ and $c_2$, the Lipschitz constant $L$ and Armijo constant $c$ define a step-size threshold $\bar{\alpha} = 2c_1(1 - c)/Lc_2^2$ below which (3) holds. This quantity is central in the following result.

**Informal Theorem** (Armijo). *Let $F$ be Lipschitz-smooth and $d_k$ gradient related. Given compatible inputs, if $\epsilon < \rho$ and $v, \hat{\rho}$ are chosen as (4), then Algorithms 1 and 2 share the same bounds on the total number of backtracking criteria evaluations up to any iteration. If $\alpha_k$ is received as the initial step-size input at iteration $k + 1$ for all $k \geq 0$, then they evaluate (3) at most $\lfloor \log_\rho(\bar{\alpha}/\alpha_0) \rfloor + 1 + k$ times up to iteration $k$. If, on the other hand, $\alpha_0$ is received as the initial step-size input at every iteration, then they evaluate (3) at most $k(\lfloor \log_\rho(\bar{\alpha}/\alpha_0) \rfloor + 1)$ times up to iteration $k$. Moreover, Algorithms 1 and 2 always return a step-size $\alpha_k$ such that $\alpha_k \geq \min(\alpha_0, \rho\bar{\alpha})$.*

**Informal Theorem** (Descent lemma). *Let $f$ be Lipschitz-smooth convex and let $\psi$ be continuous convex. Also, suppose $v$ and $\hat{\rho}$ are chosen as (8a) and (8b). If $\alpha_k \in (0, 1/L)$, then (7) holds for all $y_k$. If Algorithms 1 and 2 receive $\alpha_k$ as the initial step-size input in iteration $k+1$, then they evaluate (7) at most $\lfloor \log_\rho(1/L\alpha_0) \rfloor + 1 + k$ times up to iteration $k$. If, on the other hand, Algorithms 1 and 2 receive $\alpha_0$ as the initial step-size input in every iteration, then they evaluate (7) at most $k(\lfloor \log_\rho(1/L\alpha_0) \rfloor + 1)$ times up to iteration $k$. Moreover, they return a feasible step-size $\alpha_k$ such that $\alpha_k \geq \min\{\alpha_0, \rho/L\}$.*

## 5 Future work: further applications, extensions and limitations

Adaptive backtracking is a general idea that can be broadly applied in a variety of settings. Our goal in this paper was to rigorously validate it in classical machine learning and optimization problems. This section outlines several promising directions for future work and speculate about potential limitations.

---

[1] Instead of the usual condition that $\|\nabla F(x) - \nabla F(y)\| \leq L\|x - y\|$ hold for all $x, y$, we adopt the equivalent (Nesterov, 2018, Thm. 2.1.5.) condition (5) as the definition of Lipschitz-smoothness for the sake of convenience.

## 5.1 Further applications and extensions

**Stochastic line search.** In machine learning, models such as over-parameterized neural networks are sufficiently expressive to *interpolate* immense datasets (Zhang et al., 2016; Ma et al., 2018). Interpolation provides theoretical foundation for the stochastic line search (SLS) proposed by Vaswani et al. (2019b), which enforces the Armijo condition on the training mini-batches. In the same vein, Galli et al. (2023) replaced the Armijo condition in SLS with a nonmonotone criterion and used Polyak's step-size to devise an initial step-size heuristic, obtaining the Polyak Nonmonotone Stochastic (PoNoS) method. Below, we reproduce experiment 1 from (Galli et al., 2023) to demonstrate the potential of applying ABLS in combination with stochastic line search in the interpolating regime. Fig. 4 shows that combining ABLS with PoNoS leads to good test accuracy in fewer epochs (details in Appendix A.) We defer fully developing this application to future work.

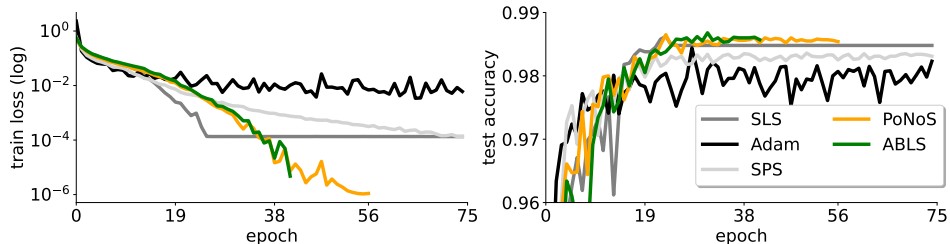

Figure 4: MLP trained on MNIST with different algorithms.

**Additional line search criteria.** Adaptive backtracking may be useful to enforce other conditions that are affine in the step-size. A prominent candidate are the Goldstein conditions (Goldstein & Price, 1967), which comprise two inequalities that are affine in the step-size. Nonmonotone line search criteria (Grippo et al., 1986; Zhang & Hager, 2004) also offer several candidates.

**Additional algorithms and increasing step-sizes.** In this paper, we experimented with increasing step-sizes through memoryless line search, where the initial step-size is fixed for every iteration, but other schemes are possible. For example, adaptive adjustments can also be used to increase step-sizes. In addition, adaptive backtracking can replace regular backtracking in line search methods that increase the current step-size and then use it as the initial step-size, such as the two-way method (Truong & Nguyen, 2021) and FISTA variants (Scheinberg et al., 2014; Calatroni & Chambolle, 2019; Rebegoldi & Calatroni, 2022). Also, it would be interesting to see how adaptive backtracking works together with schemes that handle problems where the strong convexity constant is unknown, namely restarting schemes Becker et al. (2011); O'Donoghue & Candès (2015); Aujol et al. (2024). Finally, we note that the violation of a condition can also be used indirectly to adjust step-sizes, for example to pick the degree to which a fixed $\rho$ is exponentiated, saving backtracking cycles.

## 5.2 Limitations

The weak and strong Wolfe conditions (Wolfe, 1969) are not affine in the step-size and are not satisfied by arbitrarily small step-sizes. Hence, Wolfe conditions are not enforced by backtracking (e.g., Nocedal & Wright (2006, pp. 60–61)) and it is unclear how to find analogous adaptive schemes. In turn, quasi-Newton methods, which often must enforce Wolfe conditions to guarantee global convergence, may not be suitable candidates for adaptive line search subroutines. In reality, the role of line search for these methods is to guarantee they converge globally rather than finding the "right" step-size, since they work with unit step-size locally. Hence, only few adjustments may be necessary. The same applies to Newton's method and the Barzilai–Borwein method (Barzilai & Borwein, 1988).

It is also unclear if adaptive adjustments can be useful for more general stochastic line search methods that do not rely on the interpolation property (Cartis & Scheinberg, 2017; Paquette & Scheinberg, 2020). Instead, they resample function and gradient mini-batches in every loop, whether a sufficient descent condition is violated or not. But the information conveyed by the violation for one sample need not be relevant to satisfy the same condition with a different sample, which poses a potential limitation.

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

## A  STOCHASTIC LINE SEARCH EXAMPLE

The results presented in Fig. 4 correspond to experiment 1 from (Galli et al., 2023) without any modifications, and are run using the base code from the same paper, which can be found at:

https://github.com/leonardogalli91/PoNoS.

In this experiment, a multilayer perceptron (MLP) with a single layer, width 1000 and 535818 parameters is trained on the MNIST dataset (LeCun et al., 1998) until the training loss becomes less than $10^{-6}$. This is the same stopping criterion adopted in (Galli et al., 2023). To train the MLP, the following methods are used:

1. Adam: adaptive moment estimation method (Kingma & Ba, 2015).
2. SLS: stochastic line search (Vaswani et al., 2019a).
3. SPS: stochastic Polyak step-size method (Loizou et al., 2021).
4. PoNoS: Polyak nonmonotone stochastic method (Galli et al., 2023).
5. ABLS: an adaptive backtracking variant of PoNoS, detailed below.

As Fig. 4 shows, only PoNoS and ABLS terminate within 75 epochs. PoNoS does so after 56 epochs and 583 seconds, while ABLS finishes in 41 epochs and 464 seconds.

For all the above methods, we preserve the parameters recommended in (Galli et al., 2023) unaltered from the source code. The ABLS method combines our adaptive backtracking procedure with a simplified version of PoNoS. Namely, PoNoS generates initial step-sizes with

$$\eta_k = \eta_{k,0}\delta^{\bar{l}_k + l_k}, \tag{10}$$

where $\delta \in (0,1)$ and $l_k$ is the amount of backtracks in iteration $k-1$, which are accounted for in

$$\bar{l}_k = \max\{\bar{l}_{k-1} + l_{k-1} - 1, 0\}.$$

That is, previous backtracks are used to discount an initial step-size given by

$$\eta_{k,0} = \min\{\tilde{\eta}_{k,0}, \eta^{\max}\},$$

which in turn is based on the Polyak initial step-size

$$\tilde{\eta}_{k,0} = \frac{f_{i_k}(w_k) - f_{i_k}^*}{c_p\|\nabla f_{i_k}(w_k)\|^2}, \tag{11}$$

where $c_p \in (0,1)$ is a hyperparameter, $i_k$ denotes the mini-batch sampled in iteration $k$, $w_k$ denotes the MLP parameters in iteration $k$ and $f_{i_k}^*$ refers to the minimum of the mini-batch training loss:

$$f_{i_k} = \frac{1}{|i_k|}\sum_{j \in i_k} f_j.$$

Instead, we simply use (11) as the initial step-size, with $c_p = 1/2$, the same value proposed in (Galli et al., 2023). Then, we apply ABLS to enforce

$$f_{i_k}(w_k - \eta_k\nabla f_{i_k}(w_k)) \leq C_k - c\eta_k\|\nabla f_{i_k}\|^2, \qquad c \in (0,1), \tag{12}$$

where $C_k$ denotes an exponential moving average of losses given by

$$C_k = \max\{\tilde{C}_k, f_{i_k}(w_k)\}, \qquad \tilde{C}_k = \frac{\xi Q_k C_{k-1} + f_{i_k}(w_k)}{Q_{k+1}}, \qquad Q_{k+1} = \xi Q_k + 1$$

with $\xi \in (0,1)$. The inequality (12) is a stochastic variant enforced by PoNoS of the deterministic criterion proposed by Zhang & Hager (2004). This inequality can be seen as a generalization of the Armijo condition with the current loss replaced with an average. As such, it preserves the structure of (3), which allows us to seamlessly apply (4b), with $\rho = 0.9$ and $\epsilon = 0.5$.

## B    EXAMPLES, THEOREMS AND PROOFS

In this section, we present the examples mentioned in Section 4, prove the results stated therein and provide further convergence results.

### B.1    EXAMPLES

The first example establishes the fundamental obstacle that

> **no line search procedure can be provably better than regular backtracking to enforce any set of line search criteria that includes the Armijo condition or the descent lemma, for any class of functions that includes quadratics.**

The second and third examples establish three more facts:

1. The step-size returned by backtracking is not monotone in $\rho$, *even for convex problems*.

2. For nonconvex problems, given step-sizes $\alpha'$ and $\alpha$ with $\alpha' < \alpha$, $\alpha$ being feasible does not imply $\alpha'$ is too.

3. For nonconvex problems, decreasing $\rho$ may increase the number of criteria evaluations required to compute a feasible step-size.

**Example 1** (Fundamental obstacle). *Let $F$ be defined by $F(x) = x^2/2$. If $x_k \neq 0$ and $d_k$ is a descent direction, then by definition $\langle \nabla F(x_k), d_k \rangle = d_k x_k < 0$ and it must be that $d_k = -c_1 x_k$ for some $c_1 > 0$. Hence, given some $c \in (0,1)$, (3) holds if and only if*

$$\frac{1}{2}(1 - \alpha_k c_1)^2 x_k^2 = F(x_{k+1}) \leq F(x_k) + c\alpha_k \langle \nabla F(x_k), d_k \rangle = \frac{1}{2} x_k^2 - c\alpha_k c_1 x_k^2.$$

*Therefore, (3) holds if and only if*

$$c_1(2(1-c) - c_1 \alpha_k)\alpha_k x_k^2 = (1 - 2cc_1\alpha_k - (1 - c_1\alpha_k)^2)x_k^2 \geq 0,$$

*or, equivalently, $\alpha_k \leq 2(1-c)/c_1$, since $x_k^2 > 0$. Now, suppose that $\alpha_0 = 4(1-c)/c_1 > 0$ and $\rho = 1/2$. Then, after testing (3) exactly once, backtracking returns the step-size $\alpha_k = 2(1-c)/c_1$, the greatest value that is guaranteed to satisfy (3). Thus, in this example, backtracking is optimal in the sense that it tests (3) only once to return the greatest feasible step-size possible. Therefore, no other line search procedure can be provably better than backtracking to enforce any set of line search criteria that includes the Armijo condition for any class of functions that includes quadratics.*

*For the sake of concreteness, Fig. 5 illustrates the particular case where $c = 1/2$, $x_0 = -1$, $d_0 = -\nabla f(x_0) = 1$, $\alpha_0 = 2$ and $\rho = 1/2$. The initial step-size $\alpha_0 = 2$, is too large and leads to a tentative iterate $x_0 + \alpha_0 d_0 = 1$ that lies outside of the shaded region of iterates allowable by the Armijo condition. After one adjustment, however, the step-size is reduced to $\rho\alpha_0 = 1/2$, leading to the tentative iterate $x_0 + \rho\alpha_0 d_0 = 0$, which lies at the boundary of the region of allowable iterates by the Armijo condition. That is, regular backtracking returns the greatest feasible step-size after one adjustment. Furthermore, this step-size leads to the minimum, therefore solving the problem after one iteration and one adjustment.*

*Next, we show an analogous result for the descent lemma. Keeping $F(x) = x^2/2$, for any $x \neq y$, (5) holds with an estimate $L_k$ of the Lipschitz constant if and only if*

$$\frac{x^2}{2} = F(y) \leq F(y) + \langle \nabla f(y), x - y \rangle + \frac{L_k}{2}\|x - y\|^2 = \frac{L_k - 1}{2}(y^2 - 2xy) + \frac{L_k}{2} x^2,$$

*which is equivalent to $L_k \geq 1$. Hence, if $L_0 \in (0,1)$ and $\rho = L_0$, then backtracking returns the optimal estimate $L_0/\rho = 1$ after testing (5) exactly once (requiring two function evaluations), no matter what $x \neq y$ are. Hence, in this example, backtracking is optimal in the sense that it tests (5) only once to return the tightest Lipschitz constant estimate possible. Therefore, no other line search procedure can be provably better than backtracking to enforce any set of line search criteria that includes the descent lemma for any class of functions that includes quadratics.*

*As a side note, we look into what adaptive backtracking would do. For this problem, (8a) becomes*

$$v(\alpha_k) = \frac{\frac{1}{2\alpha_k}\|p_{\alpha_k}(y_k) - y_k\|^2}{f(p_{\alpha_k}(y_k)) - f(y_k) - \langle \nabla f(y_k), p_{\alpha_k}(y_k) - y_k \rangle} = L_k.$$

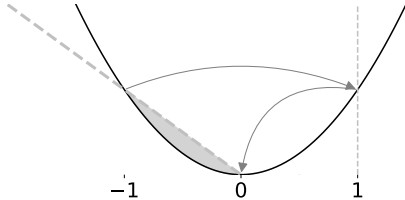

$$-1 \qquad 0 \qquad 1$$

Figure 5: Regular backtracking returns the greatest feasible step size after one adjustment.

*Therefore, after one function evaluation, adaptive backtracking returns $\rho v(\alpha_k)\alpha_k = \rho$, which matches the theoretical lower bound of backtracking and corresponds to the Lipschitz constant estimate $1/\rho$.*

**Example 2** (Fact 1). *Let $F$ be defined by $F(x) = x^2$ and fix $x_k = -1$, $d_k = -F'(x_k) = -2x_k = 2$. Then, the Armijo condition with $c = 1/4$ is satisfied if and only if*

$$(-1 + 2\alpha_k)^2 \leq 1 - \alpha_k.$$

*To find the critical step-size values for which this inequality is satisfied, we simply solve a second-order equation, which gives the positive value of $\alpha_k^* = 0.75$ with corresponding iterate $x_k + \alpha_k^* d_k = 0.5$. Thus, if the initial tentative step-size is $\alpha_k = 1$, which produces the tentative iterate is $1$, then the Armijo condition is not satisfied and the step-size must be adjusted. If $\rho = 0.75$, then the adjusted step-size $0.75$ produces the tentative iterate $0.5$ and the Armijo condition is satisfied, therefore the step-size requires no further adjustments. On the other hand, if $\rho = 0.8$, then the adjusted step-size $0.8$ produces the tentative iterate $0.6$ and the Armijo condition is not satisfied. Adjusting the step-size once more produces a step-size of $0.64 < \alpha_k^*$ and an corresponding iterate $x_k + \alpha_k d_k = -1 + 0.64 \cdot 2 = 0.28$, satisfying the Armijo condition. Therefore, increasing $\rho = 0.75$ to $\rho = 0.8$ decreases the step-size that backtracking returns.*

**Example 3** (Facts 2 and 3). *Let $F$ be defined by $F(x) = \cos x - ax$, where $a = \frac{1}{5\pi}$, fix $x_k = \frac{\pi}{2}$ and*

$$d_k = -F'(x_k) = \sin x_k + a = 1 + a.$$

*Given the above choices, the Armijo condition parameterized by $c = \frac{1}{2\pi}$ is satisfied if and only if*

$$\cos\left(\frac{\pi}{2} + (1+a)\alpha_k\right) - a\left(\frac{\pi}{2} + (1+a)\alpha_k\right) \leq \cos\left(\frac{\pi}{2}\right) - \frac{a\pi}{2} - (1+a)^2 \frac{\alpha_k}{2\pi},$$

*or, equivalently, if and only if*

$$\cos\left(\frac{\pi}{2} + (1+a)\alpha_k\right) \leq a(1+a)\alpha_k - (1+a)^2 \frac{\alpha_k}{2\pi}.$$

*If the initial tentative step-size is picked as $\frac{7\pi}{2(1+a)}$, then $(1+a)\alpha_k = \frac{7\pi}{2}$, so that*

$$\cos\left(\frac{\pi}{2} + 7\frac{\pi}{2}\right) = 1 \geq -1.16 \approx 7/10 - 7\frac{5\pi + 1}{20\pi}.$$

*That is, the Armijo condition is not satisfied, therefore the step-size must be adjusted. If $\rho = \frac{5}{7}$, then the step-size is adjusted to $\frac{5\pi}{2(1+a)}$, so that $(1+a)\alpha_k = \frac{5\pi}{2}$ and the Armijo condition is satisfied, since*

$$\cos\left(\frac{\pi}{2} + (1+a)\alpha_k\right) = \cos(3\pi) = -1 \leq -0.83 \approx \frac{1}{2} - 5\frac{5\pi + 1}{20\pi}.$$

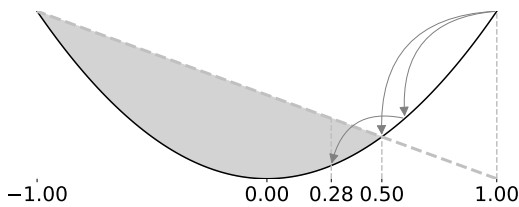

$$-1.00 \qquad 0.00 \quad 0.28 \; 0.50 \qquad 1.00$$

Figure 6: Reducing $\rho$ can lead to greater step-sizes.

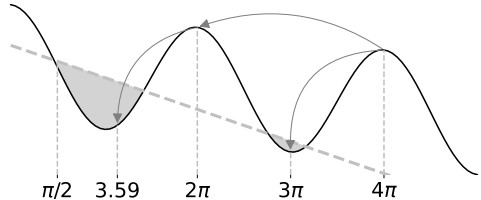

Figure 7: Reducing $\rho$ can lead to more step-size adjustments.

*On the other hand, if $\rho = \frac{3}{7}$, then $(1+a)\alpha_k = \frac{3\pi}{2}$ and the Armijo condition is not satisfied, since*

$$\cos\left(\frac{\pi}{2} + (1+a)\alpha_k\right) = 1 \geq -0.5 \approx 3/10 - 3\frac{5\pi+1}{20\pi}.$$

*Adjusting the step-size once more by $\rho = \frac{3}{7}$ produces the step-size $\frac{9\pi}{14(1+a)}$ and, in turn, the iterate $x_k + \alpha_k d_k = \frac{\pi}{2} + \frac{9\pi}{14} = \frac{8\pi}{7} \approx 3.59$, which is feasible since*

$$\cos\left(\frac{8\pi}{7}\right) \approx -1.13 \leq -0.21 \approx \frac{9}{70} - 9\frac{5\pi+1}{140\pi}.$$

*In this example, the step-size $\frac{3\pi}{2(1+a)}$ is not feasible although it is smaller than $\frac{5\pi}{2(1+a)}$, which is feasible. More generally, this establishes that feasibility is not monotone in the step-size for nonconvex functions. Moreover, by reducing $\rho = \frac{5}{7}$ to $\rho = \frac{3}{7}$, backtracking must adjust the initial step-size one additional time. Therefore, reducing $\rho$ might increase the number of criteria evaluations to return a feasible step-size.*

## B.2 CONVEX PROBLEMS

In this subsection, we present and proof the results for convex problems stated in Section 4.

**Proposition 3.** *Let $F$ be convex differentiable. Given a point $x_k$, a direction $d_k$ and a step-size $\alpha_k > 0$ satisfying (3) for some c, then $x_k$, $d_k$ and $\alpha'_k$ also satisfy (3) for any $\alpha'_k \in (0, \alpha_k)$.*

*Proof.* Let $\beta := \alpha'_k/\alpha_k \in (0, 1)$. Then, expressing $x_k + \alpha'_k d_k$ as $\beta(x_k + \alpha_k d_k) + (1-\beta)x_k$, we obtain

$$F(x_k + \alpha'_k d_k) = F(\beta(x_k + \alpha_k d_k) + (1-\beta)x_k) \leq \beta F(x_k + \alpha_k d_k) + (1-\beta)F(x_k)$$
$$\leq \beta(F(x_k) + c\alpha_k\langle\nabla F(x_k), d_k\rangle) + (1-\beta)F(x_k)$$
$$= c\alpha'_k\langle\nabla F(x_k), d_k\rangle + F(x_k),$$

where the first and second follow from $F$ being convex and $x_k$, $d_k$ and $\alpha_k$ satisfying (3), respectively. □

**Proposition 4.** *Let $F$ be convex differentiable. Fixing all other inputs, the number of backtracking criteria evaluations that Algorithm 1 takes to return a feasible step-size is nondecreasing in the input $\rho$.*

*Proof.* Consider the inputs $\alpha_0, c$ and $v$ to Algorithm 1 fixed. Then, let $0 < \rho_1 < \rho_2 < 1$ and let $N_1$ and $N_2$ denote the number of adjustments Algorithm 1 takes to compute a feasible step-size when it receives respectively $\rho_1$ and $\rho_2$ as inputs. If $\rho_i^{N_i}\alpha_0$ is a feasible step-size and $N'_i > N_i$ for some $i \in \{1, 2\}$, then so is $\rho_i^{N'_i}\alpha_0$, in view of the fact that $\rho_i^{N_i} < \rho_i^{N'_i}$ and of Proposition 3. Moreover, Algorithm 1 must test if the step-size $\rho_i^{N_i}$ is feasible before testing the step-size $\rho_1^{N'_1}$ and therefore cannot return $\rho_1^{N'_1}\alpha_0$. Inductively, we conclude that $N_1$ and $N_2$ are the least nonnegative integers such that $\rho_i^{N_i}$ are feasible. Now, since $\rho_2^{N_2}$ is feasible, if $N_1 > N_2$, then so is $\rho_1^{N_2} < \rho_2^{N_2}$, in view of the assumption that $\rho_1 < \rho_2$ and of Proposition 3. That is, $N_1$ is not the least nonnegative integer such that $\rho_1^{N_1}$ is feasible, a contradiction. Moreover, each adjustment requires evaluating the objective function once, so the total number of function evaluations Algorithm 1 takes to return a feasible step-size is $N_i + 2$. Therefore, if Algorithm 1 receives $\rho_1$ as input, then it takes **no more function evaluations** to return a feasible step-size than if receives $\rho_2$ as input.

□

**Definition 4.** *The inputs to Algorithms 1 and 2 are said to be* compatible *if $\alpha_0, c, v$ coincide and the input $\hat{\rho}$ to Algorithm 2 is parameterized by the same $\rho$ that Algorithm 1 takes as input.*

**Proposition 5.** *Let $F$ be convex differentiable. Given compatible inputs with a descent direction $d_k$ and $\epsilon < \rho$, Algorithm 2 takes no more function evaluations to return a feasible step-size than Algorithm 1 does.*

*Proof.* Suppose Algorithms 1 and 2 receive compatible inputs. If (3) is violated for some tentative step-size $\alpha_k$, then $v(\alpha_k) < 1$ which together with $c \in (-0, 1)$ imply $1 - c \cdot v(\alpha_k) > 1 - c > 0$. In turn, $\hat{\rho}(v(\alpha_k)) < \rho$ because $\epsilon < \rho$, by assumption. The result follows by repeating the arguments used to prove Proposition 4 above. □

### B.3 NONCONVEX PROBLEMS

#### B.3.1 ARMIJO CONDITION

**Proposition 6** (Armijo feasibility for $C^2$ functions)**.** *Let $F$ be twice continuously differentiable. Given a base point $x_k$, a descent direction $d_k$, an initial step-size $\alpha_0$ and a constant $c \in (0, 1)$ for the Armijo condition (3), there is some $\bar{\alpha} = \bar{\alpha}(x_k, d_k, c) \leq \alpha_0$ such that $x_k + \alpha_k d_k$ satisfies (3) for all $\alpha_k \in (0, \bar{\alpha})$.*

*Proof.* Assuming $F$ twice continuously differentiable, then by Taylor's theorem (Nocedal & Wright, 2006, p. 14), there exists some $t = t(x_k, d_k, \alpha_k) \in (0, 1)$ such that

$$F(x_k + \alpha_k d_k) = F(x_k) + \alpha_k \langle \nabla F(x_k), d_k \rangle + \alpha_k^2 \frac{1}{2} \langle d_k, \nabla^2 F(x_k + t\alpha_k d_k) d_k \rangle. \quad (13)$$

Moreover, the eigenvalues of $\nabla^2 F$ are continuous and the line segment $\{x_k + \alpha_k d_k : \alpha_k \in [0, \alpha_0]\}$ is compact, therefore there is some $\lambda > 0$ such that for all $\alpha_k \in (0, \alpha_0)$ and $t \in (0, 1)$

$$\left| d_k^\top \nabla^2 F(x_k + t\alpha_k d_k) d_k \right| \leq \lambda \|d_k\|^2. \quad (14)$$

So, let $\bar{\alpha} = \bar{\alpha}(x_k, d_k, c) := 2(1 - c) \langle \nabla F(x_k), -d_k \rangle / (\lambda \|d_k\|^2) > 0$, which is positive since $d_k$ is a descent direction, by assumption. Combining (13) with (14), it follows that if $\alpha_k \in (0, \bar{\alpha})$, then (3) holds. □

For the sake of convenience, we now restate some definition from Section 4.

**Definition 5** (Smoothness)**.** *A function $F$ is said to be* Lipschitz-smooth *if (5) holds for some $L > 0$.*

**Definition 6** (Gradient related)**.** *The directions $d_k$ are said to be* gradient related *if there are $c_1 > 0$ and $c_2 > 0$ such that $\langle \nabla F(x_k), -d_k \rangle \geq c_1 \|\nabla F(x_k)\|^2$ and $\|d_k\| \leq c_2 \|\nabla F(x_k)\|$, for all $k \geq 0$.*

Given a Lipschitz-smooth function $F$, we are particularly interested in applying the Descent Lemma (5) with $x = x_k$ and $y = x_k + \alpha_k d_k$, which gives

$$F(x_k + \alpha_k d_k) \leq F(x_k) + \alpha_k \langle \nabla F(x_k), d_k \rangle + \alpha_k^2 \frac{L}{2} \|d_k\|^2. \quad (15)$$

**Proposition 7** (Armijo feasibility for Lipschitz-smooth functions)**.** *Let $F$ be Lipschitz-smooth. Given a base point $x_k$, a descent direction $d_k$, an initial step-size $\alpha_0$ and a constant $c \in (0, 1)$ for the Armijo condition (3), there is some $\bar{\alpha} = \bar{\alpha}(x_k, d_k, c) \leq \alpha_0$ such that (3) holds for all $\alpha_k \in (0, \bar{\alpha})$. If, in addition, $d_k$ are gradient related, then (3) holds for all $\alpha_k \in (0, \frac{2(1-c)c_1}{Lc_2^2}]$, independent of $x_k$ and $d_k$.*

*Proof.* To guarantee (3) holds, we impose that the right-hand side of (15) is less than the right-hand side of (3):

$$F(x_k) + \alpha_k \langle \nabla F(x_k), d_k \rangle + \alpha_k^2 \frac{L}{2} \|d_k\|^2 \leq F(x_k) + c\alpha_k \langle \nabla F(x_k), d_k \rangle.$$

In turn, simplifying the above inequality, it follows that if

$$\alpha_k \leq \frac{2(1-c)\langle \nabla F(x_k), -d_k \rangle}{L \|d_k\|^2}, \quad (16)$$

then (3) holds, where we note that (16) is positive, since $d_k$ is assumed a descent direction.

Now, suppose that $\langle \nabla F(x_k), -d_k \rangle \geq c_1 \|\nabla F(x_k)\|^2$ and $\|d_k\| \leq c_2 \|\nabla F(x_k)\|$ for some $c_1 > 0$ and $c_2 > 0$. Then, for all $\alpha_k$ such that $\alpha_k \leq 2(1-c)c_1/Lc_2^2$, we have that

$$\alpha_k \leq \frac{2(1-c)}{L} \frac{c_1 \|\nabla F(x_k)\|^2}{c_2^2 \|\nabla F(x_k)\|^2} \leq \frac{2(1-c)\langle \nabla F(x_k), -d_k \rangle}{L \|d_k\|^2}.$$

That is, (16) holds. Therefore, (3) also holds. $\qquad\square$

**Proposition 8.** *Let $F$ be Lipschitz-smooth, $\epsilon < \rho$ and assume $v, \hat{\rho}$ are given by (4). Also, suppose $d_k$ are gradient related. If Algorithms 1 and 2 receive $\alpha_k$ as the initial step-size input at iteration $k+1$ for all $k \geq 0$, then they evaluate (3) at most $\lfloor \log_\rho(\bar{\alpha}/\alpha_0) \rfloor + 1 + k$ times up to iteration $k$, where $\bar{\alpha} := 2(1-c)c_1/Lc_2^2$. If, on the other hand, Algorithms 1 and 2 receive $\alpha_0$ as the initial step-size input at every iteration, then they evaluate (3) at most $k(\lfloor \log_\rho(\bar{\alpha}/\alpha_0) \rfloor + 1)$ times up to iteration $k$.*

*Proof.* Suppose that Algorithm 2 evaluates (3) and it does not hold for a given tentative step-size $\alpha_k$. Then,

$$F(x_k + \alpha_k d_k) - F(x_k) > c\alpha_k \langle \nabla F(x_k), d_k \rangle.$$

Dividing both sides above by $c\alpha_k \langle \nabla F(x_k), d_k \rangle < 0$ gives $v(\alpha_k) < 1$. In turn, since $c \in (0,1)$, it follows that $1 - c > 1 - cv(\alpha_k) > 0$ and $(1-c)/(1 - c \cdot v(\alpha_k)) < 1$. Plugging this inequality into (4b), we obtain

$$\hat{\rho}(\alpha_k) = \max(\epsilon, \rho(1-c)/(1 - c \cdot v(\alpha_k))) < \rho,$$

since by assumption $\epsilon < \rho$. Therefore, if (3) does not hold for a given tentative step-size, then Algorithms 1 and 2 multiply it by a factor of at most $\rho$ to adjust it.

Moreover, by Proposition 7, (3) is satisfied for all $\alpha_k \in (0, \bar{\alpha})$, independently of $x_k$ and $d_k$.

Hence, if Algorithms 1 and 2 use $\alpha_0$ as the initial step-size for the first iteration and $\alpha_k$ at iteration $k+1$ for $k \geq 0$, then at most $\lfloor \log_\rho(\bar{\alpha}/\alpha_0) \rfloor + 1$ adjustments are necessary until a step-size that is uniformly feasible is found. Each adjustment entails evaluating (3) once. In addition, (3) must be evaluated once every iteration. Therefore, (3) is evaluated at most $\lfloor \log_\rho(\bar{\alpha}/\alpha_0) \rfloor + 1 + k$ times up to iteration $k$.

Now, suppose Algorithms 1 and 2 use $\alpha_0$ as the initial step-size in every iteration. Then, at most $\lfloor \log_\rho(\bar{\alpha}/\alpha_0) \rfloor + 1$ adjustments are necessary in every iteration until a feasible step-size is found. As before, each adjustment entails evaluating (3) once, in addition to the first evaluation. Therefore, (3) is evaluated at most $k(\lfloor \log_\rho(\bar{\alpha}/\alpha_0) \rfloor + 1)$ times up to iteration $k$. $\qquad\square$

**Proposition 9** (step-size lower bounds). *Let $F$ be Lipschitz-smooth. Also, suppose $d_k$ are gradient related. Given appropriate inputs, Algorithm 2 with the choices specified by (4) and Algorithm 1 return a step-size $\alpha_k$ such that*

$$\alpha_k \geq \min \left\{ \alpha_0, \rho \frac{2(1-c)c_1}{Lc_2^2} \right\} > 0.$$

*Proof.* Since $d_k$ is a descent direction, dividing both sides of (15) by $\langle \nabla F(x_k), -d_k \rangle > 0$ yields

$$-cv(\alpha_k) = -\frac{F(x_k + \alpha_k d_k) - F(x_k)}{\alpha_k \langle \nabla F(x_k), d_k \rangle} \leq -1 - \frac{\alpha_k L \|d_k\|^2}{2\langle \nabla F(x_k), d_k \rangle}.$$

Hence, if $\hat{\rho}$ is chosen as (4b) and (3) does not hold, then step-sizes $\alpha_k$ returned by Algorithm 2 satisfy

$$\hat{\rho}(v(\alpha_k))\alpha_k \geq \rho \frac{1-c}{1 - cv(\alpha_k)} \alpha_k \geq \rho \frac{2(1-c)\langle \nabla F(x_k), -d_k \rangle}{L \|d_k\|^2} \geq \rho \frac{2(1-c)c_1}{Lc_2^2} > 0.$$

Moreover, by Proposition 7, the greatest step-size for which (3) is guaranteed to hold is $2(1 - c)c_1/Lc_2^2$. If $\alpha_0 \geq 2(1-c)c_1/Lc_2^2$, then Algorithm 1 returns a step-size at least within a $\rho$ factor of $2(1-c)c_1/Lc_2^2$. $\qquad\square$

B.3.2  DESCENT LEMMA

First, note the proximal operator $p_{\alpha_k}$ given by (6) is well-defined. Indeed, given a continuous convex function $g$, a point $y_k$ and some $\alpha_k > 0$, the map $x \mapsto g(x) + (1/2\alpha_k)\|x - (y - \alpha_k \nabla f(y))\|^2$ is continuous strongly convex and therefore admits a unique minimum.

**Proposition 10** (Lipschitz step-size feasibility). *Let $f$ be Lipschitz-smooth convex and let $g$ be continuous convex. Also, suppose $v$ and $\hat{\rho}$ are chosen as (8a) and (8b). If $\alpha_k \in (0, 1/L)$, then (7) holds for all $y_k$. If Algorithms 1 and 2 receive $\alpha_k$ as the initial step-size input in iteration $k + 1$, then they evaluate (7) at most $\lfloor\log_\rho(1/L\alpha_0)\rfloor + 1 + k$ times up to iteration $k$. If, on the other hand, Algorithms 1 and 2 receive $\alpha_0$ as the initial step-size input in every iteration, then they evaluate (7) at most $k(\lfloor\log_\rho(1/L\alpha_0)\rfloor + 1)$ times up to iteration $k$.*

*Proof.* Given any $y_k$, if $\alpha_k \in (0, 1/L)$, then applying (5) with $x = p_{\alpha_k}(y_k)$ and $y = y_k$, we get

$$f(p_{\alpha_k}(y_k)) \le f(y_k) + \langle\nabla f(y_k), p_{\alpha_k}(y_k) - y_k\rangle + (L/2)\|p_{\alpha_k}(y_k) - y_k\|^2$$
$$\le f(y_k) + \langle\nabla f(y_k), p_{\alpha_k}(y_k) - y_k\rangle + (1/2\alpha_k)\|p_{\alpha_k}(y_k) - y_k\|^2.$$

Adding $\psi(p_{\alpha_k}(y_k))$ to both sides, we recover (7). Thus, if $\alpha_k \in (0, \bar{\alpha})$, then (7) holds for all $y_k$.

Given an initial step-size $\alpha_k$ and the points $y_k$ and $p_{\alpha_k}(y_k)$, Algorithm 1 checks if (7) holds. If it does hold, then Algorithm 1 returns $\alpha_k$, otherwise Algorithm 1 adjusts $\alpha_k$ by $\rho$, recomputes $p_{\alpha_k}(y_k)$, checks if (7) and repeats. Since (7) is guaranteed to hold for $\alpha_k \in (0, 1/L)$, given an initial step-size $\alpha_0$, Algorithm 1 computes a feasible step-size after adjusting $\alpha_k$ at most $\lfloor\log_\rho(1/L\alpha_0)\rfloor + 1$ times. Each time Algorithm 1 adjusts $\alpha_k$, Algorithm 1 evaluates (7). In addition, Algorithm 1 evaluates Eq. (7) once every time it is called to check if the initial step-size is feasible. Hence, if Algorithm 1 receives $\alpha_k$ as the initial step-size input at iteration $k + 1$, then it evaluates Eq. (7) at most $\lfloor\log_\rho(1/L\alpha_0)\rfloor + 1 + k$ times up to iteration $k$. On the other hand, if Algorithm 1 receives the same $\alpha_0$ as initial step-size input at every iteration, then it might have to adjust $\alpha_k$ up to $\lfloor\log_\rho(1/L\alpha_0)\rfloor + 1$ in every iteration, therefore Algorithm 1 evaluates (7) at most $k(\lfloor\log_\rho(1/L\alpha_0)\rfloor + 1)$ times up to iteration $k$.

Now, consider Algorithm 2, with $v$ and $\hat{\rho}$ chosen as (8a) and (8b). Given an initial step-size $\alpha_k$ and the points $y_k$ and $p_{\alpha_k}(y_k)$, Algorithm 2 checks if (7) holds. Suppose (7) does not hold. Then, moving the terms $f(p_{\alpha_k}(y_k))$ and $\langle\nabla f(y_k), p_{\alpha_k}(y_k) - y_k\rangle$ to the left-hand side and cancelling $\psi(p_{\alpha_k}(y_k))$ on both sides, we obtain

$$f(p_{\alpha_k}(y_k)) - f(y_k) - \langle\nabla f(y_k), p_{\alpha_k}(y_k) - y_k\rangle > (1/2\alpha_k)\|p_{\alpha_k}(y_k) - y_k\|^2. \tag{17}$$

Since $\|\cdot\| \ge 0$, the left-hand side must be positive. So, dividing both sides by the left-hand side and using (8a), it follows that $v(\alpha_k) < 1$. Hence, Algorithm 2 adjusts $\alpha_k$ to $\hat{\rho}(\alpha_k)\alpha_k < \rho\alpha_k$. That is, the factor by which Algorithm 2 adjusts $\alpha_k$ is smaller than the factor by which Algorithm 1 adjusts $\alpha_k$. Therefore, Algorithm 2 evaluates Eq. (7) at most as many times as (1) does. $\square$

**Proposition 11.** *Let $f$ be Lipschitz-smooth convex and let $g$ be continuous convex. Also, suppose $v$ and $\hat{\rho}$ are chosen as (8a) and (8b). If Algorithms 1 and 2 receive an initial step-size $\alpha_0 > 0$, then they return a feasible step-size $\alpha_k$ such that $\alpha_k \ge \min\{\alpha_0, \rho/L\}$.*

*Proof.* By Proposition 10, every step-size $\alpha_k \in (0, 1/L)$ is feasible. Hence, if $\alpha_0$ is not feasible, then since Algorithm 1 adjusts step-sizes by $\rho$, it must return a feasible step-size within a $\rho$ factor of $1/L$.

Now, consider Algorithm 2, with $v$ and $\hat{\rho}$ chosen as (8a) and (8b). Algorithm 2 only adjusts $\alpha_k$ when (7) does not hold, so suppose that is the case. Applying (5) with $y = y_k$ and $x_k = p_{\alpha_k}(y_k)$ yields

$$f(p_{\alpha_k}(y_k)) - f(x_k) - \langle\nabla f(y_k), p_{\alpha_k}(y_k) - y_k\rangle \le (L/2)\|p_{\alpha_k}(y_k) - y_k\|^2.$$

Dividing both sides by $(L/\rho)(f(p_{\alpha_k}(y_k)) - f(x_k) - \langle\nabla f(y_k), p_{\alpha_k}(y_k) - y_k\rangle)$, which is positive by (17), we obtain

$$\frac{\rho}{L} \le \rho\frac{\frac{1}{2}\|p_{\alpha_k}(y_k) - y_k\|^2}{f(p_{\alpha_k}(y_k)) - f(x_k) - \langle\nabla f(y_k), p_{\alpha_k}(y_k) - y_k\rangle} = \rho v(\alpha_k)\alpha_k,$$

where the identity follows from (8a). Hence, Algorithm 2 adjusts $\alpha_k$ to $\hat{\rho}(v(\alpha_k))\alpha_k \ge \rho/L$. $\square$

## B.4 CONVERGENCE RESULTS

### B.4.1 A GENERAL CONVERGENCE RESULT FOR ADAPTIVE BACKTRACKING

Under mild conditions, we now show that $\lim_{k \to +\infty} \|\nabla f(x_k)\|^2 = 0$ for iterates $x_k$ in the form (2) with gradient related $d_k$ and step-sizes generated by adaptive backtracking. We emphasize that the following results make no further assumptions on how the descent directions are generated and that (Nocedal & Wright, 2006, p. 40):

*For line search methods of the general form (2), the limit $\lim_{k \to +\infty} \|\nabla f(x_k)\|^2 = 0$ is the strongest global convergence result that can be obtained: We cannot guarantee that the method converges to a minimizer, but only that it is attracted by stationary points. Only by making additional requirements on the search direction $d_k$—by introducing negative curvature information from the Hessian $\nabla^2 f(x_k)$, for example—can we strengthen these results to include convergence to a local minimum*

**Proposition 12.** *Let $f$ be bounded below and Lipschitz-smooth on an open set containing the level set $\{x : f(x) \leq f(x_0)\}$, where $x_0$ is the initial point of iterates (2) where $d_k$ are gradient related and $\alpha_k$ are generated by adaptive backtracking (Algorithm 2) with some $\alpha_0 > 0$ and using $\hat{\rho}$ and $v$ given by (4). Then, $\lim_{k \to +\infty} \|\nabla f(x_k)\|^2 = 0$.*

*Proof.* Under the above assumptions, we have that $\alpha_k \geq \min\{\alpha_0, \rho\overline{\alpha}\}$, where $\overline{\alpha} = 2(1-c)c_1/(Lc_2^2)$, by Proposition 9. Moreover, we have that $\langle \nabla f(x_k), d_k \rangle \leq -c_1 \|\nabla f(x_k)\|^2$, because $d_k$ are gradient related. Hence, since adaptive backtracking enforces the Armijo condition, (3), it follows that

$$f(x_{k+1}) - f(x_k) \leq -\alpha_k c \|\nabla f(x_k)\|^2 \leq -c \min\{\alpha_0, \rho 2(1-c)c_1/(Lc_2^2)\} \|\nabla f(x_k)\|^2.$$

Telescoping the above difference, we get

$$f(x_{k+1}) - f(x_0) = \sum_{t=1}^{k} (f(x_{t+1}) - f(x_t)) \leq -c \min\left\{\alpha_0, \rho\frac{2(1-c)c_1}{Lc_2^2}\right\} \sum_{t=1}^{k} \|\nabla f(x_t)\|^2.$$

Rearranging the above inequality and using the assumption that $f$ is lower bounded, we obtain

$$c \min\left\{\alpha_0, \rho\frac{2(1-c)c_1}{Lc_2^2}\right\} \sum_{t=1}^{k} \|\nabla f(x_t)\|^2 \leq f(x_0) - f(x_{k+1}) < +\infty.$$

That is, $\|\nabla f(x_k)\|^2$ are square-summable. Therefore, it follows that

$$\lim_{k \to +\infty} \|\nabla f(x_k)\|^2 = 0.$$

$\square$

### B.4.2 CONVERGENCE RESULTS FOR GRADIENT DESCENT

We show that the standard convergence results for gradient descent are preserved if step-sizes are generated by adaptive backtracking. We address smooth and then smooth strongly convex objectives.

**Proposition 13.** *Let $f$ be convex, Lipschitz-smooth and suppose $\nabla f(x^*) = 0$ for some $x^*$. If the step-sizes $\alpha_k$ of gradient descent (Algorithm 3) are chosen by adaptive backtracking (Algorithm 2) using $\hat{\rho}$ and $v$ given by (4) with $c \in [1/2, 1)$ and $\alpha_0 > 0$, then $\alpha_k \geq \min\{\alpha_0, \rho\overline{\alpha}\}$, where $\overline{\alpha} = 2(1-c)/L$, and*

$$f(x_k) - f(x^*) \leq \frac{\|x_0 - x^*\|^2}{2\min\{\alpha_0, \rho\overline{\alpha}\}k}.$$

*Proof.* Under the above assumptions, all iterates of gradient descent satisfy the Armijo condition, (3). Moreover, since $f$ is convex, we have that

$$f(x_k) \leq f(x^*) + \langle \nabla f(x_k), x_k - x^* \rangle.$$

Hence, combining the above inequality with (3), it follows that

$$f(x_{k+1}) \leq f(x_k) - c\alpha_k \|\nabla f(x_k)\|^2 \leq f(x^*) + \langle \nabla f(x_k), x_k - x^* \rangle - c\alpha_k \|\nabla f(x_k)\|^2.$$

In turn, since $c \geq 1/2$, rearranging the above inequality and completing a square, we get

$$
\begin{aligned}
f(x_{k+1}) - f(x^*) &\leq \frac{1}{2\alpha_k} (2\alpha_k \langle \nabla f(x_k), x_k - x^* \rangle - \alpha_k^2 \|\nabla f(x_k)\|^2) \\
&= \frac{1}{2\alpha_k} (2\alpha_k \langle \nabla f(x_k), x_k - x^* \rangle - \alpha_k^2 \|\nabla f(x_k)\|^2 \pm \|x_k - x^*\|^2) \\
&= \frac{1}{2\alpha_k} (\|x_k - \alpha_k \nabla f(x_k) - x^*\|^2 - \|x_k - x^*\|^2) \\
&= \frac{1}{2\alpha_k} (\|x_k - x^*\|^2 - \|x_{k+1} - x^*\|^2).
\end{aligned}
$$

Now, since gradient descent sets $d_k = -\nabla f(x_k)$, then $d_k$ are gradient related with $c_1 = c_2 = 1$. Moreover, since $f$ is Lipschitz-smooth, then $\alpha_k \geq \min\{\alpha_0, \rho\overline{\alpha}\}$, where $\overline{\alpha} = 2(1-c)/L$, by Proposition 9. Plugging this lower bound into the above inequality, it follows that

$$
f(x_{k+1}) - f(x^*) \leq \frac{1}{2\min\{\alpha_0, \rho\overline{\alpha}\}} (\|x_k - x^*\|^2 - \|x_{k+1} - x^*\|^2).
$$

Telescoping the above, we get

$$
\begin{aligned}
\sum_{t=1}^{k} (f(x_{t+1}) - f(x^*)) &\leq \frac{1}{2\min\{\alpha_0, \rho\overline{\alpha}\}} \sum_{t=1}^{k} (\|x_t - x^*\|^2 - \|x_{t+1} - x^*\|^2) \\
&\leq \frac{\|x_0 - x^*\|^2 - \|x_{k+1} - x^*\|^2}{2\min\{\alpha_0, \rho\overline{\alpha}\}} \\
&\leq \frac{\|x_0 - x^*\|^2}{2\min\{\alpha_0, \rho\overline{\alpha}\}}.
\end{aligned}
$$

Since $\nabla f(x^*) = 0$ and $f$ is convex, we have that $f(x_{k+1}) - f(x^*) \geq 0$. Moreover, $f(x_k)$ are decreasing because the Armijo condition holds in every iteration. Therefore

$$
f(x_{k+1}) - f(x^*) \leq \frac{\|x_0 - x^*\|^2}{2\min\{\alpha_0, \rho\overline{\alpha}\}k}.
$$

$\square$

Next, we show that adaptive backtracking also preserves the convergence rate of gradient descent on strongly convex objectives, which we define below.

**Definition 7** (Strong convexity). *A continuously differentiable function $f$ is said to be strongly convex if there exists some $m > 0$ such that for every $x$ and $y$*

$$
f(y) \geq f(x) + \langle \nabla f(x), y - x \rangle + \frac{m}{2} \|y - x\|^2. \tag{18}
$$

**Proposition 14.** *Let $f$ be Lipschitz-smooth and strongly convex. If the step-sizes $\alpha_k$ of gradient descent (Algorithm 3) are chosen by adaptive backtracking (Algorithm 2) using $\hat{\rho}$ and $v$ given by (4) with $c \in [1/2, 1)$ and $\alpha_0 \in (0, 1/m)$, then*

$$
f(x_k) - f(x^*) \leq (1 - m\min\{\alpha_0, \rho\overline{\alpha}\})^k \frac{L+m}{2} \|x_0 - x^*\|^2.
$$

*In particular, if $c = 1/2$ and $\alpha_0 > \rho/L$, then*

$$
f(x_k) - f(x^*) \leq (1 - \rho q)^k \frac{L+m}{2} \|x_0 - x^*\|^2,
$$

*where $q = m/L$ is the reciprocal of the condition number of $f$.*

*Proof.* Let $L$ and $m$ denote the Lipschitz-smoothness and strong convexity constants of $f$. The assumption that $f$ is strongly convex implies the existence of a unique global minimizer $x^*$ for $f$. We then use $x^*$ to define a Lyapunov function $V$, given by

$$
V(x_k) = f(x_k) - f(x^*) + \frac{m}{2} \|x_k - x^*\|^2,
$$

which is positive for $x_k \neq x^*$. To prove the result, we show that $(1+\delta_k)V(x_{k+1}) - V(x_k) \leq 0$, where

$$\delta_k = \frac{1}{Q_k - 1}, \qquad Q_k = \frac{L_k}{m} \qquad \text{and} \qquad L_k = \frac{1}{\alpha_k}.$$

And we note that the assumption that $\alpha_k \leq \alpha_0 < 1/m$ implies $L_k > m$, thus $\delta_k$ are well-defined.

By assumption, the iterates of gradient descent satisfy (3) with $c \in [1/2, 1)$, hence

$$f(x_{k+1}) - f(x_k) \leq -c\alpha_k \|\nabla f(x_k)\|^2 \leq -\frac{\alpha_k}{2}\|\nabla f(x_k)\|^2.$$

Moreover, by strong convexity, we have that

$$f(x_k) - f(x^*) \leq \langle \nabla f(x_k), x_k - x^* \rangle - \frac{m}{2}\|x_k - x^*\|^2.$$

Next, expanding quadratic terms, it follows that

$$(1+\delta_k)\|x_{k+1} - x^*\|^2 - \|x_k - x^*\|^2 = (1+\delta_k)(\alpha_k^2\|\nabla f(x_t)\|^2 - 2\alpha_k\langle \nabla f(x_k), x_k - x^* \rangle) \\ + \delta_k\|x_k - x^*\|^2.$$

Now, from the definition of $\delta_k$, we obtain

$$(1+\delta_k)(1 - m\alpha_k) = \frac{Q_k}{Q_k - 1}\frac{Q_k - 1}{Q_k} = 1 \quad \text{and} \quad (1+\delta_k)m\alpha_k = \frac{Q_k}{Q_k - 1}\frac{1}{Q_k} = \delta_k.$$

Then, we put everything together to get

$$(1+\delta_k)V(x_{k+1}) - V_k(x_k) \leq -(1+\delta_k)(1 - m\alpha_k)\frac{\alpha_k}{2}\|\nabla f(x_k)\|^2 \\ - (\delta_k - (1+\delta_k)m\alpha_k)\langle \nabla f(x_k), x_k - x^* \rangle \\ \leq -\frac{\alpha_k}{2}\|\nabla f(x_k)\|^2.$$

Applying the above inequality inductively, it follows that

$$V(x_{k+1}) \leq V(x_0)\prod_{t=1}^{k}\frac{1}{1+\delta_t}.$$

Moreover, applying (5) with $y = x_0$ and $x = x^*$, and noticing that $\nabla f(x^*) = 0$, we obtain

$$V(x_0) = f(x_0) - f(x^*) + \frac{m}{2}\|x_0 - x^*\|^2 \leq \frac{L+m}{2}\|x_0 - x^*\|^2.$$

Furthermore, under the above assumptions, we have that $\alpha_k \geq \min\{\alpha_0, \rho\overline{\alpha}\}$, where $\overline{\alpha} = 2(1-c)/L$, which implies that

$$1 + \delta_k = \frac{Q_k}{Q_k - 1} = \frac{1}{1 - m\alpha_k} \geq \frac{1}{1 - m\min\{\alpha_0, \rho\overline{\alpha}\}}.$$

Finally, we put everything together and obtain

$$f(x_k) - f(x^*) \leq V(x_{k+1}) \leq \frac{L+m}{2}\|x_0 - x^*\|^2 \prod_{t=1}^{k}\frac{1}{1+\delta_t} \\ \leq (1 - m\min\{\alpha_0, \rho\overline{\alpha}\})^k\frac{L+m}{2}\|x_0 - x^*\|^2.$$

$\square$

### B.4.3 A CONVERGENCE RESULT FOR ACCELERATED GRADIENT DESCENT

To establish that adaptive backtracking preserves the convergence rate of accelerated gradient descent, we employ a Lyapunov argument based on the function $V_k$ defined by

$$V_t(x_k, y_k) = f(y_k) - f(x^*) + \frac{m}{2}\|z_t - x^*\|^2, \tag{19}$$

where the point $z_t = z_t(x_k, y_k)$ is defined as

$$z_t = x_k + \sqrt{Q_{t-1}}(x_k - y_k), \tag{20}$$

and the estimated condition number $Q_t$ and estimated Lipschitz constant are given by

$$Q_t = \begin{cases} L_0/m, & t < 0, \\ L_t/m, & t \geq 0, \end{cases} \qquad \text{and} \qquad L_t = \frac{1}{\alpha_t}. \tag{21}$$

Note that the index $t$ of $z_t$ follows that of $V_t$ but is independent of the indices of $x_k$ and $y_k$, which allows us to split the Lyapunov analysis in two auxiliary lemmas. First, we show that for a fixed index $k+1$, the Lyapunov function $V_{k+1}$ decreases along consecutive AGD iterates at an accelerated rate. Second, we bound by how much $V_{k+1}$ can increase with respect to $V_k$ for the same AGD iterate.

**Lemma 1.** *Let $f$ be Lipschitz-smooth and strongly convex. If the Lipschitz constant estimates $L_k$ of accelerated gradient descent (Algorithm 4) are generated by adaptive backtracking (Algorithm 2) using $\hat{\rho}$ and $v$ given by (4) with $c \in [1/2, 1)$ and $L_0 > m$, then for $k \geq 0$*

$$(1 + \delta_{k+1})V_{k+1}(y_{k+1}, x_{k+1}) - V_{k+1}(y_k, x_k) \leq 0,$$

*where $\delta_{k+1} = 1/(\sqrt{Q_k} - 1)$.*

*Proof.* We start by splitting $(1 + \delta_{k+1})(f(y_{k+1}) - f(x^*))$ into three further differences:

$$(1 + \delta_{k+1})(f(y_{k+1}) - f(x^*)) - (f(y_k) - f(x^*)$$
$$= (1 + \delta_{k+1})(f(y_{k+1}) - f(x_k)) + \delta_{k+1}(f(x_k) - f(x^*)) + (f(x_k) - f(y_k)).$$

Since $c \in [1/2, 1)$, then adaptive backtracking generates $L_k$ such that

$$(1 + \delta_{k+1})(f(y_{k+1}) - f(x_k)) \leq -(1 + \delta_{k+1})\frac{1}{2L_k}\|\nabla f(x_k)\|^2. \tag{22}$$

Moreover, applying (18) with $x = x_k$ and $y = x^*$ and using that $f$ is convex, we get

$$\delta_{k+1}(f(x_k) - f(x^*)) \leq \delta_{k+1}\langle \nabla f(x_k), x_k - x^* \rangle - \delta_{k+1}\frac{m}{2}\|x_k - x^*\|^2, \tag{23}$$

$$f(x_k) - f(y_k) \leq \langle \nabla f(x_k), x_k - y_k \rangle. \tag{24}$$

Next, we express the difference $z_{k+1} - x^*$ as

$$z_{k+1} - x^* = x_{k+1} + \sqrt{Q_k}(x_{k+1} - y_{k+1}) - x^*$$
$$= y_{k+1} + \beta_k(y_{k+1} - y_k) + \sqrt{Q_k}\beta_k(y_{k+1} - y_k) - x^*$$
$$= -\frac{1}{L_k}(1 + \beta_k(1 + \sqrt{Q_k}))\nabla f(x_k) + \beta_k(1 + \sqrt{Q_k})(x_k - y_k) + x_k - x^*$$
$$= -\frac{1}{L_k}\sqrt{Q_k}\nabla f(x_k) + (\sqrt{Q_k} - 1)(x_k - y_k) + x_k - x^*,$$

where we used the identities

$$1 + \beta_k(1 + \sqrt{Q_k}) = \sqrt{Q_t} \qquad \text{and} \qquad \beta_k(1 + \sqrt{Q_k}) = \sqrt{Q_k} - 1.$$

In the same vein, when expanding the 2-norm term $\|z_{k+1} - x^*\|^2$ below, we use the following identities after colons to simplify the coefficients of terms before colons:

$$\|\nabla f(x_k)\|^2 : \qquad (Q_k/L_k^2)(m/2) = 1/2L_k,$$
$$\langle \nabla f(x_k), x_k - y_k \rangle : \qquad m(1 + \delta_{k+1})\sqrt{Q_k}(\sqrt{Q_k} - 1)/L_k = 1,$$
$$\langle \nabla f(x_k), x_k - x^* \rangle : \qquad m(1 + \delta_{k+1})\sqrt{Q_k}/L_k = \delta_k,$$
$$\|x_k - y_k\|^2 : \qquad (1 + \delta_{k+1})(\sqrt{Q_k} - 1)^2 = \sqrt{Q_k}(\sqrt{Q_k} - 1),$$
$$\langle x_k - y_k, x_k - x^* \rangle : \qquad (1 + \delta_{k+1})(\sqrt{Q_k} - 1) = \sqrt{Q_k}.$$

As a result, the 2-norm difference in $(1 + \delta_{k+1})V_{k+1}(y_{k+1}, x_{k+1}) - V_{k+1}(y_k, x_k)$ becomes

$$
(1 + \delta_{k+1})\frac{m}{2}\|z_{k+1} - x^*\|^2 - \frac{m}{2}\|x_k - x^* + \sqrt{Q_k}(x_k - y_k)\|^2
$$

$$
= \frac{1 + \delta_{k+1}}{2L_k}\|\nabla f(x_k)\|^2 - \langle \nabla f(x_k), x_k - y_k \rangle - \delta_k \langle \nabla f(x_k), x_k - x^* \rangle
$$

$$
\frac{m}{2}\sqrt{Q_k}(\sqrt{Q_k} - 1)\|x_k - y_k\|^2 + \frac{m}{2}(2\sqrt{Q_k}\langle x_k - y_k, x_k - x^* \rangle + (1 + \delta_{k+1})\|x_k - x^*\|^2)
$$

$$
- \frac{m}{2}(Q_k\|x_k - y_k\|^2 + 2\sqrt{Q_k}\langle x_k - y_k, x_k - x^* \rangle + \|x_k - x^*\|^2)
$$

$$
= \frac{1 + \delta_{k+1}}{2L_k}\|\nabla f(x_k)\|^2 - \langle \nabla f(x_k), x_k - y_k \rangle - \delta_k \langle \nabla f(x_k), x_k - x^* \rangle
$$

$$
- \frac{m}{2}\sqrt{Q_k}\|x_k - y_k\|^2 + \delta_k\frac{m}{2}\|x_k - x^*\|^2. \tag{25}
$$

Finally, combining (22) to (25) and then canceling several terms, we obtain

$$
(1 + \delta_{k+1})V_{k+1}(y_{k+1}, x_{k+1}) - V_{k+1}(y_k, x_k) \le -\frac{m}{2}\sqrt{Q_k}\|x_k - y_k\|^2 \le 0.
$$

$\square$

**Lemma 2.** *Let $f$ be Lipschitz-smooth strongly convex. Given initial points $x_0 = y_0$, if the estimates $L_k$ of the Lipschitz constant in accelerated gradient descent (Algorithm 4) are generated monotonically by adaptive backtracking (Algorithm 2 with $L_k$ serving as the initial estimate for $L_{k+1}$) using $\hat\rho$ and $v$ given by (4) with $c \in [1/2, 1)$ and $L_0 > m$, then for $k \ge 0$*

$$
V_{k+1}(y_k, x_k) \le \frac{Q_k^2}{Q_{k-1}^2}V_k(y_k, x_k).
$$

*Proof.* We argue by induction. If $x_0$ and $y_0$ match, then

$$
z_1(y_0, x_0) = x_0 + Q_0(x_0 - y_0) = x_0 = x_0 + Q_{-1}(x_0 - y_0) = z_0(y_0, x_0).
$$

Moreover, $Q_{-1} = Q_0$, by definition. Therefore, we have that

$$
\begin{aligned}
V_1(y_0, x_0) &= f(y_0) - f(x^*) + \frac{m}{2}\|z_1(y_0, x_0) - x^*\|^2 \\
&= \frac{Q_0^2}{Q_{-1}^2}(f(y_0) - f(x^*) + \frac{m}{2}\|z_0(y_0, x_0) - x^*\|^2) \\
&= \frac{Q_0^2}{Q_{-1}^2}V_0(y_0, x_0),
\end{aligned}
$$

which establishes the base case. To prove the inductive step, we divide the analysis in two cases, each representing a possible sign of $\langle x_k - y_k, x_k - x^* \rangle$. For each case, we bound

$$
\begin{aligned}
&\|x_k - x^* + \sqrt{Q_k}x_k - y_k\|^2 - \|z_k - x^*\|^2 \\
&= 2(\sqrt{Q_k} - \sqrt{Q_{k-1}})\langle x_k - x^*, x_k - y_k \rangle + (Q_k - Q_{k-1})\|x_k - y_k\|^2. \tag{26}
\end{aligned}
$$

In turn, bounds on (26) translate into bounds on $V_{k+1}(y_k, x_k) - V_k(y_k, x_k)$, since

$$
V_{k+1}(y_k, x_k) - V_k(y_k, x_k) = \frac{m}{2}(\|x_k - x^* + \sqrt{Q_k}(x_k - y_k)\|^2 - \|z_k - x^*\|^2). \tag{27}
$$

Then, to prove the inductive step, we express bounds on (27) in terms of $V_{k+1}$ and $V_k$.

First, suppose $\langle x_k - y_k, x_k - x^* \rangle \ge 0$. Also assuming $L_k \ge L_{k-1}$, then $\sqrt{Q_{k-1}}/\sqrt{Q_k} \le 1$, so that

$$
\sqrt{Q_k} - \sqrt{Q_{k-1}} \le \frac{Q_k}{\sqrt{Q_k}} - \sqrt{Q_{k-1}}\frac{\sqrt{Q_{k-1}}}{\sqrt{Q_k}} = \frac{Q_k - Q_{k-1}}{\sqrt{Q_k}}.
$$

Hence, applying the above inequality to (26) and then adding a nonnegative $\|x_k - x^*\|^2$ term to it, we get

$$
\begin{aligned}
&\|x_k - x^* + \sqrt{Q_k}(x_k - y_k)\|^2 - \|z_k - x^*\|^2 \\
&\leq 2\frac{Q_k - Q_{k-1}}{\sqrt{Q_k}}\langle x_k - x^*, x_k - y_k\rangle + (Q_k - Q_{k-1})\|x_k - y_k\|^2 + \frac{Q_k - Q_{k-1}}{Q_k}\|x_k - x^*\|^2 \\
&= \frac{Q_k - Q_{k-1}}{Q_k}\|x_k - x^* + \sqrt{Q_k}(x_k - y_k)\|^2.
\end{aligned}
\tag{28}
$$

Plugging (28) back into (27) yields

$$
\begin{aligned}
V_{k+1}(y_k, x_k) - V_k(y_k, x_k) &\leq \frac{Q_k - Q_{k-1}}{Q_k}\frac{m}{2}\|x_k - x^* + \sqrt{Q_k}(x_k - y_k)\|^2 \\
&\leq \frac{Q_k - Q_{k-1}}{Q_k}V_{k+1}(y_k, x_k),
\end{aligned}
\tag{29}
$$

where the last inequality follows from the definition of $V_k$, as $f(y_k) - f(x^*) \geq 0$ implies

$$
V_{k+1}(y_k, x_k) \geq \frac{m}{2}\|x_k - x^* + \sqrt{Q_k}(x_k - y_k)\|^2.
\tag{30}
$$

Thus, rearranging terms in (29) and then multiplying both sides by $Q_k/Q_{k-1}$, we obtain

$$
V_{k+1}(y_k, x_k) \leq \frac{Q_k}{Q_{k-1}}V_k(y_k, x_k) \leq \frac{Q_k^2}{Q_{k-1}^2}V_k(y_k, x_k),
$$

where the second inequality holds because $Q_k/Q_{k-1} \geq 1$.

Now, suppose $\langle x_k - y_k, x_k - x^*\rangle < 0$. As in the previous case, we start by bounding the gap (26). But given the negative sign of $\langle x_k - y_k, x_k - x^*\rangle$ term, we bound the $\|x_k - y_k\|^2$ term instead. To this end, we first invoke the assumption that $\langle x_k - y_k, x_k - x^*\rangle < 0$ to establish that

$$
\begin{aligned}
\|y_k - x^*\|^2 &= \|x_k - x^* - (x_k - y_k)\|^2 \\
&= \|x_k - x^*\|^2 - 2\langle x_k - x^*, x_k - y_k\rangle + \|x_k - y_k\|^2 \\
&\geq \|x_k - x^*\|^2.
\end{aligned}
\tag{31}
$$

To use the above inequality on (26), first we rewrite it more conveniently as

$$
\begin{aligned}
&\|x_k - x^* + \sqrt{Q_k}x_k - y_k\|^2 - \|z_k - x^*\|^2 \\
=&2\frac{\sqrt{Q_k} - \sqrt{Q_{k-1}}}{\sqrt{Q_k}}\langle x_k - x^*, \sqrt{Q_k}(x_k - y_k)\rangle + \sqrt{Q_k}(\sqrt{Q_k} - \sqrt{Q_{k-1}})\|x_k - y_k\|^2 \\
&+ \sqrt{Q_{k-1}}(\sqrt{Q_k} - \sqrt{Q_{k-1}})\|x_k - y_k\|^2 \pm \frac{\sqrt{Q_k} - \sqrt{Q_{k-1}}}{\sqrt{Q_k}}\|x_k - x^*\|^2 \\
=&\frac{\sqrt{Q_k} - \sqrt{Q_{k-1}}}{\sqrt{Q_k}}\|x_k - x^* + \sqrt{Q_k}(x_k - y_k)\|^2 + \sqrt{Q_{k-1}}(\sqrt{Q_k} - \sqrt{Q_{k-1}})\|x_k - y_k\|^2 \\
&- \frac{\sqrt{Q_k} - \sqrt{Q_{k-1}}}{\sqrt{Q_k}}\|x_k - x^*\|^2.
\end{aligned}
\tag{32}
$$

Next, we use the following elementary inequality, which is a consequence of $\|a/c + bc\|^2 \geq 0$:

$$
\|a - b\|^2 = \|a\|^2 - 2\langle a, b\rangle + \|b\|^2 \leq (1 + 1/c^2)\|a\|^2 + (1 + c^2)\|b\|^2.
$$

Namely, we apply the above inequality with $a = z_k - x^*$, $b = x_k - x^*$ and $c^2 = \sqrt{Q_{k-1}}/\sqrt{Q_k}$ to bound the $\|x_k - y_k\|^2$ term on (32) and obtain

$$
\begin{aligned}
&\sqrt{Q_{k-1}}(\sqrt{Q_k} - \sqrt{Q_{k-1}})\|x_k - y_k\|^2 \\
&= \sqrt{Q_{k-1}}(\sqrt{Q_k} - \sqrt{Q_{k-1}})\|x_k - y_k \pm (x_k - x^*)/\sqrt{Q_{k-1}}\|^2 \\
&= \frac{\sqrt{Q_k} - \sqrt{Q_{k-1}}}{\sqrt{Q_{k-1}}}\|z_k - x^* - (x_k - x^*)\|^2 \\
&\leq \frac{\sqrt{Q_k} - \sqrt{Q_{k-1}}}{\sqrt{Q_{k-1}}}\Big(1 + \frac{\sqrt{Q_k}}{\sqrt{Q_{k-1}}}\Big)\|z_k - x^*\|^2 + \frac{\sqrt{Q_k} - \sqrt{Q_{k-1}}}{\sqrt{Q_{k-1}}}\Big(1 + \frac{\sqrt{Q_{k-1}}}{\sqrt{Q_k}}\Big)\|x_k - x^*\|^2 \\
&= \frac{Q_k - Q_{k-1}}{Q_{k-1}}\|z_k - x^*\|^2 + \frac{\sqrt{Q_k} - \sqrt{Q_{k-1}}}{\sqrt{Q_k}}\frac{\sqrt{Q_k} + \sqrt{Q_{k-1}}}{\sqrt{Q_{k-1}}}\|x_k - x^*\|^2.
\end{aligned} \tag{33}
$$

Plugging (33) back into (32) and then using (31), we get

$$
\begin{aligned}
&\|x_k - x^* + \sqrt{Q_k}(x_k - y_k)\|^2 - \|z_k - x^*\|^2 \\
&\leq \frac{\sqrt{Q_k} - \sqrt{Q_{k-1}}}{\sqrt{Q_k}}\|x_k - x^* + \sqrt{Q_k}(x_k - y_k)\|^2 + \frac{Q_k - Q_{k-1}}{Q_{k-1}}\|z_k - x^*\|^2 \\
&\quad + \frac{\sqrt{Q_k} - \sqrt{Q_{k-1}}}{\sqrt{Q_k}}\Big(\frac{\sqrt{Q_k} + \sqrt{Q_{k-1}}}{\sqrt{Q_{k-1}}} - 1\Big)\|x_k - x^*\|^2 \\
&\leq \frac{\sqrt{Q_k} - \sqrt{Q_{k-1}}}{\sqrt{Q_k}}\|x_k - x^* + \sqrt{Q_k}(x_k - y_k)\|^2 + \frac{Q_k - Q_{k-1}}{Q_{k-1}}\|z_k - x^*\|^2 \\
&\quad + \frac{\sqrt{Q_k} - \sqrt{Q_{k-1}}}{\sqrt{Q_{k-1}}}\|y_k - x^*\|^2.
\end{aligned} \tag{34}
$$

In turn, plugging (34) back into (27) and then using the assumptions that $m \geq m$ and $m \leq m$ yields

$$
\begin{aligned}
&V_{k+1}(y_k, x_k) - V_k(y_k, x_k) \\
&\leq \frac{m}{2}\frac{\sqrt{Q_k} - \sqrt{Q_{k-1}}}{\sqrt{Q_k}}\|x_k - x^* + \sqrt{Q_k}(x_k - y_k)\|^2 + \frac{m}{2}\frac{Q_k - Q_{k-1}}{Q_{k-1}}\|z_k - x^*\|^2 \\
&\quad + \frac{m}{2}\frac{\sqrt{Q_k} - \sqrt{Q_{k-1}}}{\sqrt{Q_{k-1}}}\|y_k - x^*\|^2 \\
&\leq \frac{\sqrt{Q_k} - \sqrt{Q_{k-1}}}{\sqrt{Q_k}}\frac{m}{2}\|x_k - x^* + \sqrt{Q_k}(x_k - y_k)\|^2 + \frac{Q_k - Q_{k-1}}{Q_{k-1}}\frac{m}{2}\|z_k - x^*\|^2 \\
&\quad + \frac{\sqrt{Q_k} - \sqrt{Q_{k-1}}}{\sqrt{Q_{k-1}}}\frac{m}{2}\|y_k - x^*\|^2.
\end{aligned} \tag{35}
$$

Now, as in (30), the fact that $f(y_k) - f(x^*) \geq 0$ implies

$$
V_k(y_k, x_k) = f(y_k) - f(x^*) + \frac{m}{2}\|z_k - x^*\|^2 \geq \frac{m}{2}\|z_k - x^*\|^2. \tag{36}
$$

In the same vein, applying (18) with $x = x^*$ and $y = y_k$ to the definition of $V_k$, we obtain

$$
V_k(y_k, x_k) = f(y_k) - f(x^*) + \frac{m}{2}\|z_k - x^*\|^2 \geq \frac{m}{2}\|y_k - x^*\|^2. \tag{37}
$$

Plugging in (30), (36) and (37) back into (35), and then moving all $V_{k+1}^{acc}(y_k, x_k)$ terms to the left-hand side and all $V_k(y_k, x_k)$ to the right-hand side, we obtain

$$
\frac{\sqrt{Q_{k-1}}}{\sqrt{Q_k}}V_{k+1}(y_k, x_k) \leq \Big(\frac{Q_k}{Q_{k-1}} + \frac{\sqrt{Q_k} - \sqrt{Q_{k-1}}}{\sqrt{Q_{k-1}}}\Big)V_k(y_k, x_k) \tag{38}
$$

Multiplying both sides of (38) by $\sqrt{Q_k}/\sqrt{Q_{k-1}}$, and then using the fact that $\sqrt{Q_k} \geq \sqrt{Q_{k-1}}$ yields

$$
V_{k+1}(y_k, x_k) \leq \frac{\sqrt{Q_k}}{\sqrt{Q_{k-1}}}\Big(\frac{Q_k}{Q_{k-1}} + \frac{\sqrt{Q_k} - \sqrt{Q_{k-1}}}{\sqrt{Q_{k-1}}}\Big)V_k(y_k, x_k) \leq \frac{Q_k^2}{Q_{k-1}^2}V_k(y_k, x_k),
$$

where the last inequality above holds because $Q_k \geq Q_{k-1}$ implies the following equivalences hold:

$$\frac{Q_k}{Q_{k-1}} + \frac{\sqrt{Q_k} - \sqrt{Q_{k-1}}}{\sqrt{Q_{k-1}}} \leq \frac{Q_k^{3/2}}{Q_{k-1}^{3/2}} \iff \sqrt{Q_{k-1}}Q_k + Q_{k-1}(\sqrt{Q_k} - \sqrt{Q_{k-1}}) \leq Q_k^{3/2},$$

$$\iff Q_{k-1}(\sqrt{Q_k} - \sqrt{Q_{k-1}}) \leq Q_k(\sqrt{Q_k} - \sqrt{Q_{k-1}}).$$

Therefore, both when $\langle x_k - x^*, x_k - y_k \rangle \geq 0$ and when $\langle x_k - x^*, x_k - y_k \rangle < 0$, the inequality

$$V_{k+1}(y_k, x_k) \leq \frac{Q_k^2}{Q_{k-1}^2} V_k(y_k, x_k)$$

holds generically for all $y_k, x_k$, proving the lemma. $\qquad\square$

**Proposition 15.** *Let $f$ be Lipschitz-smooth strongly convex. Given initial points $x_0 = y_0$, if the estimates $L_k$ of the Lipschitz constant in accelerated gradient descent (Algorithm 4) are generated monotonically by adaptive backtracking (Algorithm 2 with $L_k$ serving as the initial estimate for $L_{k+1}$) using $\hat{\rho}$ and $\upsilon$ given by (4) with $c \in [1/2, 1)$ and $L_0 > m$, then for $k \geq 0$*

$$f(y_{k+1}) - f(x^*) \leq \left( \frac{\sqrt{Q} - \sqrt{2(1-c)\rho}}{\sqrt{Q}} \right)^k \frac{Q^2}{4(1-c)^2\rho^2} \frac{L+m}{2} \|x_0 - x^*\|^2.$$

*Proof.* Combining Lemmas 1 and 2, we have that for every $k \geq 0$

$$V_{k+1}(y_{k+1}, x_{k+1}) \leq \frac{1}{1 + \delta_k} V_{k+1}(y_k, x_k) \leq \frac{1}{1 + \delta_k} \frac{Q_k^2}{Q_{k-1}^2} V_k(y_k, x_k).$$

Moreover, from Proposition 9 and the assumption that $L_0 > m$, it follows that

$$L_k \leq \max\{L_0, L/(2(1-c)\rho)\} \leq \max\{m, L/(2(1-c)\rho)\}$$

and, in turn, we obtain

$$\frac{1}{1 + \delta_k} \leq \frac{\sqrt{Q_k} - 1}{\sqrt{Q_k}} \leq \frac{\sqrt{Q} - \sqrt{2(1-c)\rho}}{\sqrt{Q}} \qquad \text{where} \qquad Q = \frac{L}{m}. \qquad (39)$$

Furthermore, assuming $y_0 = x_0$, we have that

$$V_0(y_0, x_0) = f(y_0) - f(x^*) + \frac{m}{2} \|z_0 - x^*\|^2 \leq \frac{L+m}{2} \|x_0 - x^*\|^2.$$

Arguing inductively, all but $Q_k^2$ and $Q_{-1}^2 = Q_0^2$ cancel and, since $L_0 > m$, we get

$$f(y_{k+1}) - f(x^*) \leq V_{k+1}(y_{k+1}, x_{k+1})$$

$$\leq \left( \frac{\sqrt{Q} - \sqrt{2(1-c)\rho}}{\sqrt{Q}} \right)^k \frac{L_k^2}{L_0^2} V_0(y_0, x_0)$$

$$\leq \left( \frac{\sqrt{Q} - \sqrt{2(1-c)\rho}}{\sqrt{Q}} \right)^k \frac{Q^2}{4(1-c)^2\rho^2} \frac{L+m}{2} \|x_0 - x^*\|^2.$$

$\qquad\square$

## C    METHODS

In this subsection, we briefly state standard implementations of the base methods that we use in the paper. For the sake of simplicity, we only state a single iteration of the corresponding method.

Algorithms 3 and 4 summarize gradient descent and Nesterov's accelerated gradient descent in the formulation with constant momentum coefficient (Nesterov, 2018, 2.2.22). Algorithm 5 summarizes Adagrad (Duchi et al., 2011).

To state the last base method that we consider in this paper, we must introduce an auxiliary operator. Given a convex Lipschitz-smooth function $f$ with Lipschitz constant $L$ and a continuous convex function, the proximal operator $p_L$ is defined by

$$p_L(y) := \arg\min_x \left\{ g(x) + \frac{L}{2} \left\| x - \left( y - \frac{1}{L} \nabla f(y) \right) \right\|^2 \right\}. \tag{40}$$

With this definition, we can state Algorithm 6, which summarizes FISTA (Beck & Teboulle, 2009).

---

**Algorithm 3** Gradient Descent.

---

**Input:** $x_k, \nabla f(x_k), \alpha_k > 0$
**Output:** $x_{k+1}$
  1: $x_{k+1} \leftarrow x_k - \alpha_k \nabla f(x_k)$

---

**Algorithm 4** Nesterov's accelerated gradient descent (Nesterov, 2018, 2.2.22).

---

**Input:** $x_k, y_k, \nabla f(x_k), L_k > m > 0$
**Output:** $x_{k+1}, y_{k+1}$
  1: $y_{k+1} \leftarrow x_k - (1/L_k) \nabla f(x_k)$
  2: $\beta_k \leftarrow \frac{\sqrt{L_k} - \sqrt{m}}{\sqrt{L_k} + \sqrt{m}}$
  3: $x_{k+1} \leftarrow (1 + \beta_k) y_{k+1} - \beta_k y_k$

---

**Algorithm 5** Adagrad (Duchi et al., 2011). Superscript $i$ means the $i$-th entry of the vector.

---

**Input:** $x_k, \nabla f(x_k), y_k, x_k \alpha_k > 0$
**Output:** $x_{k+1}, s_{k+1}$
  1: $s_{k+1}^i = y_k, x_k^i + (\nabla f(x_k)^i)^2$
  2: $x_{k+1}^i \leftarrow x_k^i - \frac{\alpha_k}{\sqrt{s_{k+1}^i}} \nabla f(x_k^i)$

---

**Algorithm 6** FISTA (Beck & Teboulle, 2009).

---

**Input:** $x_k, x_{k-1}, y_k, t_k, \nabla f(x_k)$
**Output:** $x_{k+1}, y_{k+1}, t_{k+1}$
  1: $x_{k+1} \leftarrow p_L(y_k)$
  2: $t_{k+1} \leftarrow \frac{1 + \sqrt{1 + 4t_k^2}}{2}$
  3: $y_{k+1} \leftarrow x_k + \frac{t_k - 1}{t_{k+1}} (x_k - x_{k-1})$

---

## D LOGISTIC REGRESSION EXPERIMENTS

In this section, we provide further details of the logistic regression experiments and present full plots of all runs.

Table 4: Details of datasets and method precisions used in the logistic regression problem.

| dataset | datapoints | dimensions | AGD | GD | GD (monotone) | Adagrad |
|---|---|---|---|---|---|---|
| a9a | 32561 | 123 | $10^{-9}$ | $10^{-6}$ | $10^{-5}$ | $10^{-6}$ |
| gisette_scale | 6000 | 5000 | $10^{-9}$ | $10^{-9}$ | $10^{-5}$ | $10^{-9}$ |
| MNIST | 60000 | 784 | $10^{-9}$ | $10^{-6}$ | $10^{-3}$ | $10^{-9}$ |
| mushrooms | 8124 | 112 | $10^{-9}$ | $10^{-9}$ | $10^{-5}$ | $10^{-9}$ |
| phishing | 11055 | 68 | $10^{-9}$ | $10^{-9}$ | $10^{-6}$ | $10^{-6}$ |
| protein | 102025 | 75 | $10^{-9}$ | $10^{-9}$ | $10^{-5}$ | $10^{-9}$ |
| web-1 | 2477 | 300 | $10^{-9}$ | $10^{-9}$ | $10^{-8}$ | $10^{-9}$ |

**Dataset details.** We take observation from seven datasets: A9A, GISETTE_SCALE (G_SCALE), MUSHROOMS, PHISHING and WEB-1 from LIBSVM (Chang & Lin, 2011), PROTEIN from KDD Cup 2004 (Caruana et al., 2004) and MNIST (LeCun et al., 1998) The dataset A9A is a preprocessed version of the ADULT dataset (Becker & R, 1996), while WEB-1 is subsample of the WEB dataset (Platt, 1998).

**Initialization details.** For Lipschitz-smooth problems, a step-size of $1/\bar{L}$ is guaranteed to satisfy the Armijo condition (with $c = 1/2$) if $\bar{L} \geq L$. Accordingly, we consider four choices of initial step-sizes, $\alpha = \{10^1, 10^2, 10^3, 10^4\}/\bar{L}$, which capture the transition from initial step-sizes that do not require adjustments to satisfy the Armijo condition to step-sizes that do. In practice, $L$ is unknown and the transition would occur as one attempted an arbitrary initial step-size and adjusted it correspondingly until the line search was activated. Hence, using $\bar{L}$ to anchor the choice of initial step-sizes is merely an educated guess of the transition values that would be found in practice. We adopt the standard choice $c = 10^{-4}$ (Nocedal & Wright, 2006, p. 62) in (3) for BLS used with GD and Adagrad but, motivated by both theory and practice, we choose $c = 1/2$ in the case of AGD. Also, we use the regularization parameter $\gamma$ as the strong convexity parameter input for AGD.

**Evaluation details.** We run all base method and their variants for long enough to produce solutions with designated precision; Then, we account for the number of gradient and function evaluations and elapsed time each variant takes to produce that solution. Finally, for each BLS variant we average those numbers over the four initial step-sizes that we considered. All methods compute exactly one gradient per iteration. To account for elapsed time, we record wall clock time after every iteration. Although somewhat imprecise, elapsed time reflects the relative computational cost of gradient and function evaluations and, especially in larger problems, is a reasonable metric to compare performance.

**Additional comments.** We make the following additional remarks and observations:

- We considered two ways to initialize the step-size for line search at each iteration: (1) using the step-size from the previous iteration and (2) using the same fixed step-size at every iteration. We refer to the corresponding line search subroutines as *monotone* and *memoryless*. The monotone variants are robust to every choice of $\rho$ while some values of $\rho$ may turn the memoryless variants of AGD unstable or unacceptably slow. When the memoryless variants work, however, they generally work much better than the corresponding monotone variants and the baseline methods.

- Monotone backtracking is not as appealing as memoryless backtracking because although both variants take fewer iterations than the baseline method does to reach a given precision, the savings in iterations generated by the monotone variants are not enough to outweigh the additional computational cost of function evaluations that the same variants accrue. Therefore, we only report results for the memoryless variant in the main text and defer results for the monotone variant to Appendix D.1.

- The initial step-sizes greatly impact performance. For some starting step-sizes, vanilla backtracking is better suited for finding the optimal solution than our adaptive method. However, we find that there tends to be more variance in the performance of vanilla backtracking.

- When $\bar{L}$ is a good estimate of the true Lipschitz constant, the computational cost of function evaluations may outweigh the savings in gradient evaluation and even memoryless backtracking might

not improve on the baseline method. This is the case for the COVTYPE dataset from LIBSVM (Chang & Lin, 2011), as shown by Fig. 19b, in Appendix D.2.

- The corresponding stable values of $\rho$ for the adaptive counterpart of AGD lie in the upper interval $(0.7, 1)$ and usually greater values of $\rho$ make the adaptive variant more stable but also more computationally expensive. AGD with regular memoryless backtracking fails to consistently converge for values of $\rho$ outside the interval $(0.3, 0.5)$. In fact, on COVTYPE, for at least one of the initial step-sizes, AGD with regular memoryless backtracking line search fails to converge. On the other hand, as shown in Fig. 19b in Appendix D.2, the adaptive variant converges for $\rho = 0.9$ and even for $\rho = 0.7$, the more unstable end of feasible $\rho$ values.

### D.1 MONOTONE BACKTRACKING LINE SEARCH

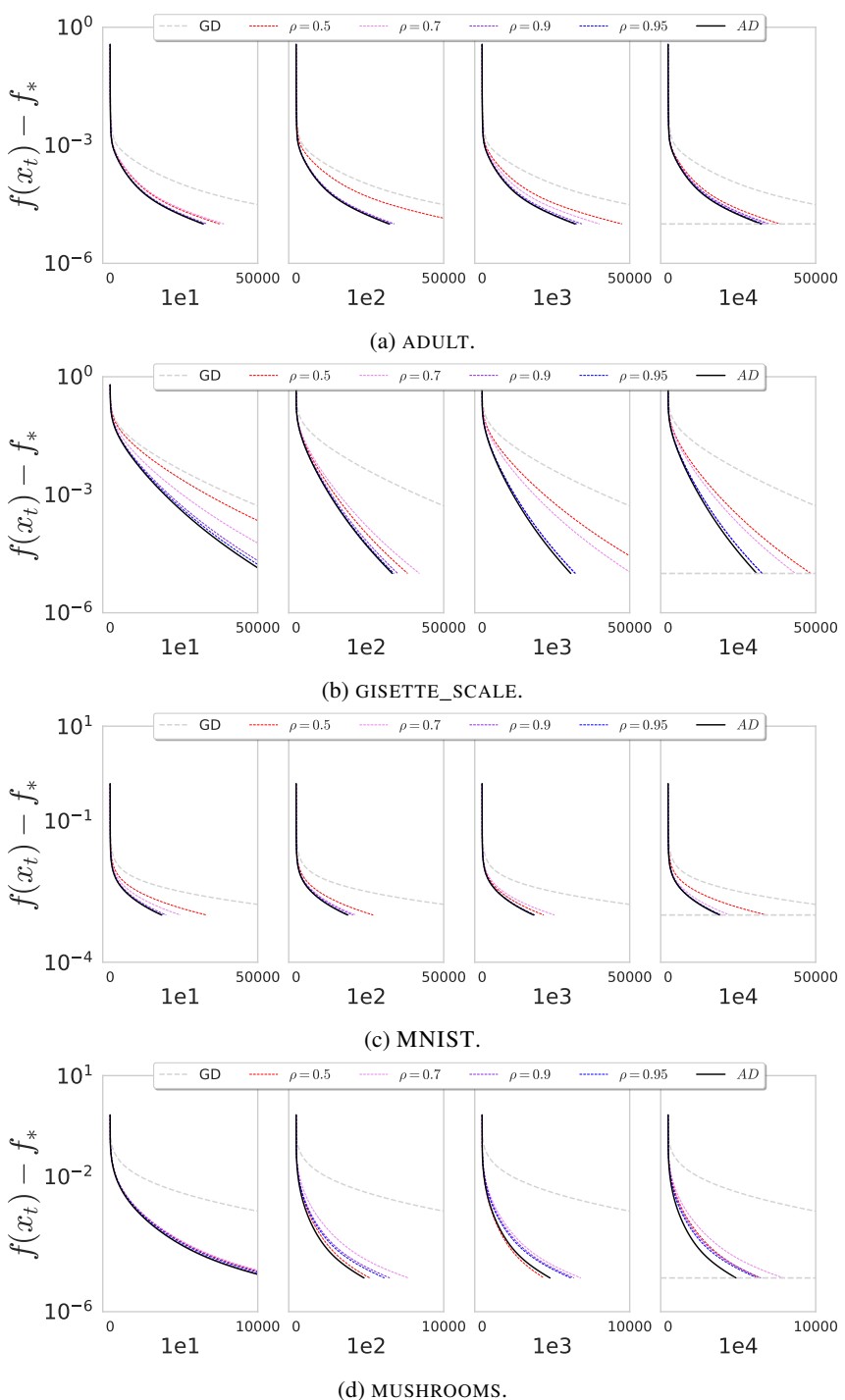

Figure 8: Logistic regression on four different datasets and four initial step-sizes $\alpha_0 = \{10^1, 10^2, 10^3, 10^4\}/\bar{L}$: suboptimality gap for GD, GD with standard backtracking line search using $\rho \in \{0.5, 0.7, 0.9, 0.95\}$ and GD with adaptive memoryless backtracking line search using $\rho = 0.9$. The light gray horizontal dashed line shows the precision used to compute performance for each dataset.

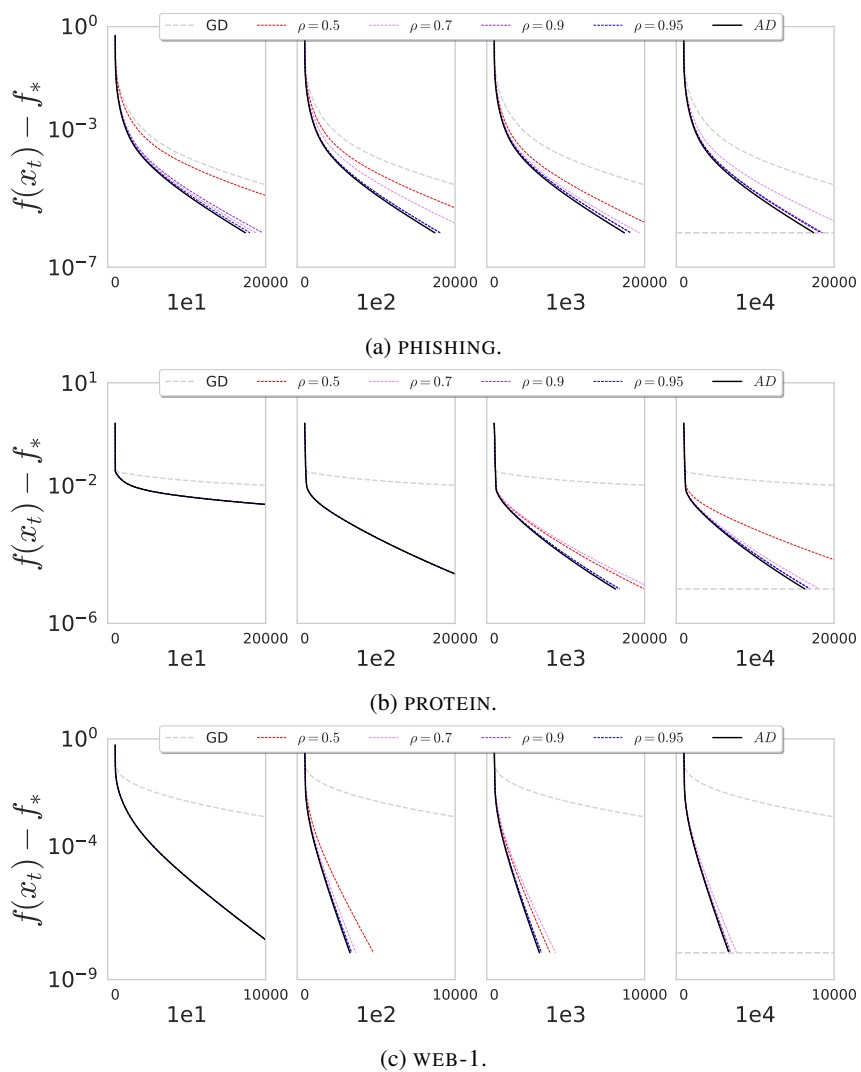

(a) PHISHING.

(b) PROTEIN.

(c) WEB-1.

Figure 9: Logistic regression on four different datasets and four initial step-sizes $\alpha_0 = \{10^1, 10^2, 10^3, 10^4\}/\bar{L}$: suboptimality gap for GD, GD with standard backtracking line search using $\rho \in \{0.5, 0.7, 0.9, 0.95\}$ and GD with adaptive memoryless backtracking line search using $\rho = 0.9$. The light gray horizontal dashed line shows the precision used to compute performance for each dataset.

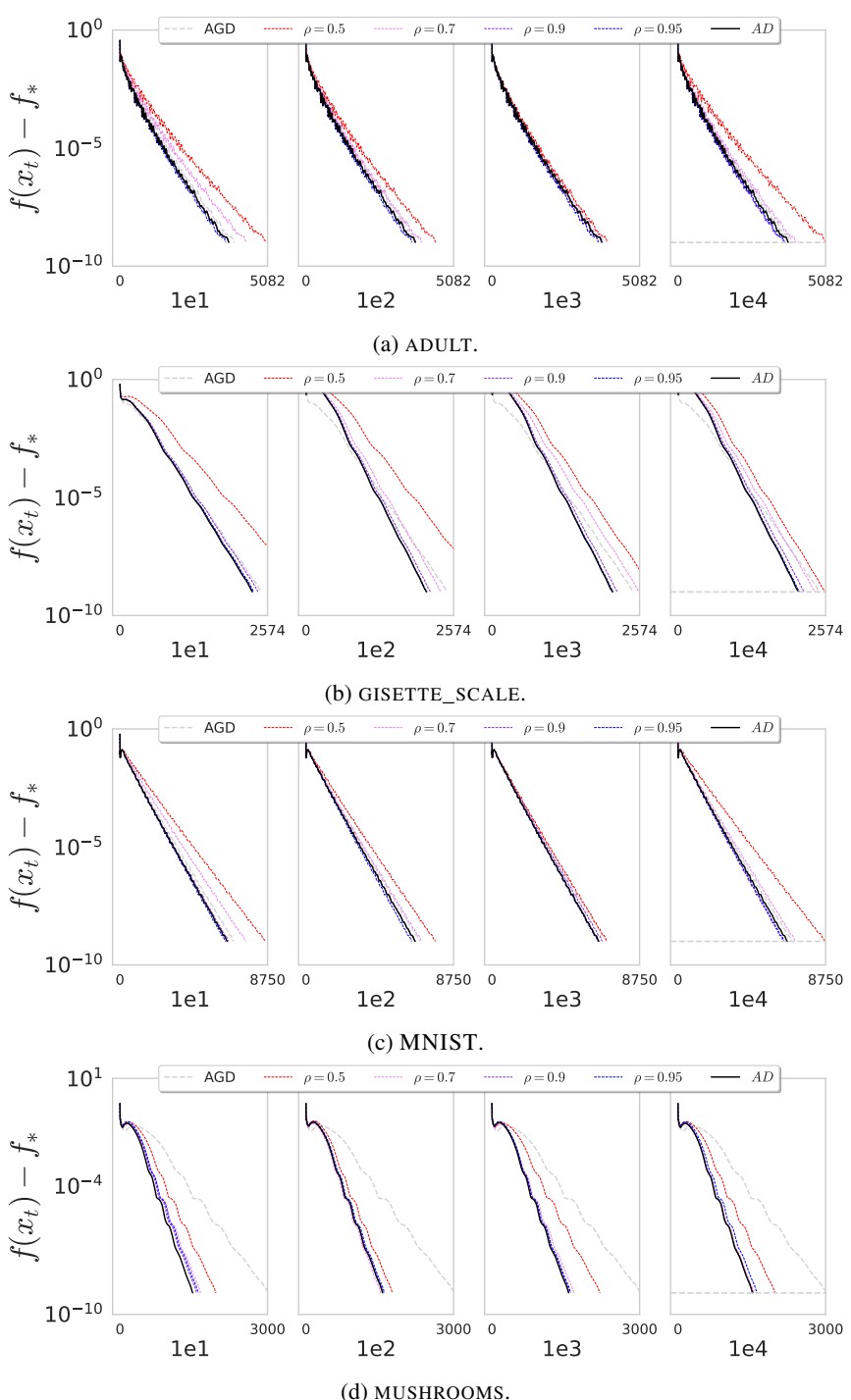

Figure 10: Logistic regression on four different datasets and four initial step-sizes $\alpha_0 = \{10^1, 10^2, 10^3, 10^4\}/\bar{L}$: suboptimality gap for AGD, AGD with standard backtracking line search using $\rho \in \{0.5, 0.7, 0.9, 0.95\}$ and AGD with adaptive memoryless backtracking line search using $\rho = 0.9$. The light gray horizontal dashed line shows the precision used to compute performance for each dataset.

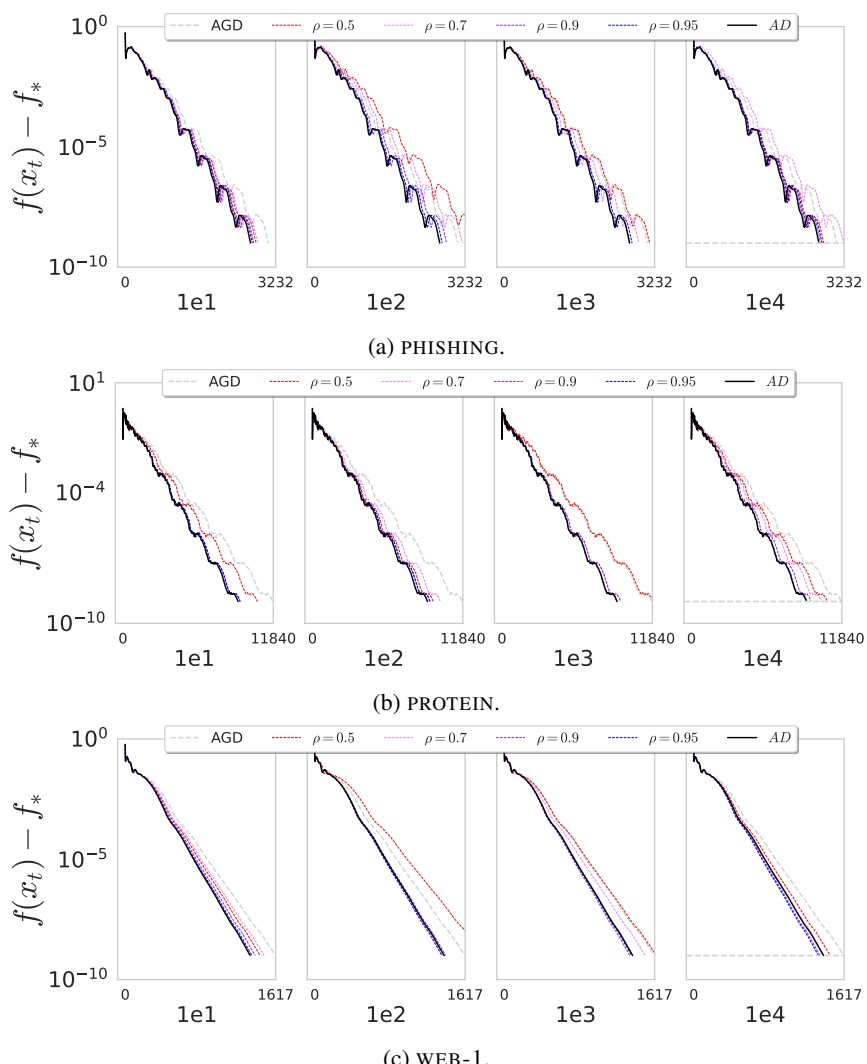

(a) PHISHING.

(b) PROTEIN.

(c) WEB-1.

Figure 11: Logistic regression on four different datasets and four initial step-sizes $\alpha_0 = \{10^1, 10^2, 10^3, 10^4\}/\bar{L}$: suboptimality gap for AGD, AGD with standard backtracking line search using $\rho \in \{0.5, 0.7, 0.9, 0.95\}$ and AGD with adaptive memoryless backtracking line search using $\rho = 0.9$. The light gray horizontal dashed line shows the precision used to compute performance for each dataset.

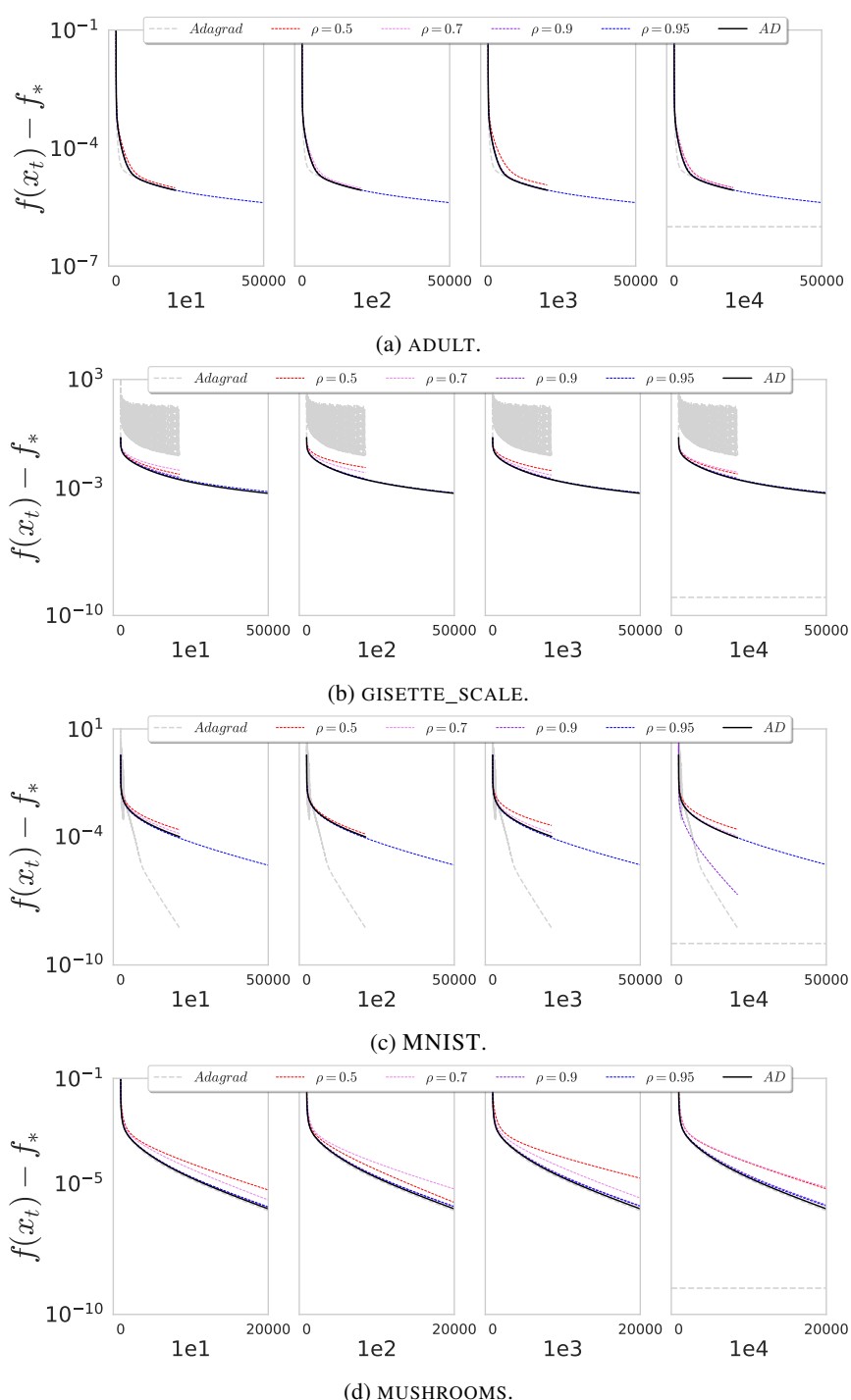

(a) ADULT.

(b) GISETTE_SCALE.

(c) MNIST.

(d) MUSHROOMS.

Figure 12: Logistic regression on four different datasets and four initial step-sizes $\alpha_0 = \{10^1, 10^2, 10^3, 10^4\}/\bar{L}$: suboptimality gap for Adagrad, Adagrad with standard backtracking line search using $\rho \in \{0.5, 0.7, 0.9, 0.95\}$ and Adagrad with adaptive memoryless backtracking line search using $\rho = 0.9$. The light gray horizontal dashed line shows the precision used to compute performance for each dataset.

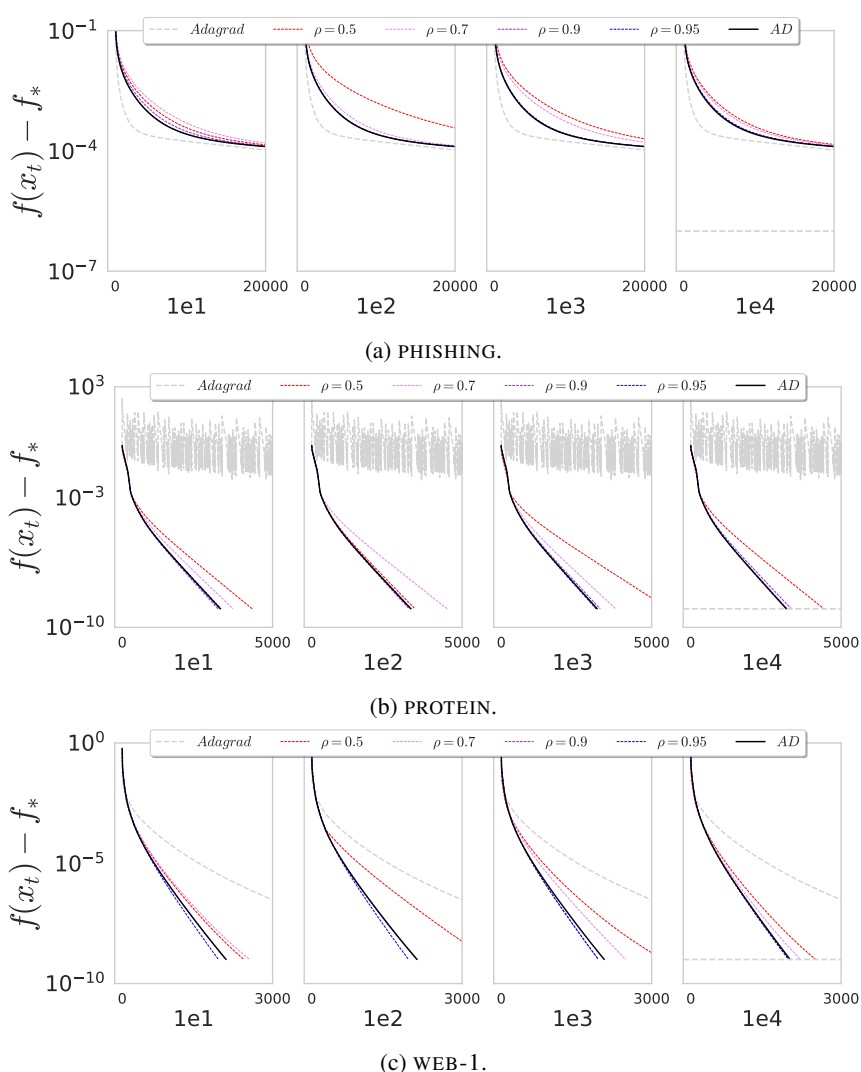

(a) PHISHING.

(b) PROTEIN.

(c) WEB-1.

Figure 13: Logistic regression on four different datasets and four initial step-sizes $\alpha_0 = \{10^1, 10^2, 10^3, 10^4\}/\bar{L}$: suboptimality gap for Adagrad, Adagrad with standard backtracking line search using $\rho \in \{0.5, 0.7, 0.9, 0.95\}$ and Adagrad with adaptive memoryless backtracking line search using $\rho = 0.9$. The light gray horizontal dashed line shows the precision used to compute performance for each dataset.

## D.2 MEMORYLESS BACKTRACKING LINE SEARCH

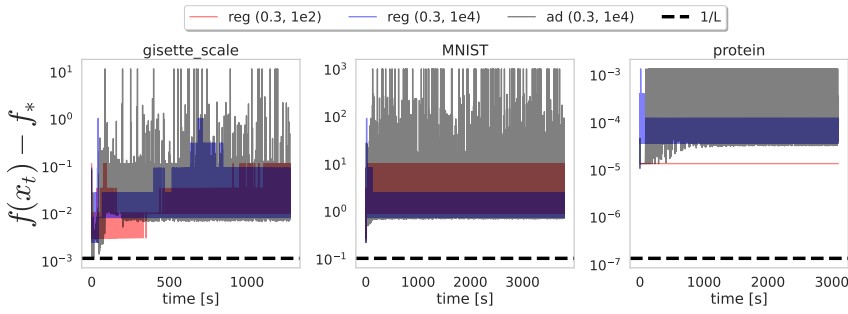

Figure 14: step-sizes for experiments shown in Fig. 1. Baseline: GD with constant $\alpha_k = 1/\bar{L}$; reg $(\rho, \beta)$ and ad $(\rho, \beta)$: GD with, respectively, regular and adaptive memoryless BLS parameterized by $\rho$ and $\alpha_0 = \beta/\bar{L}$. The thick black dashed line denotes $1/\bar{L}$, where $\bar{L} = \lambda_{\max}(A^\top A)/4n$.

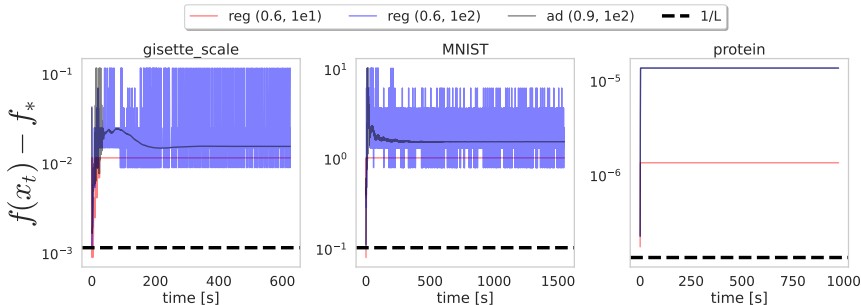

Figure 15: step-sizes for experiments shown in Fig. 2. Baseline: AGD with constant $\alpha_k = 1/\bar{L}$; reg $(\rho, \beta)$ and ad $(\rho, \beta)$: AGD with, respectively, regular and adaptive memoryless BLS parameterized by $\rho$ and $\alpha_0 = \beta/\bar{L}$. The thick black dashed line denotes $1/\bar{L}$, where $\bar{L} = \lambda_{\max}(A^\top A)/4n$.

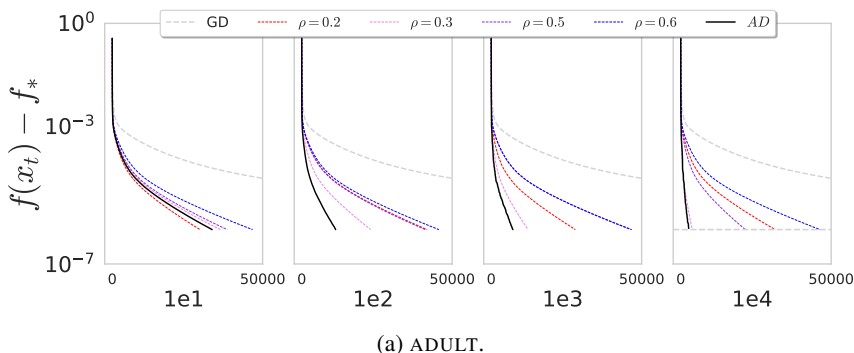

(a) ADULT.

Figure 16: Logistic regression on four different datasets and four initial step-sizes $\alpha_0 = \{10^1, 10^2, 10^3, 10^4\}/\bar{L}$: suboptimality gap for GD, GD with standard memoryless backtracking line search using $\rho \in \{0.2, 0.3, 0.5, 0.6\}$ and GD with adaptive memoryless backtracking line search using $\rho = 0.3$. The light gray horizontal dashed line shows the precision used to compute performance for each dataset.

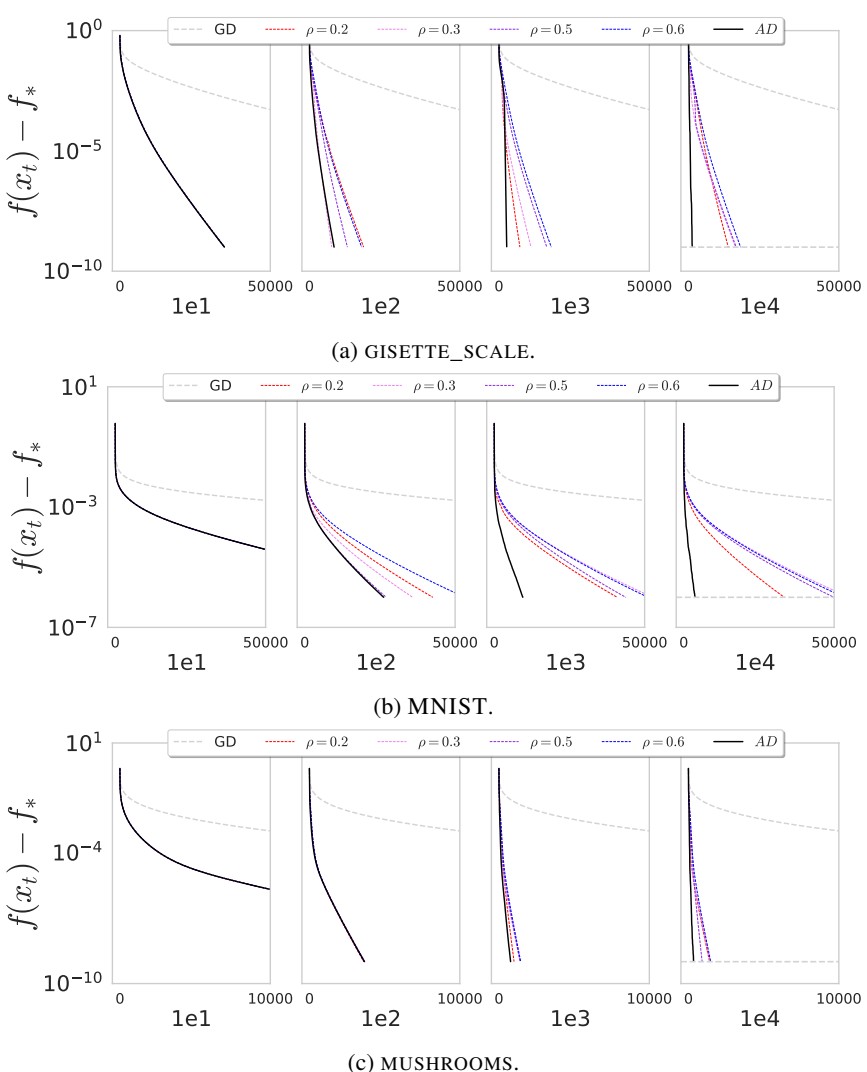

(a) GISETTE_SCALE.

(b) MNIST.

(c) MUSHROOMS.

Figure 17: Logistic regression on four different datasets and four initial step-sizes $\alpha_0 = \{10^1, 10^2, 10^3, 10^4\}/\bar{L}$: suboptimality gap for GD, GD with standard memoryless backtracking line search using $\rho \in \{0.2, 0.3, 0.5, 0.6\}$ and GD with adaptive memoryless backtracking line search using $\rho = 0.3$. The light gray horizontal dashed line shows the precision used to compute performance for each dataset.

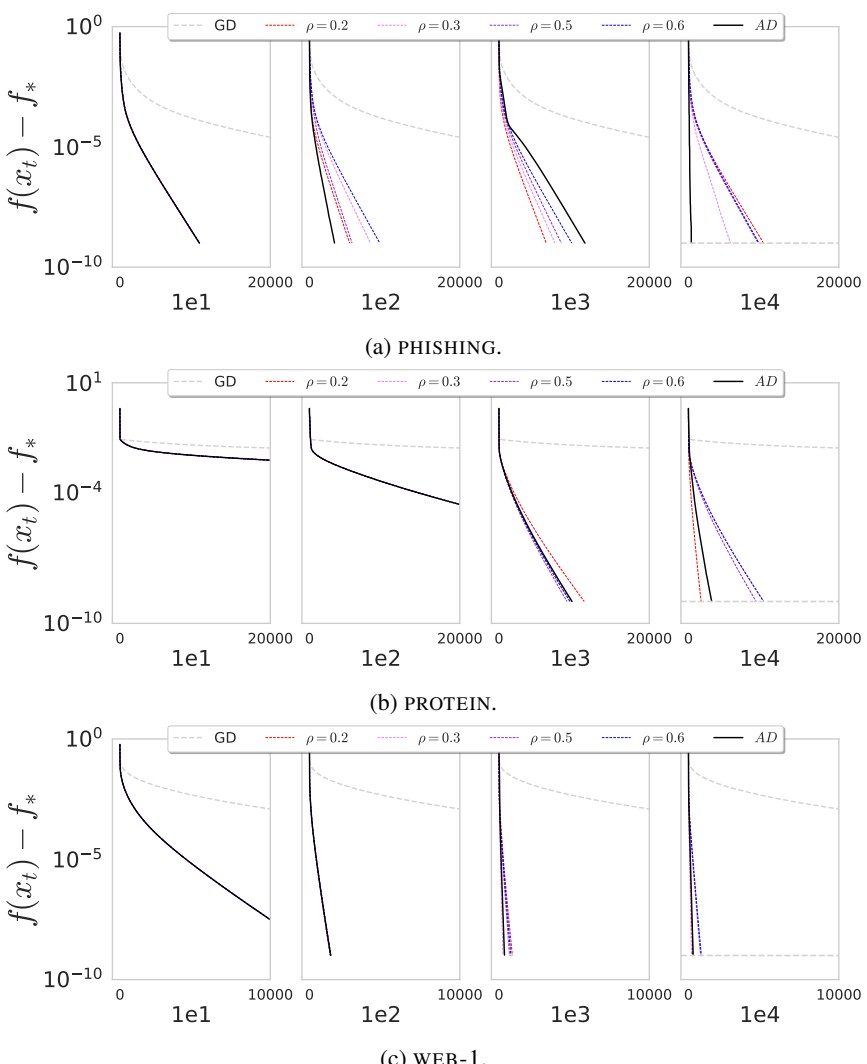

(a) PHISHING.

(b) PROTEIN.

(c) WEB-1.

Figure 18: Logistic regression on four different datasets and four initial step-sizes $\alpha_0 = \{10^1, 10^2, 10^3, 10^4\}/\bar{L}$: suboptimality gap for GD, GD with standard memoryless backtracking line search using $\rho \in \{0.2, 0.3, 0.5, 0.6\}$ and GD with adaptive memoryless backtracking line search using $\rho = 0.3$. The light gray horizontal dashed line shows the precision used to compute performance for each dataset.

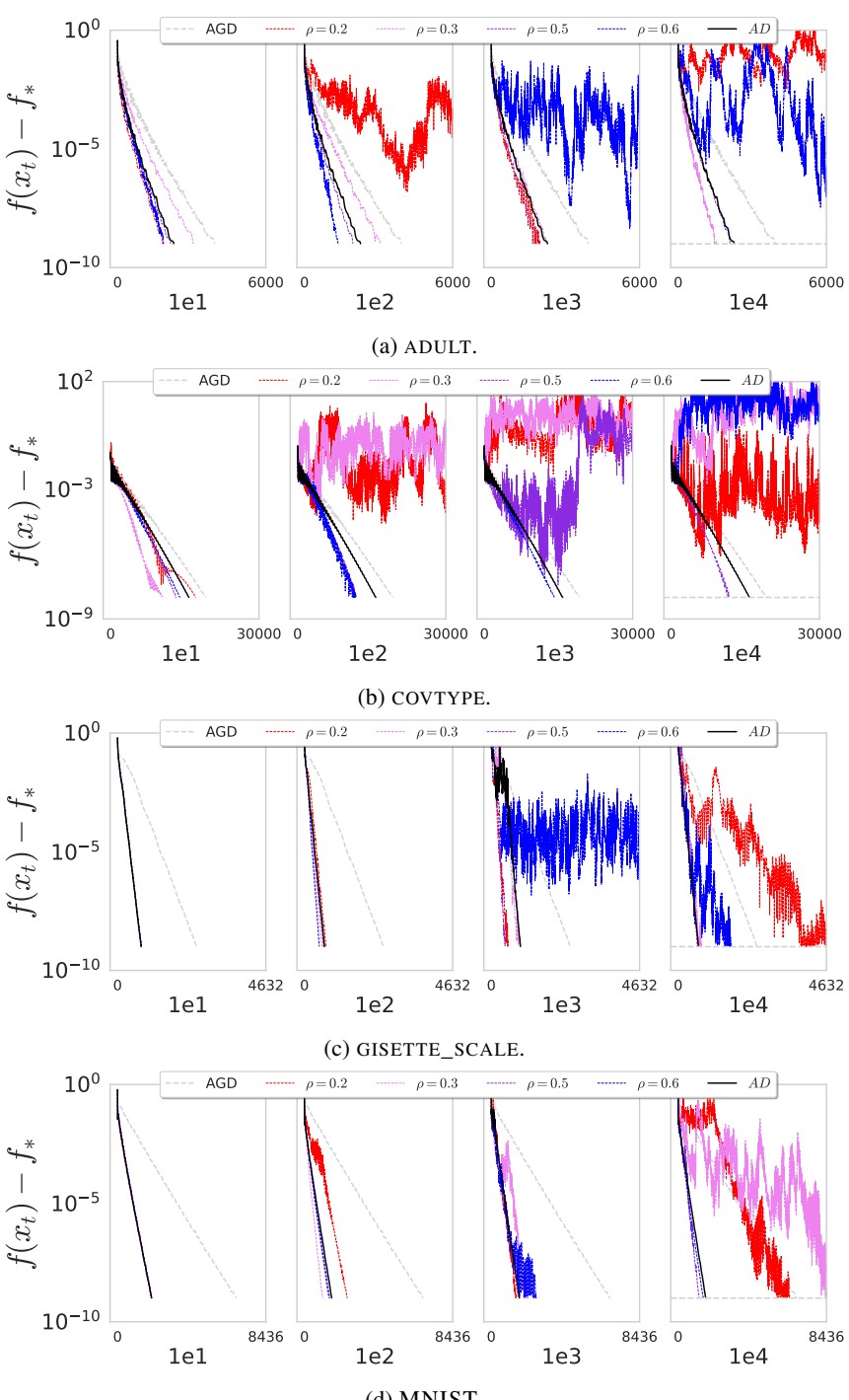

(a) ADULT.

(b) COVTYPE.

(c) GISETTE_SCALE.

(d) MNIST.

Figure 19: Logistic regression on four different datasets and four initial step-sizes $\alpha_0 = \{10^1, 10^2, 10^3, 10^4\}/\bar{L}$: suboptimality gap for AGD, AGD with standard memoryless backtracking line search using $\rho \in \{0.2, 0.3, 0.5, 0.6\}$ and AGD with adaptive memoryless backtracking line search using $\rho = 0.9$. The light gray horizontal dashed line shows the precision used to compute performance for all methods, $10^{-9}$.

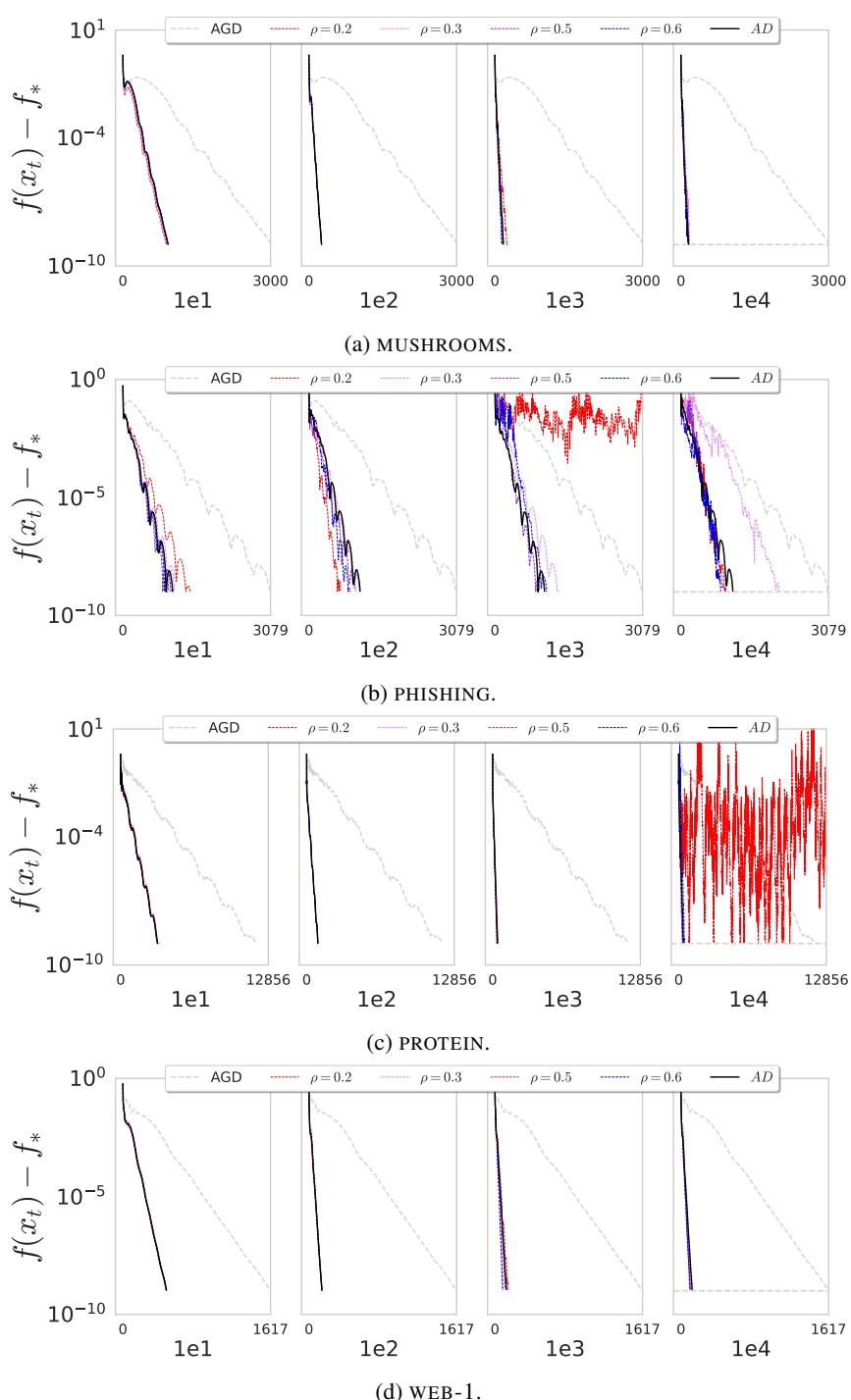

(a) MUSHROOMS.

(b) PHISHING.

(c) PROTEIN.

(d) WEB-1.

Figure 20: Logistic regression on four different datasets and four initial step-sizes $\alpha_0 = \{10^1, 10^2, 10^3, 10^4\}/\bar{L}$: suboptimality gap for AGD, AGD with standard memoryless backtracking line search using $\rho \in \{0.2, 0.3, 0.5, 0.6\}$ and AGD with adaptive memoryless backtracking line search using $\rho = 0.9$. The light gray horizontal dashed line shows the precision used to compute performance for all methods, $10^{-9}$.

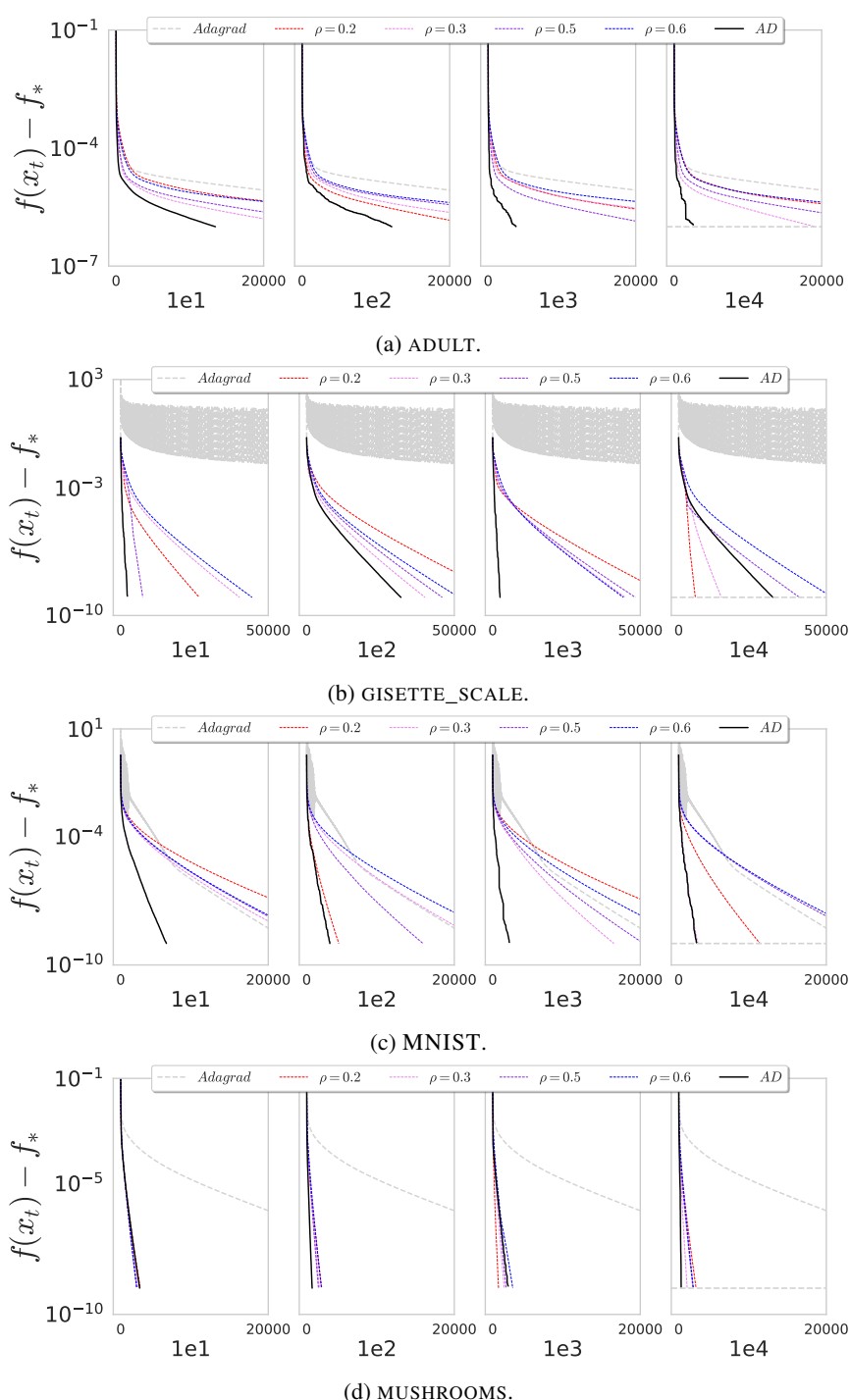

(a) ADULT.

(b) GISETTE_SCALE.

(c) MNIST.

(d) MUSHROOMS.

Figure 21: Logistic regression on four different datasets and four initial step-sizes $\alpha_0 = \{10^1, 10^2, 10^3, 10^4\}/\bar{L}$: suboptimality gap for Adagrad, Adagrad with standard memoryless backtracking line search using $\rho \in \{0.2, 0.3, 0.5, 0.6\}$ and Adagrad with adaptive memoryless backtracking line search using $\rho = 0.3$. The light gray horizontal dashed line shows the precision used to compute performance for each dataset.

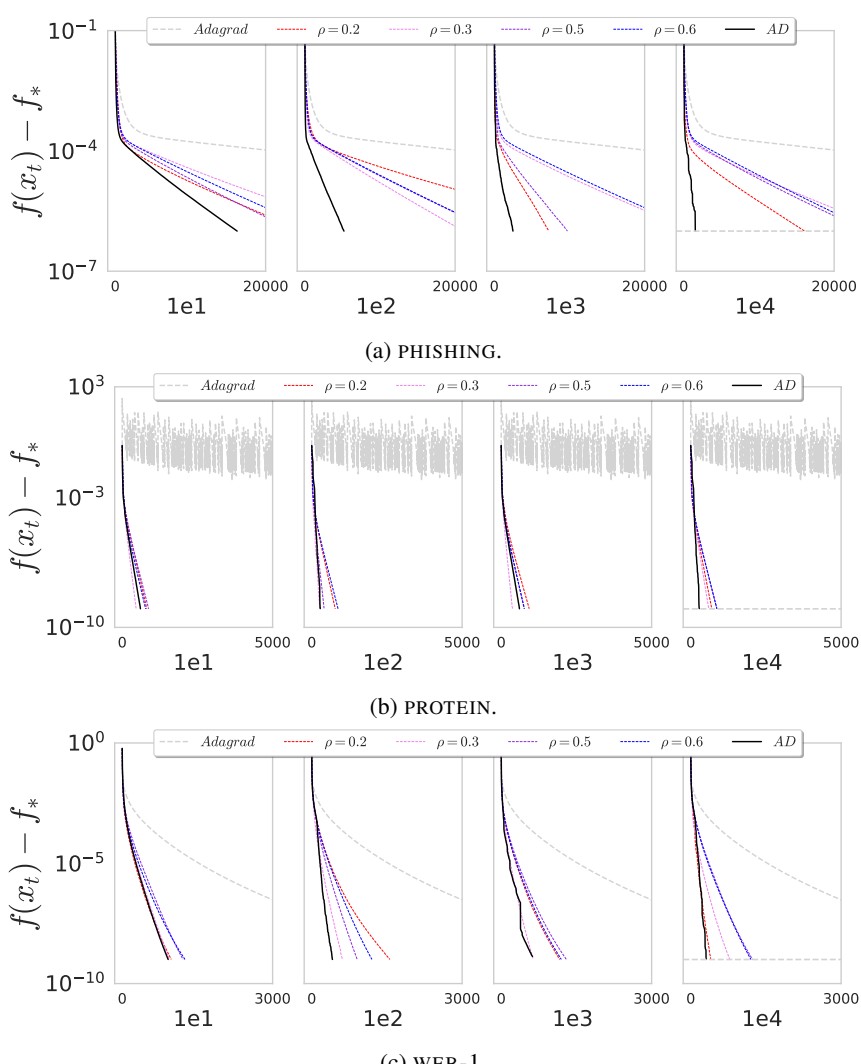

(a) PHISHING.

(b) PROTEIN.

(c) WEB-1.

Figure 22: Logistic regression on four different datasets and four initial step-sizes $\alpha_0 = \{10^1, 10^2, 10^3, 10^4\}/\bar{L}$: suboptimality gap for Adagrad, Adagrad with standard memoryless backtracking line search using $\rho \in \{0.2, 0.3, 0.5, 0.6\}$ and Adagrad with adaptive memoryless backtracking line search using $\rho = 0.3$. The light gray horizontal dashed line shows the precision used to compute performance for each dataset.

# E  LINEAR INVERSE PROBLEMS

**Dataset details.** We consider $A$ observations from eight datasets: IRIS, DIGITS, WINE, OLIVETTI_FACES and LFW_PAIRS from scikit-learn (Pedregosa et al., 2011), SPEAR3 and SPEAR10 (Lorenz et al., 2014) and SPARCO (van den Berg et al., 2007). For multi-class datasets, the first two are considered. The number of datapoints and dimensions of each dataset can be found on Table 5,

Table 5: Details of FISTA experiments.

| dataset | datapoints | dimensions | $\lambda$ | $L_0$ |
|---|---|---|---|---|
| digits | 360 | 64 | $10^{-1}$ | $1, 10^1, 10^2, 10^3$ |
| iris | 100 | 4 | $10^{-2}$ | $10^{-1}, 1, 10^1, 10^2$ |
| lfw_pairs | 2200 | 5828 | 1 | $10^{-3}, 10^{-2}, 10^{-1}, 1$ |
| olivetti_faces | 20 | 4096 | $10^{-2}$ | $1, 10^1, 10^2, 10^3$ |
| Spear3 | 512 | 1024 | $10^{-1}$ | $10^{-3}, 10^{-2}, 10^{-1}, 1$ |
| Spear10 | 512 | 1024 | $10^{-2}$ | $10^{-3}, 10^{-2}, 10^{-1}, 1$ |
| Sparco3 | 1024 | 2048 | $10^{-2}$ | $10^{-3}, 10^{-2}, 10^{-1}, 1$ |
| wine | 130 | 13 | $10^{-2}$ | $1, 10^1, 10^2, 10^3$ |

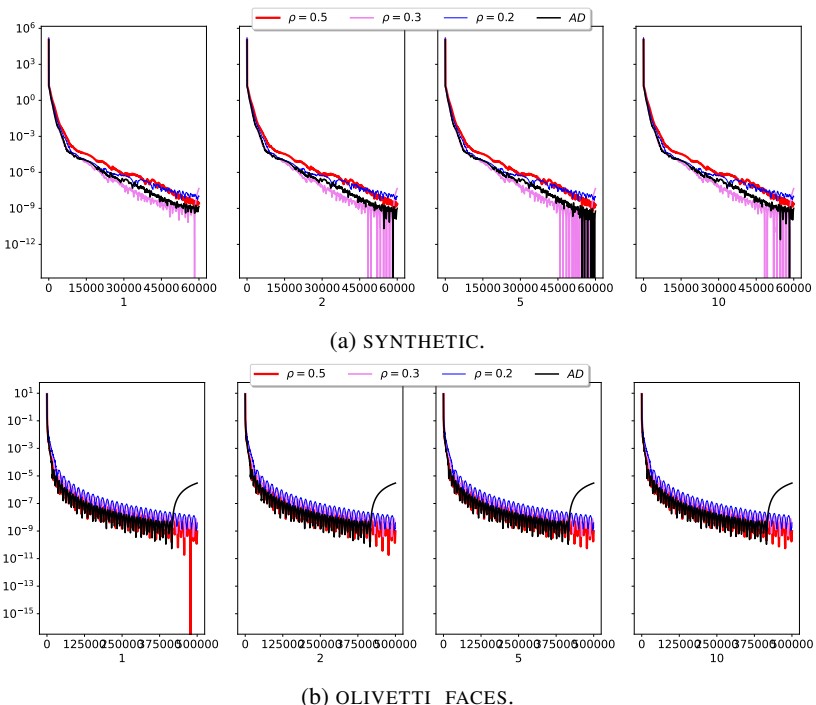

(a) SYNTHETIC.

(b) OLIVETTI_FACES.

Figure 23

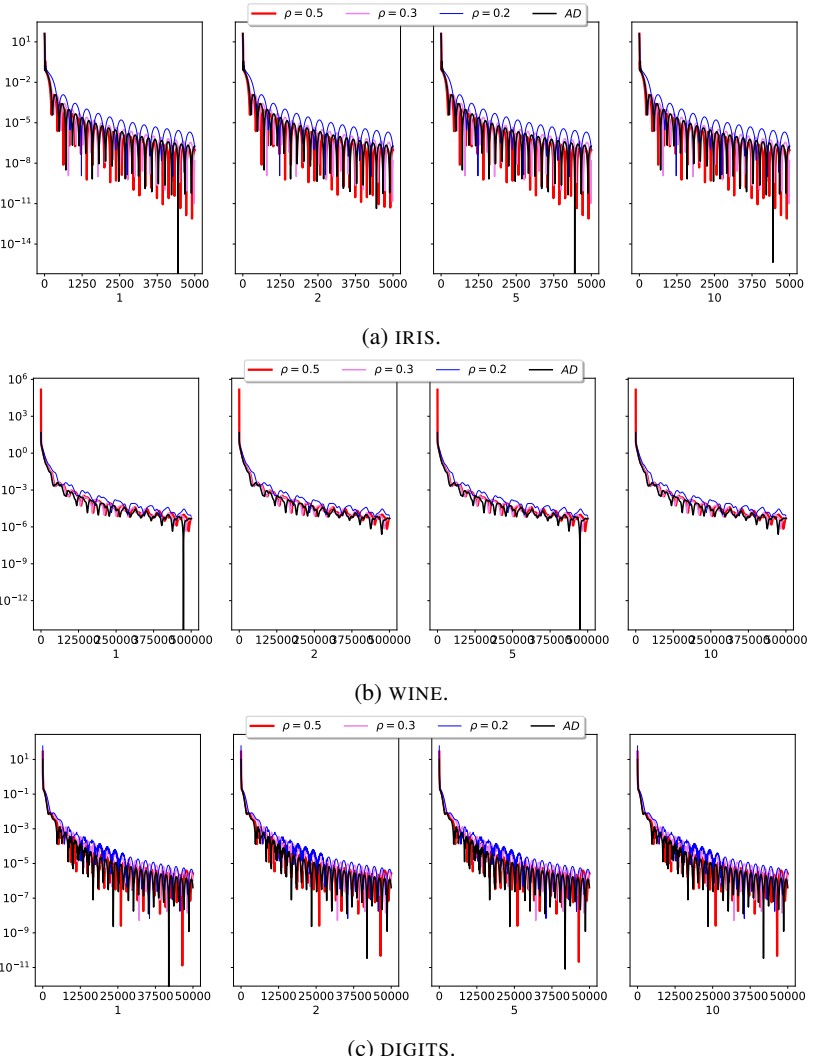

(a) IRIS.

(b) WINE.

(c) DIGITS.

Figure 24

# F  MATRIX FACTORIZATION EXPERIMENTS

We sample $A$ from the file "u.data", part of the MovieLens 100K dataset (grouplens.org/datasets/movielens/100k/). Moreover, we choose the precision representing a reduction to $10^{-12}$ in the suboptimality gap, which corresponds to a lower bound of $10^{-5}$ as the initial objective values typically hover around $10^7$.

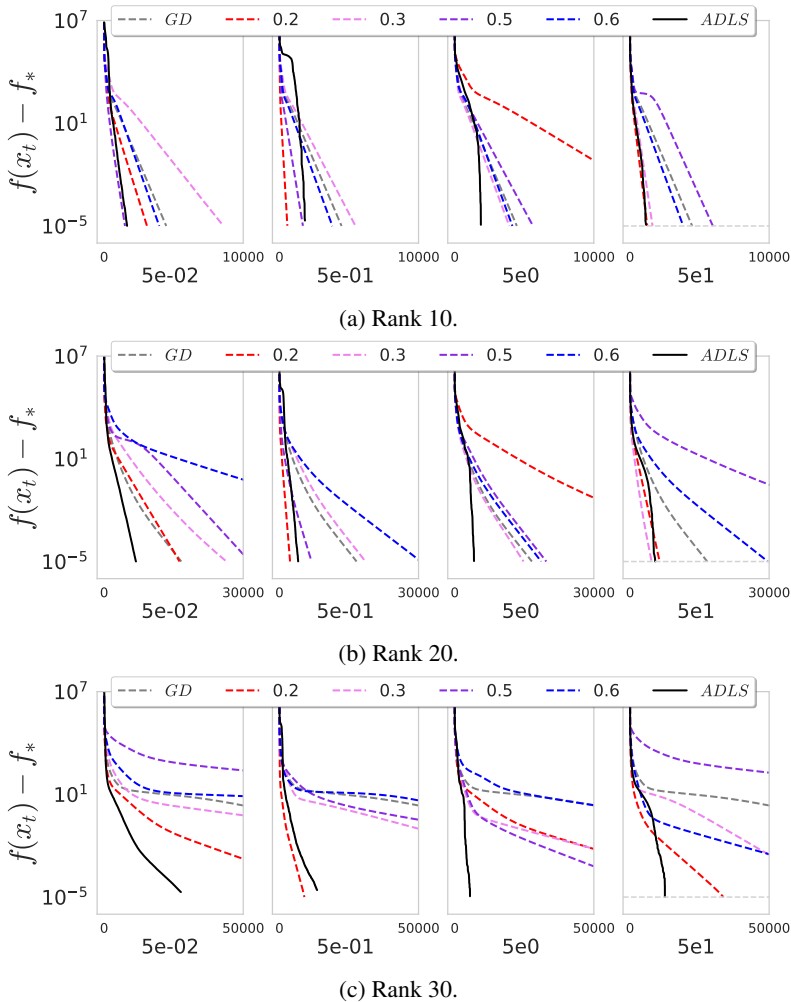

Figure 25: Matrix factorization and three values of rank: suboptimality gap for gradient descent, gradient descent with standard backtracking line search using $\rho \in \{0.2, 0.3, 0.5, 0.6\}$ and gradient descent with adaptive backtracking line search using $\rho = 0.3$. The light gray horizontal dashed line shows the precision used to compute performance for all methods, $10^{-5}$.

