# OpenReview forum: "Adaptive backtracking line search"
_ICLR.cc/2025/Conference — ICLR 2025 Poster_

### Official Review · Reviewer_vFwJ · 2024-10-28

**Soundness:** 4
**Presentation:** 4
**Contribution:** 3
**Rating:** 6
**Confidence:** 3

**Summary:**

The paper proposes an enhancement to backtracking line search by introducing an adaptive adjustment to step size rather than the traditional multiplicative retraction. The basic idea is to not just backtrack when the acceptance criterion is not met but rather to use the violation information to adjust the backtracking factor. The empirical results demonstrate significant improvements in several cases and the authors also provide some theoretical justification for the performance.

**Strengths:**

The empirical results are strong which in my books is also the main merit of the paper. The theory while nice/ok is not particularly spectacular or unexpected but does provide a nice complement to the observed performance in practice.

Still I like the paper quite a bit and set my initial score to accept based on the computational benefits of the proposed approach.

(while not strictly required to be read the appendix provides some additional computational results which are nice)

**Weaknesses:**

The paper feels a little thin on the contribution mostly consisting of strong empirical results. the underlying idea of a dynamic adjustment is nice but not particularly surprising. I personally very much like the paper, however I could understand if other reviewers find its theoretical contributions too little.

**Questions:**

no questions - see above

---

> ### Author Response · Authors · 2024-11-20
>
> We are glad that the reviewer liked the paper and we are grateful for the notes.
>
> We agree that one of the main merits of the paper is that the proposed adaptive backtracking routines seem to work well in practice, but we find the impression that ``the paper feels a little thin on the contribution mostly consisting of strong empirical results'' unjustified.
> In our view, there are two more important contributions.
> One is that the proposed subroutines are simple, which makes them easy to incorporate by different methods, as illustrated by the four different methods that could benefit from adaptive backtracking in the experiments.
> Given the prevalence of backtracking in optimization methods, presenting a simple way to potentially improve it can have a broad impact.
> Moreover, we believe that the paper also has an important conceptual contribution, which is to take into account the information conveyed by line search criteria more explicitly when choosing step sizes, which could lead to several new line search methods.
>
> Furthermore, it is important to mention if the statement that ``the theory while nice/ok is not particularly spectacular or unexpected but does provide a nice complement to the observed performance in practice'' is true for our paper, then it would also be for many relevant line search methods.
> As argued at the beginning of section 4, the guarantees that we provide for adaptive backtracking are essentially the best one can hope for without making further assumptions on the objective function and without specifying the base method that calls adaptive backtracking.
> We included in Appendix B.4.1 (first paragraph) a quote from (J. Nocedal and S. J. Wright, Numerical optimization, 2006) that corroborates this opinion.
> Also, we reiterate that if the paper can be reproached for perceived lack of theory, then perhaps so could the highly influential original paper that proposed the Armijo condition, which is less than three pages long.
> Notwithstanding, in response to the reviewer's comments we have now added several convergence results in Appendix B.4.
> Most of them follow by substituting the corresponding line search condition and step size lower bounds in standard proofs, with the exception of the convergence result for AGD which is proven with a novel technique, to the best of our knowledge.

---

> > ### Comment · Reviewer_vFwJ · 2024-11-27
> >
> > thanks a lot for the answer - I read the answer and followed the discussion. I am keeping my score.

---

### Official Review · Reviewer_QtPJ · 2024-11-02

**Soundness:** 4
**Presentation:** 4
**Contribution:** 4
**Rating:** 8
**Confidence:** 5

**Summary:**

This paper describes a backtracking strategy suitable to both convex/non-convex and smooth/non-smooth optimisation which relies on the definition of a violation factor adjusting the decrease of the step-size along the iteration. Such factor quantifies the amount at which standard backtracking conditions (e.g., Armijo) are *not* satisfied and is thus cleverly used to choose an optimal decreasing factor. In comparison to, e.g., standard Armijo condition, this avoids unnecessary overdecrease of the step (in case of small factors) and high computational costs (in case of big ones, which makes the decrease slower).
The paper is a pleasure to read: the motivation is clearly given, some examples are used to motivate the strategy used and, before providing the main theoretical results, some triggering empirical evidence is provided. The main sets of experiments reported in the body of the paper are convincing and show improvements with standard backtracking line search w.r.t. to a suitable gain measure related mostly to computational evaluation. Experiments are performed in both convex and non-convex settings as well as in a convex non-smooth scenario where comparisons with FISTA are made.
The presentation of the theoretical results is complete and the appendices are very well written and substantiated.
I congratulate with the authors for such a nice piece of work which, in my opinion, opens the path to many questions and future research directions additionally to the ones highlighted in the final section of this work.

**Strengths:**

The paper is very nicely written, the intuitition behind the proposed adaptive strategy is well described and the numerical tests are convincing. I liked the choice of the authors of presenting first the numerical results raather than the theoretical ones, since the reader is really triggered to keep reading and understanding more. The exhaustive list of experiments reported in the appendix is very convincing and shows that the proposed strategy performs indeed very well, as supported by the thoeretical results which I have checked and, up to some minor typos, appear correct to me.

**Weaknesses:**

The only "weaknesses" I could highlight relate to some (minor) lack of referencing to existing work and/or to some refinement that could be made to guarantee that the proposed strategy maintains (as it seems) the convergence speed of stnadard algorithm also in more 'regular' scenario (e.g., strong convexity). Some minor imprecisions can be easily fixed as I will suggest below.
Upon the modifications suggested (at all, or even partially), I think I would be happy to change my note to 10 as I think that this is a very good piece of work.

**Questions:**

- I think that some further referencing should be added. In these works (DOI1: 10.1137/17M1149390, DOI2: 10.1137/21M1391699) adaptive backtracking strategies that could be generally employed for smooth/non-smooth and convex/strongly-convex optimisation schemes are considered. The main idea there is the possibility of the step-size to increase locally as long as the desired backtrcking conditions (there defined in terms of a Bregman distance of the smooth component) are used. Despite the adaptivity of the approach you propose, your backtracking strategy is still monotone (at least in the, indeed, monotone case). Is this really needed? Couldn't you allow also the local increase of the step-size, check if the condition is violated and, if not, allow it and, if yes, perform your strategy with suitable adjustments? I think that a comparison with these approaches should be added and such extension possibly considered.
The use of non-monotone approaches as the one mentioned above is interesting, for instance, where you have, for some reason, a very bad estimation of \alpha_0  (e.g., very small) which as far as I undertsand, not even with your approach you would be able to inrease along iterations.
- Another approach that is worth mentioning is the one studied here (Malitsky, Mishchenko Adaptive Gradient Descent without Descent
 PMLR 119:6702-6712, 2020) which is computationally very effective and relies on different backtracking rules adaptive to local geometry.
- Your theoretical results aim at quantifying, essentially, the amount of backtracking iterations in comparison with the standard approaches, but can you indeed guarantee that usual convergence rates are obtained whenever the strategy is employed as an 'inner loop' of any underlying algorithm?
- In some sections I found a bit of overlooking on the regularity properties of the problems considered and/or of the schemes employed. This in particular when you considered strongly convex optimistation schemes. In the non-smooth examples you considered AGD schemes adapted to strong convexity (e.g., relying on the knowledge/estimation of strong-convexity parameters) as in the case of Algorithm 4 (where I whink that m is indeed the strong convexity parameter although never mentioned) or not (FISTA). For strongly convex algorithms linear convergence is normally achieved, contrarily to convex ones where only O(1/k^2) speed is obtained. Some more precision on the reasons why you consider one or the other and on how potentially your idea adapts to situations where the strongly convex parameter is unknown (and estimated via restarting strategies, see, e.g., DOI3: 10.1137/23M158961X) should be added. This is interesting, for instance, when one wants to control oscillations due to inertia in accelerated schemes. Interestingly, you don't seem to have that many oscillations. Do you have any guess on why this happens? It is surely related to the adaptation to the local geometry of the function.
- It's a bit surprising that for standard BLS, in the results in Figure 1 and t2, with the same \rho, the convergence curve obtained with a higher value of \beta (higher initial step-size), pink curves, lies *above* and not below the one with smaller step size. Is this due to the 'bad' adjustment of the step-size in the standard backtracking inner routine?
I think it would be very interesting comparing the evolution of the step-size estimation along the iterations, in comparison with the (over)estimation of 2/L (convex) or 1/L (non-convex) used as a fixed reference.




Minor points:
- I would change notation in (5) on page 3, lines 151 and subsequent since, as you said, (5) has to hold for F (capital) while it looks it has to hold only for the smooth part.
- Page 4, line (191) is somehow bizarre (the regularisation parameter can be anyhing), is there any reason why you relate this to \bar{L}?
- What is the stopping criterion for achieveing "designated precision" used in the experiments in section 3?
- What is the interest of using accelerated schemes in the non-convex objective example? It's not clear that they improve at all the speed of convergence right?
- In Figure 3 and the corresponding table you use the value of the non-convex loss functions as a measure of how good one approach is w.r.t. others. This may suggest that convergence to different local minimisers is achieved, althoutgh from the examples it seems that the trajectory is pointing towards the same local minimiser. So why is the loss smaller? Did you stop iterations after a max number of iterations and notice that in one case you got a lower value (closer to the one of the minimiser you are converging to)?
- The two 1d illustrations in Figure 4 are enlightening but not clear at all without looking at the appendix. I would provide some more details in the main body of the paper
- In the Informal Theorem on page 9 (line 477) what is the function "g"?
- I would be careful with the use of the word "restarting" as in the standard optimisation community this is a strategy more concerned with the estimation  of the strongly convex parameter (see for instance DOI4: 10.1007/s10208-013-9150-3) than the step-size. Also, the use of the restarting strategy you propose sounds unnecessarily expensive. Why keep starting with the initial step-size especially when this is too big when one can use the last estimated one? You say that they seem to work much better but I imagine they require more inner checks?

---

> ### Author Response · Authors · 2024-11-20
>
> We really appreciate the reviewer's supportive review, extensive comments and suggestions.
> We hope to have addressed every point raised, but we are happy to take into account any further suggestions that the reviewer might have.
>
> * "**I think that some further referencing should be added**":
> We thank the reviewer for pointing out to us these interesting references, which we now discuss in the "Additional algorithms and increasing step sizes'' part of Section 5.1. These approaches slightly increase the final step size of one iteration and then use it as the initial backtracking step size in the following one, similar to what the two-way method by Truong does.  "**The use of non-monotone approaches as the one mentioned above is interesting, for instance, where you have, for some reason, a very bad estimation of $\alpha_{0}$ (e.g., very small) which as far as I understand, not even with your approach you would be able to increase along iterations.**'' In the memoryless adaptive backtracking variant, the initial step size is constant, therefore the step sizes are generally nonmonotone. Moreover, as we mention in the paper, in principle adaptive adjustments can also be used to increase step sizes, but hybrid approaches are possible as well, where only the backtracking portion of the line search is adaptive, for example. We leave the investigation of further variants of adaptive line search to future work.
>
> * "**Another approach that is worth mentioning is the one studied here (Malitsky, Mishchenko Adaptive Gradient Descent without Descent**"
> We are aware of that interesting paper, but their adaptive method does not actually enforce the same criterion that our adaptive backtracking subroutine does (the Armijo condition in this case), which is why had not mentioned it before.
>
> * "**can you indeed guarantee that usual convergence rates are obtained whenever the strategy is employed as an 'inner loop' of any underlying algorithm?**"
> Yes, adaptive backtracking preserves convergence rates. We have added several such results to the appendix. Most of them follow from standard proofs by plugging in the step size lower bounds and noticing that (when $c \geq 1/2$) the Armijo condition implies the descent lemma, which is the main requirement on the step size for those arguments. The exception is the proof of accelerated convergence for AGD with adaptive backtracking, which to the best of our knowledge is novel.
>
> * "**Some more precision on the reasons why you consider one or the other and on how potentially your idea adapts to situations where the strongly convex parameter is unknown [...] should be added**"
> We chose methods that we felt were representative of different broad algorithmic families and that would show that adaptive backtracking performs well in various settings and is flexible enough to be seamlessly incorporated by many further methods. Indeed, the reference mentioned by the reviewer (now cited in the ``additional algorithms'' paragraph of section 5.1) is an example where it seems that adaptive backtracking could easily replace regular backtracking, since the enforced condition seems to be affine in the step size. "**Interestingly, you don't seem to have that many oscillations. Do you have any guess on why this happens?**'' In addition to its flexibility, adaptive backtracking seemingly picks more appropriate step sizes that end up reducing oscillations, which makes us optimistic that it could work well for methods that adopt nonmonotone step sizes and also simultaneously estimate other parameters.
>
> * "**I think it would be very interesting comparing the evolution of the step-size estimation along the iterations**" Plotting step sizes was a good suggestion and we have now included Figures 12 and 13 in Appendix D.2 on page X that show the corresponding step sizes for Figures 1 and 2. In the new figures, we can see that initially adaptive backtracking returns smaller step sizes than regular backtracking, which seems to have lasting impacts afterwards. Regular backtracking returns the largest feasible step size within a factor of rho, which seemingly is not always good as the step sizes can be excessively large and lead to worse optimization paths (e.g., more zig-zagging).
> For AGD, adaptive backtracking step sizes converge, but regular backtracking step sizes do not necessarily.
> Regular backtracking can only return step sizes that are powers of rho multiplied by the initial step size, therefore when the steady state step size is somewhat in between two grid points, the returned step size can keep oscillating instead of settling down, which can perhaps degrade performance.

---

> > ### Author Response · Authors · 2024-11-20
> >
> > Minor points:
> >
> > * "**(5) has to hold for F (capital)**" Perhaps our phrasing was a little ambiguous but in the original formulation of composite problems in the FISTA paper, (5) indeed holds only for the smooth part of F. As a sanity check, (5) cannot hold for F since it is not differentiable in general.
> >
> > * "**is there any reason why you relate this to $\bar{L}$?**" The idea was to define the strong convexity constant, which in this case is essentially the regularization parameter, in a relative scale with respect to the theoretical upper bound of the Lipschitz constant and the number of examples to have a bit of control over the condition number of the problem.
> >
> > * "**What is the stopping criterion for achieving "designated precision"?**" To produce the experimental data, we simply ran all algorithms for a designated number of iterations that was enough for at the least the best method to achieve a prescribed precision. Then, to quantify the performance of each method, we determined the iteration in which each method reached the prescribed precision and then used that iteration as the final one.
> >
> > * "**What is the interest of using accelerated schemes in the non-convex objective example? It's not clear that they improve at all the speed of convergence right?**" That's right, we only included an accelerated method to diversify the experiment.
> >
> > * "**This may suggest that convergence to different local minimisers is achieved, althoutgh from the examples it seems that the trajectory is pointing towards the same local minimiser. So why is the loss smaller?**" Although the Rosenbrock function is not convex, it has a unique global minimum and no local minima. Indeed, as the plot shows, the iterates of all methods are converging to the same point. The methods that perform better allegedly approach the minimum via a more favorable optimization path.
> >
> > * "**I would provide some more details in the main body of the paper**" We would like to include more details in the main body of the paper but unfortunately we have already reached the page limit, so hopefully with additional details and insight from the appendix the message of the figures will become clearer.
> >
> > * "**In the Informal Theorem on page 9 (line 477) what is the function "g"?**" Sorry, that was a typo. We meant $\psi$, the possibly non-smooth component of the objective. Thanks for the catch!
> >
> > * "**I would be careful with the use of the word "restarting**" Fair point. We changed ``restarting'' to "memoryless" to avoid any confusion.
> > "**Why keep starting with the initial step-size especially when this is too big when one can use the last estimated one? You say that they seem to work much better but I imagine they require more inner checks?**"
> > It is possible that other heuristics for choosing the initial step size could provide a better trade-off between function evaluations and step size, but we chose the presented version for its simplicity.

---

> > > ### Comment · Reviewer_QtPJ · 2024-11-26
> > > **Very good review!**
> > >
> > > Thanks so much for taking into account my review and comments to address. I do appreciate the simplicity of your presentation and the effectiveness of these results and I believe that with these further modifications you have addressed in your revision the paper should definitely be accepted.
> > > I agree that the extension to non-monotone backtracking strategies coulda and should (for page limit) be included in further extensions of this work and that a theoretical explanation on why oscillations are reduced should be given.
> > > I confirm my mark to the authors!

---

> > > > ### Author Response · Authors · 2024-11-26
> > > >
> > > > We are grateful for the reviewer's curiosity and constructive suggestions.
> > > > It seems that we have addressed the points raised but, since the reviewer signaled this possibility, if there are any outstanding issues we can improve on to further increase the paper's score, we are happy to update the manuscript taking them into account.

---

### Official Review · Reviewer_Qk4F · 2024-11-04

**Soundness:** 3
**Presentation:** 2
**Contribution:** 3
**Rating:** 6
**Confidence:** 3

**Summary:**

This paper proposes an "adaptive" version of backtracking line search (BLS) for descent steps in optimization. Namely, while in classical BLS the candidate step-size shrinks by a fixed factor of $\rho$, this paper proposes an adaptive choice of $\rho$ at each iteration which depends on how much the line-search condition is violated. The paper demonstrates in a series of experiments that this adaptive version of BLS seems to be more robust to starting point and usually leads to significant savings in compute-time if compared to classical BLS with a fixed rate. Moreover, they show theoretical results demonstrating that, in convex and/or Lipschitz smooth problems, Adaptive BLS is *effectively* no worse than BLS and the worst-case step-size guarantee are similar.

**Strengths:**

- Usually papers on backtracking line-search tend to focus on effects of non-monotonic searches or on changing the condition of the line-search. This paper proposes a seemingly simple modification to the choice of the backtracking factor that preserves most of the theoretical guarantees. This seems like one of those simple observations that is of interest to the optimization community in general. In fact, I think the contribution is good exactly because of the simplicity: there seems to be a easy modification to BLS that most likely improves it;
- The presentation is relatively clear and easy to understand;
- The variety of experiments gives confidence about the empirical performance of the method if compared to BLS with a fixed parameter;

**Weaknesses:**

I believe the main area of improvement of this paper is its discussion previous work, which is even more important in a paper about such a classical topic.  I will detail two aspects of the paper that I believe could be improved.

- **Discussion of related research work**: Backtracking line-search is a classical topic with a vast amount of work on it (even if most of it is not necessarily on the backtracking factor). Yet, besides references to classical books and the original works on BLS, there is almost no discussion of previous work. Although I am not familiar with the most recent work on line-search, while reviewing this paper I found at least two papers [1,2] that should at least have been cited, and probably this line of research discussed in more depth.

- **Lack of comparison with line-search variants**: The paper focuses on backtracking line-search with a fixed backtracking parameter $\rho$ as the main method they are comparing against. However, there are classical variants of this idea that are more often used in practice, and the main one that comes to mind is polynomial interpolation, and there are likely other heuristics the authors could have at least discussed.  Moreover, a quote from the classical textbook [3, Sec. 3.1] mentions that BLS with a varying backtracking parameter is more often used in practice:

> In practice, the contraction factor $\rho$ is often allowed to vary at each iteration of the line search. For example, it can be chosen by safeguarded interpolation, as we describe later. We need ensure only that at each iteration we have $\rho\in [\rho_{\mathrm{lo}},\rho_{\mathrm{hi}}]$, for some fixed constants $0 < \rho_{\mathrm{lo}} < \rho_{\mathrm{hi}} < 1$.

---
**Summary of review**: I believe the contribution of this paper is simple (in a good way) and relevant to the optimization community. However, the lack of discussion of recent related work and the sole focus on comparing against BLS with a fixed backtracking parameter does not seem to properly situate the contribution in the literature and to paint a fair picture of its impact in line-search methods in general, which is why I'm currently inclined to recommend a rejection of the paper.

I hope to discuss with the authors about what related work they believe is relevant to be discussed and which other methods would be valid comparison points. Finally, I just want to make it clear that I do not believe that the adaptive BLS that this paper proposes needs to outperform all other variants of BLS to be a good contribution, or that the authors need to perform experiments against all other variants. However, there definitely should be a more in-depth discussion of related work and comparison with BLS vairants and heuristics.


---


** References **

[1] de Oliveira, Ivo Fagundes David, and Ricardo Hiroshi Caldeira Takahashi. "Efficient solvers for Armijo's backtracking problem." arXiv preprint arXiv:2110.14072 (2021).

[2] Fridovich-Keil, Sara, and Benjamin Recht. "Approximately exact line search." arXiv:2011.04721 (2020). (This one seems to be a full version of a workshop paper: https://opt-ml.org/papers/2019/paper_17.pdf)

[3] Nocedal, Jorge, and Stephen J. Wright. "Numerical optimization", Second Edition (1999)

**Questions:**

- Do you think it would be interesting to compare adaptive BLS with LS with polynomial interpolation (mainly quadratic or cubic)? It seems to me it would be a more interesting comparison than simply comparing with fixed-parameter BLS, but the authors could disagree with me. Also, I believe there is likely other methods or heuristics used in the literature that should be at least acknowledge an briefly discussed in the paper;

- Are there other relevant research work on BLS that the authors are aware? The two papers I referenced I found on a brief web search, thus I only found somewhat recent work that hasn't been necessarily published yet (although one of them appeared in 2019 as a workshop paper), and I am quite certain that the authors are more better acquainted with the literature in the area than myself;

---

> ### Author Response · Authors · 2024-11-20
>
> We would like to thank the reviewer for taking the time to evaluate our paper and for contributing with relevant references.
>
> * "**Discussion of related research work**":The page limit constraint forced us to make some compromises and we ended up shortening the introduction, perhaps excessively.
> Also, most references are spread over the paper, where we felt was most pertinent, rather than at the beginning.
> For example, on page 2 we mentioned Armijo, Wolfe and nonmonotone conditions, as well as polynomial interpolation, where we cited Nocedal and Wright.
> But the majority of references can be found in the last section, where we thoroughly discuss several connections with recent line search methods.
> That being said, we are glad that the reviewer brought this issue to our attention and we will add a more thorough discussion of recent line search references, including the first two references.
>
> * "**Lack of comparison with line-search variants**":  Our focus in this paper is on backtracking subroutines that enforce a specific condition (Armijo's or the descent lemma.)
> The second paper mentioned by the reviewer does not enforce any specific condition, but is a rather ``general purpose'' subroutine that aims at finding an approximation of the step size returned by exact line search.
> Since the line search condition plays a fundamental role on the behavior of an optimization method, we did not consider appropriate to compare adaptive backtracking with their line search subroutine.
> On the other hand, the first reference does enforce the Armijo condition, but apparently did not release any code.
> Moreover, they also only compared their subroutine with regular backtracking.
> In particular, they do not consider a polynomial interpolation subroutine.
> This is also the case in the two papers mentioned by reviewer QtPJ, which also mostly compare the benefits of their subroutines with respect to standard backtracking subroutines enforcing the descent lemma.
> In fact, these two papers improve on the performance of the base method (FISTA), which originally implemented regular backtracking to enforce the descent lemma, without ever mentioning polynomial interpolation. Furthermore, we have not found any standard packages for polynomial interpolation to enforce the Armijo condition, because it is often not done.
> With this in mind, we believe that our decision of not including a comparison with polynomial interpolation is justified.
> Indeed, our goal was to showcase the potential benefits of adaptive backtracking relative to regular backtracking in terms of performance improvement, irrespective of the choice of $\rho$. This is in particular because standard backtracking is still the de facto standard given its simplicity and  ease of implementation. Our method similarly simple and easy to implement.
>
> Questions
> * We do believe it would be an interesting comparison, but given the reasons mentioned in the first bullet above, we instead decided to dedicate our available computational effort to thoroughly sweeping over several values of $\rho$ for regular backtracking.
> * Yes, there are quite a lot of exciting recent developments in line search that provide several lines of work for future research, as we discuss in section 5.

---

> > ### Comment · Reviewer_Qk4F · 2024-11-23
> >
> > I'd like to start by thanking the authors for taking the time to respond to my and other reviewers' concerns.
> >
> > Reading the first point on the authors' response to my review, I want to make it clear that I do not think that the authors do a poor job of citing the original and classical work on line-search. My point is that, even though I am not an expert in line-search, I believed the paper was lacking a more thorough discussion of more recent work on line-search. As I mentioned in my review, the papers I mentioned were not papers I knew beforehand, but ones that I could find fairly easily and seemed relevant to this paper. The point was to show the lack of discussion of more recent work on backtracking LS. Personally, I thought there would be work on heuristics used in practice on using different values of $\rho$, but I could not quickly find works discussing such heuristics. If the authors know about work along these lines, it would be very interesting to discuss.
> >
> > On the point about polynomial interpolation (PI), I understand that my suggestion that it would be a good comparison point was not great: PI makes sense if we have some condition lower bounding the step-size, and in the case of the Armijo and the descent lemma conditions, any sufficiently small step-size satisfies the condition (and, as the authors argue, PI is not even implemented with just the Armijo condition in standard optimization packages). I do still believe that it would be interesting to compare adaptive LS with something that would be "clearly better" such as exact line search (in a problem where it is feasible to do so), but I'd consider this a personal preference/curiosity and not something that affects the quality of the paper.
> >
> > Given the authors' response, I think I have a better understanding of the significance of the contribution, and thus I am changing my contribution score from 2 to 3. My point on the lack of a better discussion of related work I believe is still valid, and at least another reviewer had a similar opinion. Even if the discussion of related work (and how it affects some of the choices in experiment design, for example) is deferred to the appendix, it seems that it should be added to the paper.

---

> ### Author Response · Authors · 2024-11-25
>
> We thank the reviewer for considering our response.
> To fully address the final outstanding concern regarding proper mention to related work, we have now added section 2.3 to the main paper, in which we summarize the above discussions.
> We hope that this addition further improves the presentation of the paper and allows the reviewer to adjust their score accordingly.
> If the reviewer has any additional feedback to help us improve the paper's soundness, presentation and contribution scores, we would appreciate the opportunity to address them.

---

> > ### Comment · Reviewer_Qk4F · 2024-11-25
> >
> > Great! I really appreciate the authors taking the time to engage in the discussion. I will likely not have the time to carefully check the new discussion before the end of the discussion period, but I really believe a better discussion of related work will help readers better appreciate the work. I will certainly have the time to engage in the discussion during the reviewer discussion period.

---

### Official Review · Reviewer_RAP1 · 2024-11-04

**Soundness:** 2
**Presentation:** 2
**Contribution:** 2
**Rating:** 5
**Confidence:** 3

**Summary:**

This paper studies the backtracking line search commonly used for step size selection in optimization algorithms. The main focus is to propose an adaptive approach for setting the decay parameter when searching for the step size. The authors consider several examples (e.g., Armijo, descent lemma) to illustrate the concept, which is straightforward and presented in Algorithm 2. They also conduct extensive numerical simulations to demonstrate the effectiveness of the adaptive approach.

**Strengths:**

Backtracking line search is a fundamental task in general optimization algorithms. The numerical simulations show promising potential for the adaptive method.

**Weaknesses:**

As my work primarily focuses on the theoretical aspects of optimization, my comments and concerns mainly relate to the theoretical analysis of the proposed adaptive approach. Broadly speaking, the theorems presented in Section 4 provide limited insight into the advantages of the new approach compared to the conventional backtracking line search in Algorithm 1. I will elaborate on this and discuss other concerns below.

* My main technical concern is with the lower bound of the adaptive factor, which will affect the overall complexity of the algorithm, not just the number of function evaluations in a single iteration step. As mentioned by the authors in Line 395, the search procedure should ensure a sufficiently large step size to achieve fast convergence. Following this, the authors argue that their construction guarantees that the adaptive factor is bounded away from zero. However, this is not sufficient to ensure convergence of the (outer-loop) algorithm iteration sequence, as the step size can be arbitrarily small compared to the step size selected by "regular" line search (for example, in the descent lemma setting, the regular search will produce a step size lower bounded multiplicatively by the factor $\rho$). Indeed, using some standard theorems for smooth optimization, the adaptive factor appears in the final complexity guarantee and may not be easily controlled. Therefore, an important task here is to investigate the overall evaluation complexity when the proposed adaptive line search is used.

* As for the theorems presented in Section 4, specifically Propositions 1, 2, and the two informal theorems, it seems that these results do not demonstrate the advantage of using an adaptive approach compared to the regular search method. These results essentially state that, under certain assumptions, the adaptive approach is not worse than the regular method in terms of the number of function evaluations in a single step. Also, as mentioned above, it is unclear how this affects the total number of iterations in the outer loop.

* The first question that likely arises after seeing Algorithm 2 is how to design the adaptive factor $\hat{\rho}$. The authors demonstrate some possible constructions in Section 2, but these are mostly reformulations of known methods. Thus, is there a principled way to construct this adaptive factor, even for specific settings?

**Questions:**

See above.

---

> ### Author Response · Authors · 2024-11-20
>
> We would like to thank the reviewer for reading our paper and for the comments, which we address below.
>
> * "**as the step size can be arbitrarily small compared to the step size selected by `regular' line search**": It seems there was a misunderstanding here. For both the Armijo condition and the descent lemma, the step size itself enjoys nontrivial positive lower bounds, presented in the last sentences of the informal theorems. Namely, for the Armijo condition with $L$-smooth objective and gradient related descent directions with constants $c_{1}$ and $c_{2}$, the step-size returned by our adaptive procedure satisfies
> $$
> \alpha_{k} \geq \min( \alpha_0, \frac{\rho 2(1 -c)c_{1}}{Lc_{2}^{2}} ),
> $$
> which matches the lower bound for the step-size returned via regular backtracking (please consult proofs for more details.) Likewise, the step size lower bound for the step-size returned by the descent lemma is
> $$
> \alpha_{k} \geq \min( \alpha_{0}, \rho/L ).
> $$
> With these lower bounds we are able to establish worst-case convergence results for specific base methods that call adaptive backtracking, which can be found in appendix B.4 of the updated manuscript.
> Some simplified instances of these results are:
>
> **Gradient descent**
> If $f$ is $L$-smooth, $\alpha_{0}>1/L$ and $c= 10^{-4}$ for GD (where $c_1 = c_2 = 1$)
> $$
> f(x_{k}) - f(x^{\ast})
> \leq \frac{L}{\rho}\frac{\Vert x_{0} - x^{\ast} \Vert^{2}}{k}.
> $$
> **Accelerated gradient descent**
> If $f$ is $L$-smooth and $m$-strongly convex, $\alpha_{0} < 1/m$ and $c=1/2$, then
> $$
> f(y_{k+1}) - f(x^{\ast})
> \leq \Biggl( \frac{\sqrt{Q} - \sqrt{\rho}}{\sqrt{Q}} \Biggr)^{k}\frac{Q^{2}}{\rho^{2}}\frac{L+m}{2}\Vert x_{0} - x^{\ast} \Vert^{2},
> $$
> where $Q=L/m$ denotes the condition number of the problem.
> Complete statements and proofs of these and more results are now included in Appendix B.4.
> Most of them are straightforward extensions of standard proofs, except for the AGD result, which we prove with a technique that is novel, to the best of our knowledge.
>
> * "**These results essentially state that, under certain assumptions, the adaptive approach is not worse than the regular method in terms of the number of function evaluations in a single step.**"
> Iterates are bounded for most choices of descent direction (e.g., GD), therefore to prove that adaptive backtracking requires no more function evaluations than regular backtracking, it suffices that the gradient be continuous, not even that a global smoothness constant $L$ exists.
> Furthermore, to derive explicit lower bounds on the returned step size, the smoothness assumption is necessary for both adaptive and non-adaptive backtracking and is standard in most convergence proofs of gradient-based methods (e.g., theorem 3.2 of [1]).
>
> * "**The authors demonstrate some possible constructions in Section 2, but these are mostly reformulations of known methods. Thus, is there a principled way to construct this adaptive factor, even for specific settings?**"
> We argue that requiring that $\hat{\rho}(v(\alpha_{k})) \in (0,\rho)$ while matching the lower bound of standard backtracking is a principled way of constructing the adaptive factor.
> Notwithstanding, whenever the line search criteria are affine in the step size, as in the Armijo condition and in the descent lemma, one can isolate the step size on one side of the inequality, take the other side as a candidate for the adaptive step size and then extract $\rho$ and $v$ from that expression accordingly.
> Finally, we would like to stress that none of the constructions that we present is a reformulation of a known method.
> In fact, the base methods (GD, AGD, etc.) are known, but not the line search procedure that is used to generate feasible step sizes.
>
> In summary, our theoretical results match those for backtracking line search (in addition to showing our factor is more aggressive at finding feasible step sizes) and we perform extensive experiments to showcase its advantage.
> Our procedure can be applied to many baseline algorithms and we demonstrated its advantage on several.
> We also rigorously show why we couldn't hope for much more from a theoretical characterization of a backtracking scheme unless we narrow the set of problem instances we consider and specify the base method that is calling backtracking.
> Notwithstanding, in response to the reviewer's comments we have now added several convergence results to the appendix.
> Most of them follow by substituting the corresponding line search condition and step size lower bounds in standard proofs, with the exception of the convergence result for AGD which is proven with a novel technique, to the best of our knowledge.
>
> Finally, we hope that the reviewer would agree that if the paper can be reproached for perceived lack of theory, then perhaps so could the original highly influential paper that proposed the Armijo condition, which is less than three pages long with a similar amount of theory.

---

> > ### Comment · Reviewer_RAP1 · 2024-11-25
> >
> > Thank you to the authors for their response. Regarding the first point, I understand that there are many interesting (and also well-known) cases where a lower bound can be derived. In the submission, the proposed framework is a general and adaptive factor. However, it may be very challenging to prove a lower bound for such a general factor (I reiterate that I am fully aware that such a lower bound is possible for certain well-known special cases). To clarify, I am not asking for more complex or potentially ornamental theoretical results. Elegance and simplicity are always the most desirable qualities. What I was hoping to see was some structural assumptions—on the function, the adaptive factor, or other aspects—that would clearly demonstrate the advantage of adopting an adaptive approach over simply using a fixed factor.
> >
> > That said, and after considering the comments from other reviewers, my feedback is primarily from a theoretical perspective and may be partial or biased. Since others appear to be impressed with the empirical results and heuristics, I will slightly increase my score to reflect this, but I will not raise it to a positive one.

---

> > > ### Author Response · Authors · 2024-11-26
> > >
> > > We thank the reviewer for considering our response.
> > > We have now added an example to the updated manuscript (example 3 in appendix B.1 pg. 16) which shows that **no line search procedure can be provably better than (regular) backtracking to enforce any set of line search criteria that includes the Armijo condition or the descent lemma, for any class of functions that includes quadratics**.
> > > That is, backtracking can be optimal to enforce Armijo and descent lemma for specific choices of $\rho$ and initial step sizes/Lipschitz constant estimates, but often it is not.
> > > Therefore, this impossibility result establishes that matching the theoretical guarantees of regular backtracking is the strongest result that can be proven for any reasonable line search setting, such as smooth objectives (C1, Lipschitz, etc., which contain quadratics.)
> > > We hope this addition satisfies the reviewer's request for more decisive theory.
> > > Also, we would appreciate any concrete suggestions to improve the soundness and presentation of the paper, which have been given low scores without an explicit explanation.
> > >
> > > Finally, we would like to emphasize that our paper takes the following standard research approach: we (1) offer design principles that are likely to result in better performance, (2) prove that we can adhere to these principles in a way that continues to guarantee worst-case performance (we maintain worst case function evaluation and worst-case outer-loop complexity), and (3) illustrate with extensive experiments that these design principles are often very impactful (with speed-ups averaging over 30\% on most experiments.)
> > > This kind of research agenda is ubiquitous and is very impactful in optimization and ML (for example, it is standard in almost all applied results in non-convex optimization) and ICLR in particular has several examples of these highly-cited and utilized works in their venue (e.g. the ADAM paper [https://arxiv.org/pdf/1412.6980] with almost 200K citations.)

---

### Author Response · Authors · 2024-11-21
**Global response**

We thank the reviewers for the many helpful comments and suggestions.
In the following, we respond to every single point raised by the reviewers as best as we can, but we are happy to consider any further suggestions that the reviewers might have.
Before addressing each review individually, we would like to comment on the two most common concerns raised by the reviewers.

* **The simplicity of the method (and accompanying perceived lack of theory.)**
While the majority of the reviewers seem to appreciate that the simplicity of our adaptive approach to backtracking is a virtue, there seems to be a sense that the paper could have used more theory.
We would like to emphasize that the theory provided in the submitted version of the paper contained the strongest essential results that one could expect of a line search procedure under such mild assumptions, without either strengthening the assumptions or specifying the base algorithm that calls the backtracking subroutine.
In particular, the theoretical guarantees for adaptive backtracking are at least as good as those of regular backtracking, including step size lower bounds.
Furthermore, we argue that if the submitted version of our paper could be reproached for allegedly lacking theory, then so could the highly influential paper that introduced the Armijo condition, which is less than three pages long.
Notwithstanding, we have now included several convergence results, for specific base algorithms and in general, illustrating that adaptive backtracking preserves the convergence rates of the base methods. In addition, the proof we provide for AGD is novel, as far as we know (see Appendix B.4). Some simplified instances of these results are:
**Gradient descent**
If $f$ is $L$-smooth, $\alpha_{0}>1/L$ and $c= 10^{-4}$ for GD, then adaptive line search satisfies:
$$
f(x_{k}) - f(x^{\ast})
\leq \frac{L}{\rho}\frac{\Vert x_{0} - x^{\ast} \Vert^{2}}{k}.
$$
**Accelerated gradient descent**
If $f$ is $L$-smooth and $m$-strongly convex, $\alpha_{0} < 1/m$ and $c=1/2$, then adaptive line search satisfies:
$$
f(y_{k+1}) - f(x^{\ast})
\leq \Biggl( \frac{\sqrt{Q} - \sqrt{\rho}}{\sqrt{Q}} \Biggr)^{k}\frac{Q^{2}}{\rho^{2}}\frac{L+m}{2}\Vert x_{0} - x^{\ast} \Vert^{2},
$$
where $Q=L/m$ denotes the condition number of the problem.

* "**The contextualization of our work (e.g. connection to other line search methods)**" We appreciate the relevant references that two reviewers have mentioned.
These references are now cited in section 2.3 of related work, which we have added to the main paper to reflect the discussions we have had with the reviewers.
On the other hand, we believe that our comprehensive suite of experiments does not lack comparisons with further methods, for several reasons.
Some of the line search methods proposed by those references do not actually enforce the same line search criteria as that of our adaptive subroutines, which leads entirely different algorithms.
Our goal was to showcase the benefits of adaptive backtracking over regular backtracking as the line search procedure to enforce specific line search criteria, instead of championing a particular line search criterion, given regular backtracking is still the most commonly used line search routine, in part due to its simplicity.
In addition, there was no publicly available code for one reference mentioned, which also only included comparisons with regular backtracking.
Similarly, all references mentioned only included comparisons with regular backtracking.
Furthermore, we see other references as candidates for incorporating adaptive backtracking, which we are excited to investigate in future work.

---

### Meta-Review · Area_Chair_qywr · 2024-12-23

**Metareview:**

The paper proposes a novel variant of the backtracking line search approach for setting the step size for unconstrained optimization problems. The standard backtracking line search decreases the step size by a fixed constant factor until some criterion such as the Armijo criterion is satisfied. The current paper proposes a modification of this approach that replaces the fixed constant factor by a factor that takes into account how violated the chosen criterion is. This approach is demonstrated for two criteria, the Armijo criterion and the descent lemma for proximal problems. The paper demonstrates the practical benefits of the proposed modification on several datasets. The paper also provides a theoretical analysis for convex optimization problems showing a comparable guarantee to the standard backtracking line search.

The problem addressed in this work is a well motivated one with the potential to significantly impact practical applications.  The reviewers appreciated the simplicity of the proposed method and its performance in the experimental evaluation. The reviewers noted that the empirical results are strong and they are the main strength of the work. On the theoretical front, there were several weaknesses that were noted by the reviewers. The theoretical analysis only establishes that the proposed method is no worse than the standard backtracking line search, and the paper does not provide a theoretical result showing a strict improvement in the worst case over the standard method (under some structural assumptions). Additionally, the method is demonstrated and analyzed in two main settings and on a case by case basis, and it seems unclear how to set the adaptive scaling and theoretically show that the step size remains sufficiently large beyond these two settings. Thus several of the reviewers were concerned that the method is primarily a heuristic improvement for the two specific settings considered and the overall contribution may be somewhat limited. The authors revised the paper to provide additional theoretical grounding for the proposed methods. The author response also indicated that the approach could be used in principle in broader settings. Given the potential for a high practical impact, I support accepting this paper if there is room.

**Additional Comments On Reviewer Discussion:**

There was an extensive discussion between the authors and the reviewers. The authors revised the manuscript based on the reviewers' feedback. These revisions included the inclusion of additional discussion of related work and providing additional theoretical justification for the algorithms, including an example where it is not possible to improve over the standard backtracking line search. These changes improved the manuscript and provided additional theoretical justification, which factored into the decision.

---

### Decision · Program_Chairs · 2025-01-22

Accept (Poster)